# PRDM16-dependent antigen-presenting cells induce tolerance to gut antigens

Liuhui Fu[1,6], Rabi Upadhyay[1,2,6], Maria Pokrovskii[1,3,6], Francis M. Chen[1], Gabriela Romero-Meza[1,4], Adam Griesemer[5] & Dan R. Littman[1,2,4✉]

The gastrointestinal tract is continuously exposed to foreign antigens in food and commensal microorganisms with potential to induce adaptive immune responses. Peripherally induced T regulatory (pT$_{reg}$) cells are essential for mitigating inflammatory responses to these agents[1-4]. Although RORγt[+] antigen-presenting cells (APCs) have been shown to programme gut microbiota-specific pT$_{reg}$ cells[5-7], their definition remains incomplete, and the APC responsible for food tolerance has remained unknown. Here we identify an APC subset that is required for differentiation of both food- and microbiota-specific pT$_{reg}$ cells and for establishment of oral tolerance. Development and function of these APCs require expression of the transcription factors PRDM16 and RORγt, as well as a unique *Rorc(t) cis*-regulatory element. Gene expression, chromatin accessibility, and surface marker analysis establish the pT$_{reg}$-inducing APCs as myeloid in origin, distinct from type 3 innate lymphoid cells, and sharing epigenetic profiles with classical dendritic cells, and designate them PRDM16[+]RORγt[+] tolerizing dendritic cells (tolDCs). Upon genetic perturbation of tolDCs, we observe a substantial increase in food antigen-specific T helper 2 cells in lieu of pT$_{reg}$ cells, leading to compromised tolerance in mouse models of asthma and food allergy. Single-cell analyses of freshly resected mesenteric lymph nodes from a human organ donor, as well as multiple specimens of human intestine and tonsil, reveal candidate tolDCs with co-expression of *PRDM16* and *RORC* and an extensive transcriptome shared with tolDCs from mice, highlighting an evolutionarily conserved role across species. Our findings suggest that a better understanding of how tolDCs develop and how they regulate T cell responses to food and microbial antigens could offer new insights into developing therapeutic strategies for autoimmune and allergic diseases as well as organ transplant tolerance.

APCs have pivotal roles in orchestrating immune responses. They direct T cell outcomes, ranging from diverse effector to suppressive programmes, according to potential threats from pathogenic microorganisms and the environmental context[8,9]. The adaptive immune response has evolved to be tolerant to self-antigens and to suppress responses to antigens that are present in the diet and in the mutualistic microbiota[10,11]. Tolerance to at least some commensal microorganisms with the potential to induce inflammation is mediated by pT$_{reg}$ cells, which are programmed by RORγt[+] APCs[5-7] (hereafter RORγt-APCs). Although these APCs express CD11c and ZBTB46, canonical markers of dendritic cells, they appear to have distinct features and have not been well characterized. They have been proposed to be MHC class II (MHCII)-expressing type 3 innate lymphoid cells (ILC3s) or subsets of Janus cells and Thetis cells, some of which express AIRE[5-7]. Tolerance to dietary antigens is established through induction in the proximal alimentary tract of pT$_{reg}$ cells that restrain inflammatory responses to antigen encountered locally or distally. It is not known whether differentiation of such regulatory T (T$_{reg}$) cells is dependent on similar tolerance-inducing APCs. Classical dendritic cells (cDCs)—particularly cDC1—have been proposed to be inducers of dietary antigen-specific pT$_{reg}$ cells, and it has been suggested that there is redundancy for this function among different types of APCs[12-14].

Here we set out to determine whether the nuclear receptor RORγt, which is known to have critical roles in thymopoiesis, peripheral T cell differentiation and development of innate lymphoid cells (ILCs), is similarly required for the development and/or function of the pTreg-inducing APCs. We also investigated the role of RORγt-APCs in the induction of tolerance to oral antigen. We found that RORγt drives the development of a unique PRDM16-expressing dendritic cell-like subset, and that both RORγt and PRDM16 are required for these APCs to induce pT$_{reg}$ cells that regulate inflammatory responses to food and microbiota antigens. These APCs share many features with classical dendritic cells, are present in mice across different ages from neonate to adult, and are also found in human tissues. We therefore designate

[1]Department of Cell Biology, New York University School of Medicine, New York, NY, USA. [2]Perlmutter Cancer Center, NYU Langone Health, New York, NY, USA. [3]Calico Life Sciences, South San Francisco, CA, USA. [4]Howard Hughes Medical Institute, New York, NY, USA. [5]NYU Langone Transplant Institute, NYU Langone Health, New York, NY, USA. [6]These authors contributed equally: Liuhui Fu, Rabi Upadhyay, Maria Pokrovskii. ✉e-mail: dan.littman@med.nyu.edu

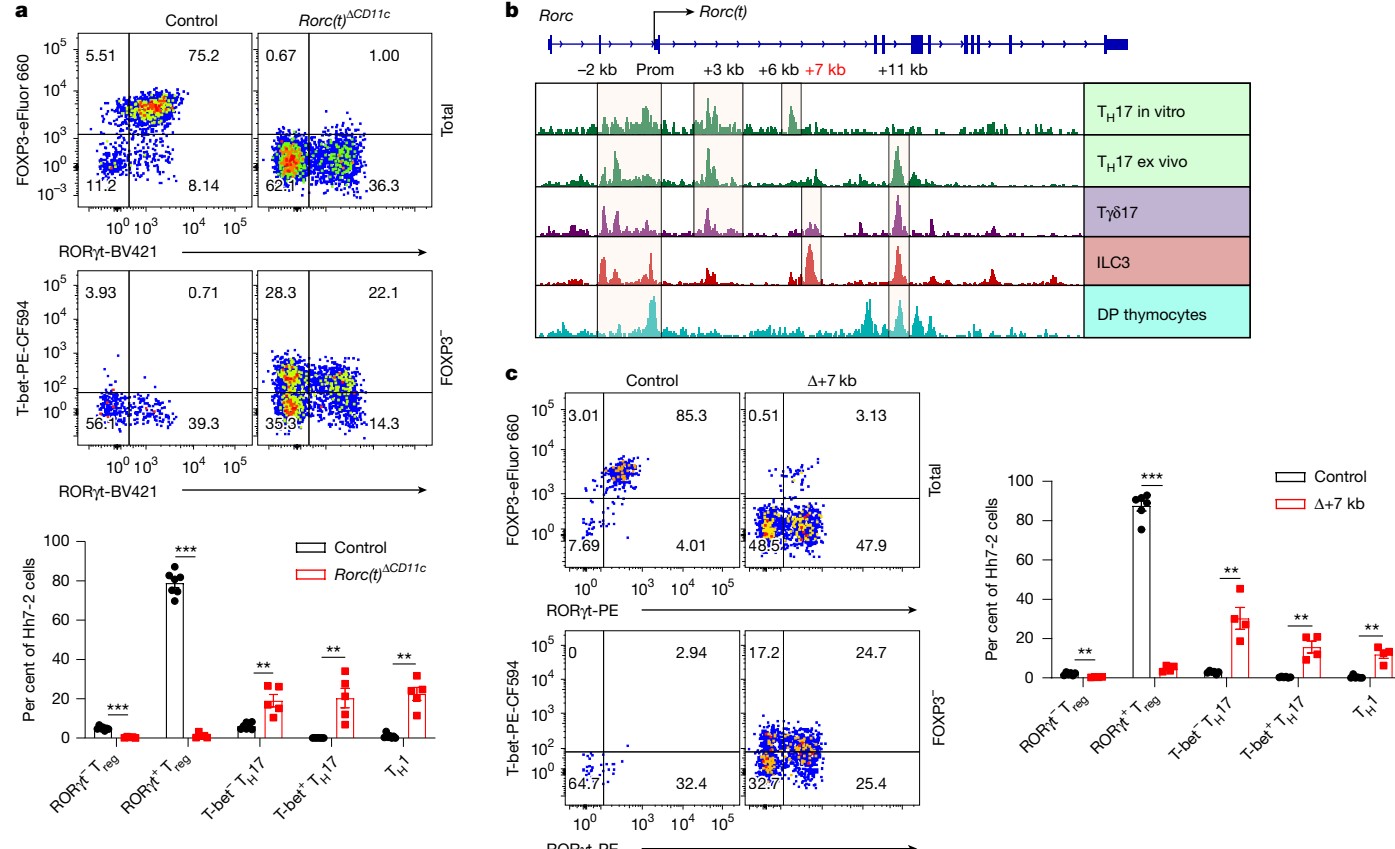

**Fig. 1 | RORγt is required by tolerance-inducing APCs to promote microbiota-specific pT_reg cell differentiation. a**, Representative flow cytometry plots (top) and frequencies (bottom) of *Hh*-specific pT_reg (FOXP3⁺RORγt⁺/⁻), T_H17 (FOXP3⁻RORγt⁺T-bet⁺/⁻) and T_H1 (FOXP3⁻RORγt⁻T-bet⁺) cells in the LILP of *Hh*-colonized control (*Rorc(t)^fl/gfp*, *Rorc(t)^wt/gfp* and *Cd11c^cre Rorc(t)^wt/gfp*; *n* = 7) and *Rorc(t)^ΔCD11c* (*Cd11c^cre Rorc(t)^fl/gfp*; *n* = 5) mice at 14 days after adoptive transfer of naive Hh7-2tg CD4⁺ T cells. The top row of flow cytometry plots is gated on total Hh7-2tg cells (CD45⁺B220⁻TCRγδ⁻TCRβ⁺CD4⁺Vβ6⁺CD90.1⁺) and the bottom row is gated on FOXP3⁻ Hh7-2tg cells. **b**, Bulk ATAC–seq data showing accessible regions in the *Rorc* locus of several RORγt-expressing cell types, including double-positive (DP; CD4⁺CD8⁺) thymocytes, in vitro differentiated T_H17 cells and SILP-derived T_H17 (TCRβ⁺CD4⁺IL23R-GFP⁺) cells, Tγδ17 (TCRγδ⁺IL23R-GFP⁺) cells and presumptive ILC3s (Lin⁻IL7R⁺Klrb1b⁺NK1.1⁻). Prom, promoter. **c**, Phenotype of *Hh*-specific T cells in the LILP of *Hh*-colonized control (*n* = 6) and Δ+7 kb (*n* = 4) mice at 14 days after adoptive transfer of naive Hh7-2tg CD4⁺ T cells. The flow cytometry plots are gated on total (top) and FOXP3⁻ (bottom) Hh7-2tg cells. Data in **a** are pooled from two independent experiments. Data in **c** are representative of two independent experiments. Data are mean ± s.e.m. Unpaired two-sided *t*-test. *P < 0.05, **P < 0.01, ***P < 0.001; NS, not significant.

them as PRDM16⁺RORγt⁺ tolDCs that have critical roles in immune homeostasis and whose dysfunction is likely to contribute to multiple inflammatory and allergic diseases.

## Tolerance-inducing APC function requires RORγt

The tolerance-inducing APCs were identified after genetic targeting by RORγt-Cre of conditional alleles for MHCII, α_Vβ_8 integrin and CCR7 resulted in loss of pT_reg cells specific for microbiota antigens[5–7]. Although these studies indicated that the relevant APC expresses RORγt during ontogeny, it remained uncertain whether RORγt itself is required for these cells to develop or perform their function. We previously showed that inactivation of the same target genes in CD11c-Cre mice also resulted in loss of pT_reg-inducing APC function[5].

To investigate a potential role for RORγt in these APCs, we therefore inactivated *Rorc(t)* in CD11c-Cre mice and determined the fate of T cells specific for the large intestinal pathobiont *Helicobacter hepaticus* (*Hh*). Naive *Hh*-specific CD4⁺ T cells from Hh7-2 T cell receptor (TCR) transgenic mice were transferred into *Hh*-colonized mice. Two weeks after transfer, Hh7-2 T cells in the large intestine lamina propria (LILP) and mesenteric lymph nodes (mLN) of control mice displayed a predominance of pT_reg cells expressing both RORγt and FOXP3 (Fig. 1a and Extended Data Fig. 1a,b). By contrast, in the LILP and mLN of

*Cd11c^cre Rorc(t)^fl/gfp* (*Rorc(t)^ΔCD11c*) mice, the differentiation of adoptively transferred *Hh*-specific pT_reg cells was abrogated. Instead, these mice exhibited an increase in RORγt- and T-bet-expressing *Hh*-specific T cells (Fig. 1a and Extended Data Fig. 1a,b), indicative of a shift towards a pro-inflammatory profile. Similarly, among endogenous T cells in these mutant mice there were fewer RORγt⁺ pT_reg cells and substantially more inflammatory T helper 17 (T_H17) and T helper 1 (T_H1) cells (Extended Data Fig. 1c). These results indicate that RORγt expression in CD11c lineage APCs is required for them to direct gut microbiota-specific pT_reg cell differentiation.

## Lineage-specific *Rorc(t) cis* elements

RORγt is a transcription factor whose expression is largely confined to diverse lymphoid lineage cells, in which it contributes to distinct phenotypic programmes[15–20]. Because different *cis*-regulatory elements (CREs) within a gene locus can govern its cell-specific expression, as best exemplified by the erythroid-specific enhancer in *Bcl11a*[21], a therapeutic target for sickle cell disease, we hypothesized that distinct CREs within the *Rorc* locus may selectively modulate expression in the RORγt⁺ lineages, including the pT_reg-inducing APCs. Although previous research has delineated several CREs involved in RORγt expression in T_H17 cells and ILC3s, *Rorc* regulatory regions in other cell types that express the

transcription factor remain less well characterized[22–24]. To identify cell-type-specific regulatory sequences, we conducted bulk assay for transposase-accessible chromatin with sequencing (ATAC–seq) analyses in several RORγt-expressing cell types, including CD4+CD8+ thymocytes, in vitro differentiated $T_H17$ cells, and small intestine lamina propria (SILP)-derived $T_H17$ cells, Tγδ17 cells, and presumptive ILC3s. These studies revealed distinct patterns of chromatin accessibility within the *Rorc* locus across the different cell types (Fig. 1b). Notably, regions situated +6 kb and +7 kb from the *Rorc(t)* transcription start site exhibited pronounced accessibility in in vitro differentiated $T_H17$ cells and intestinal ILC3, respectively (Fig. 1b). Additionally, the +11 kb element exhibited open chromatin configuration in all RORγt+ cell types, with the notable exception of in vitro differentiated $T_H17$ cells (Fig. 1b), consistent with our previous findings[23].

Further studies using dual reporter BAC transgenic mice specifically lacking a +3 kb element (Tg(Δ+3 kb *Rorc(t)*-mCherry);*Rorc(t)*[wt/gfp]) indicated that *Rorc(t)* +3 kb is a pivotal enhancer in $T_H17$ and Tγδ17 cells in vivo, as well as in vitro differentiated $T_H17$ cells, but not in ILC3 (Extended Data Fig. 2a,b). To further explore the functional importance of the *Rorc(t)* +6 kb and +7 kb elements, we engineered mice with deletions of these sequences. *Rorc(t)* +6 kb[−/−] (Δ+6 kb) mice exhibited a significant reduction in SILP $T_H17$ cells and RORγt expression, with intact ILC3 and Tγδ17 populations, and their naive CD4 T cells did not upregulate RORγt under $T_H17$ differentiation conditions (Extended Data Fig. 2c–f,h). By contrast, *Rorc(t)* +7 kb[−/−] (Δ+7 kb) mice exhibited reduced RORγt+ ILC3s and Tγδ17 cells with lower RORγt expression, but normal $T_H17$ cells in vivo and in vitro (Extended Data Fig. 2c–e,g,i), consistent with a role of this element only in innate-type lymphocytes.

The ILC3 subsets in the SILP and LILP of Δ+7 kb mice were skewed towards NCR1+ ILC3s (NCR1 is also known as NKp46), with a reduction in CCR6-expressing lymphoid tissue inducer (LTi)-like ILC3s (Extended Data Fig. 2j,k). Notably, the number of Peyer's patches remained unchanged (Extended Data Fig. 2l), indicating that RORγt-dependent LTi cells in mutant mice maintain sufficient functionality to support lymphoid organ development. When the Δ+7 kb mice were challenged with the enteric pathogen *Citrobacter rodentium*, there was no difference in bacterial titres and weight loss compared with wild-type controls, and there was no defect in IL-22 production (Extended Data Fig. 2m–o), which is essential for bacterial clearance[25]. These results suggested that mature ILC3s in the LILP of Δ+7 kb mice retain functional capacity in response to *C. rodentium*, despite the reduction or loss of RORγt, which is required early for ILC3 development and for repression of T-bet, which in turn promotes expression of NCR1 and transition to the ILC1 phenotype[26]. This finding is consistent with previous studies that show that whereas RORγt is crucial for restraining transcriptional networks associated with type 1 immunity, it is not essential for robust IL-22 production among mature ILC3s[27,28].

## A tolerance-inducing APC-specific *Rorc(t)* CRE

Despite the suggested maintenance of ILC3 function in the Δ+7 kb mice, there was severe disruption in these mice of differentiation of adoptively transferred *Hh*-specific $pT_{reg}$ cells in the LILP and mLN, accompanied by an expansion of inflammatory $T_H17$ and $T_H1$ cells (Fig. 1c and Extended Data Fig. 3a,b). There was also a reduction of endogenous RORγt+ $pT_{reg}$ cells and an increase of $T_H17$ and $T_H1$ cells in the *Hh*-colonized mice (Extended Data Fig. 3c). Thus, the *Rorc(t)* +7 kb CRE is required for RORγt-APCs to promote microbiota-specific $pT_{reg}$ cell differentiation even though known ILC3 and LTi cell functions remain intact, which raises the possibility that such APCs belong to a distinct cell lineage to that of ILCs.

Because RORγt is expressed in multiple cell types in the immune system, the T cell phenotype observed in the Δ+7 kb mice could stem from direct or indirect effects. To clarify which cell types are affected by the deletion of the CRE, we established competitive bone marrow chimeric mice by co-transplanting CD45.1/2 wild-type bone marrow along with either CD45.2 wild-type control or CD45.2 Δ+7 kb bone marrow into irradiated CD45.1 wild-type recipients (Extended Data Fig. 3d). We conducted a detailed analysis of donor chimerism across various intestinal immune cell subsets, including RORγt+ $pT_{reg}$, ILC3, $T_H17$ and Tγδ17 cells, normalizing these measurements to splenic B cells as an internal control. As expected, CCR6+RORγt+ ILC3s derived from the CD45.2 mutant donor were underrepresented, whereas NCR1+RORγt+ ILC3 from the same donor were increased. However, RORγt+ $pT_{reg}$, $T_H17$ and Tγδ17 cells repopulated to similar extents in the mixed chimeras (Extended Data Fig. 3e,f). Analysis of RORγt levels within RORγt+ cell populations confirmed that that *Rorc(t)* + 7 kb intrinsically regulates RORγt expression in ILC3 and Tγδ17, but not in RORγt+ $pT_{reg}$ and $T_H17$ cells (Extended Data Fig. 3g–i). Thus, the *Rorc(t)* + 7 kb mutation does not intrinsically affect T cell differentiation, which is consistent with the observed T cell phenotype being attributed to deficient RORγt-APC function.

## RORγt regulates PRDM16-expressing APCs

To better define the RORγt-dependent APCs required for intestinal $pT_{reg}$ cell differentiation, we performed single-cell RNA sequencing (scRNA-seq) of MHCII-expressing innate immune system cells from mLN of control and mutant mice, including both Δ+7 kb and *Rorc(t)*[ΔCD11c] models. Cells collected from Δ+7 kb mutants and littermate controls at 3 weeks of age yielded a total of 21,504 high-quality transcriptomes, comprising 11,150 and 10,354 cells from mutant and control animals, respectively (Fig. 2a). We analysed these data with an unsupervised computation (Methods). Populations of dendritic cells and ILCs and distinct clusters of B cell and macrophage lineages were readily annotated (Fig. 2b and Extended Data Fig. 4a) on the basis of established biomarkers (Methods).

On querying for *Rorc*-positive cells, we found two additional distinct clusters beyond the expected ILC3s (arrows, Fig. 2a and Extended Data Fig. 4b). We denoted one as Nrg1_Pos, given its exclusively high expression of neuregulin 1 (encoded by *Nrg1*), a trophic factor that is known to coordinate various neuronal functions including axon guidance, myelination and synapse formation (Extended Data Fig. 4c). Additional neuronal genes (*Ncam1*, *Nrxn1* and *Nrn1*) were also prominent, as was expression of *Aire*, *Trp63* and *Tnfrsf11b* (encoding osteoprotegerin), which is compatible with the Thetis cells I subset[6]. This cluster also carried a signature that is closely associated with the fibroblastic reticular cell (FRC) subset found in mucosal lymph nodes[29], including *Madcam1*, *Mfge8*, *Twist1*, *Vcam1* and *Nid1*. Both non-ILC3 *Rorc*+ clusters expressed CD45, as well as MHCII at levels similar to those in dendritic cell subsets. Therefore, Nrg1_Pos cells are not bona fide stromal cells or medullary thymic epithelial cells (mTECs), but they exhibit prominent neural, FRC and mTEC attributes.

The second *Rorc*+ cluster displayed exclusively high expression of *Prdm16* (Fig. 2b and Extended Data Fig. 4d), which was previously shown to be variably expressed across Thetis cells subsets[6]. This singular appearance of *Prdm16* in our dataset seemed notable, given that it is a zinc-finger protein and histone methyltransferase that was previously identified as a transcriptional regulator in multiple cell types, including brown adipocytes, cortical neurons and vascular endothelial cells[30]. The *Prdm16*+ cluster was notably higher in *Itgb8*, *Aire*, *Cd40* and *Ccr7* expression compared with Nrg1_Pos cells (Fig. 2b).

After annotation, we deconvolved each cluster according to its origin from Δ+7 kb mutant or control mice. Whereas the contribution from each condition varied minimally around the numeric split of total high-quality transcriptomes captured (Fig. 2c, dotted line), we observed a marked 60% reduction of *Prdm16*+ cluster cells from the expected Δ+7 kb contribution. Violin plots examining *Rorc* and *Ccr6* illustrate clear decreases of those gene expressions within the mutant-origin ILC3 population (Extended Data Fig. 4e), yet the overall cluster of ILC3s

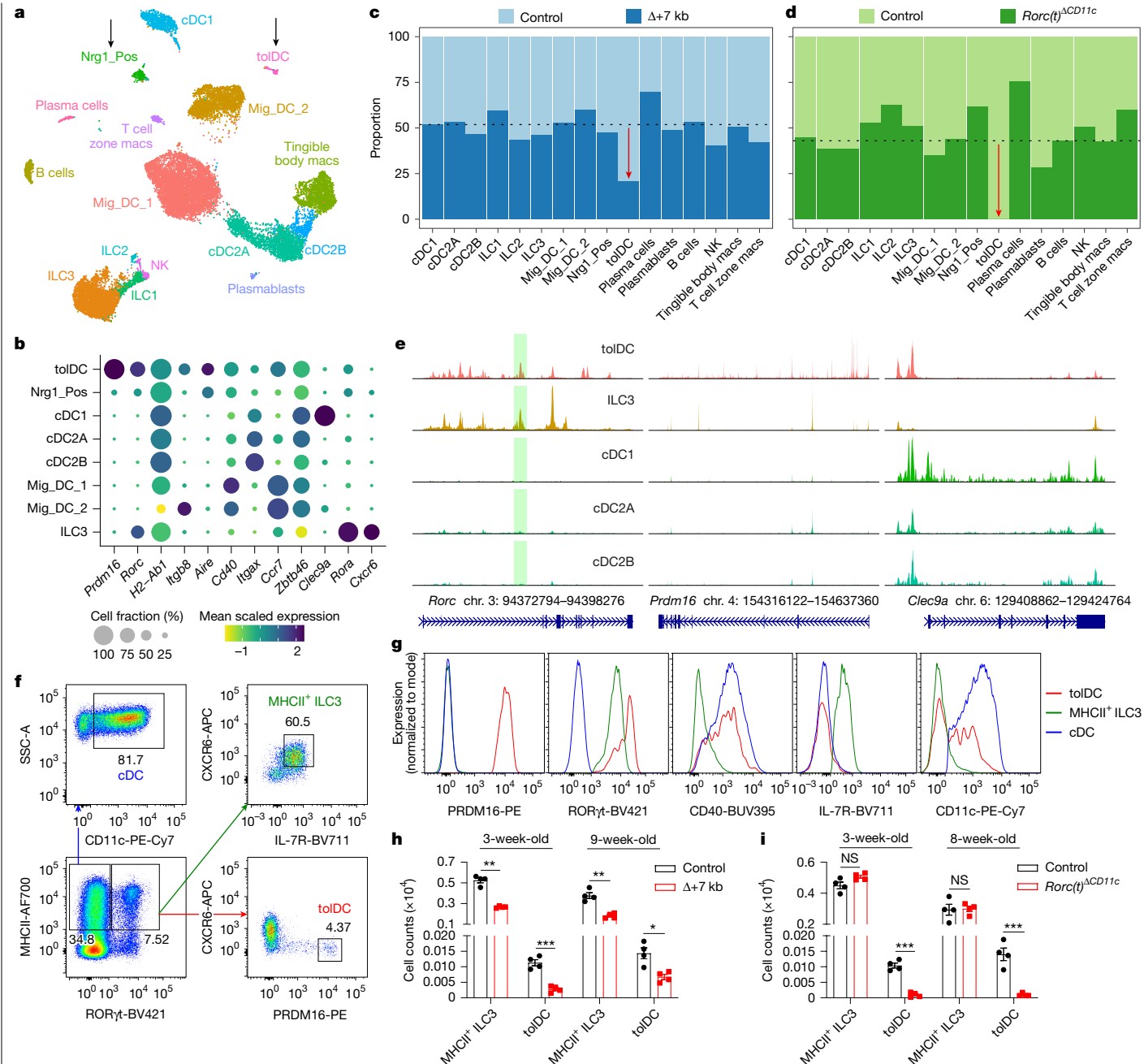

**Fig. 2 | RORγt regulates development of PRDM16-expressing tolDCs within mLNs. a**, Uniform manifold approximation and projection (UMAP) representation of 21,504 transcriptomes obtained from scRNA-seq of MHCII-expressing innate immune system cells (CD45⁺Ly6G⁻B220⁻TCRγδ⁻TCRβ⁻MHCII⁺) in the mLN, combining data from 3-week-old control and Δ+7 kb mice for joint clustering. Macs, macrophages; NK, natural killer cells. **b**, Dot plot of indicated clusters from **a** examining expression of genes that have been previously ascribed to proposed RORγt-APC subsets. **c**, Stacked bar plots comparing the proportion of each cluster in **a**, as derived from control and Δ+7 kb mice. The dashed line at 51.2% indicates total contribution from Δ+7 kb mutants. **d**, Stacked bar plots analogous to experiment **a**–**c**, but comparing proportion of cell clusters as derived from control and *Rorc(t)^ΔCD11c* mice. The dashed line at 43% indicates the

total contribution from *Rorc(t)^ΔCD11c* mutants. **e**, Chromatin accessibility profiles for *Rorc*, *Prdm16* and *Clec9a* loci across the indicated cell types. The green shaded region demarcates the *Rorc(t)* +7 kb CRE. **f**, Gating strategy for tolDC, MHCII⁺ ILC3 and cDC populations. The bottom left flow cytometry plot is gated on CD45⁺Ly6G⁻B220⁻TCRγδ⁻TCRβ⁻ and was generated by concatenating samples from four wild-type mice. **g**, Expression of the indicated proteins in tolDCs, MHCII⁺ ILC3s and cDCs, as gated in **f**. **h**, Numbers of tolDCs and MHCII⁺ ILC3s in the mLN of 3-week-old and 9-week-old control and Δ+7 kb mice. *n* = 4 per group. **i**, Numbers of tolDCs and MHCII⁺ ILC3s in the mLN of 3-week-old and 8-week-old control and *Rorc(t)^ΔCD11c* mice. *n* = 4 per group. Data in **f**–**i** are representative of two (**h,i**) or three (**f,g**) independent experiments. Data are mean ± s.e.m. Unpaired two-sided *t*-test.

from mutant mice remained otherwise numerically intact. As there was also no apparent difference in the Nrg1_Pos distribution when comparing control mice with Δ+7 kb mutant mice, the results suggested that the specialized ability to induce pT_reg is contained in the *Prdm16*⁺ population.

Comparison of mLN single-cell transcriptomes from *Rorc(t)^ΔCD11c* (8,474 cells) and littermate control mice (11,245 cells) yielded results congruent with those observed in the Δ+7 kb model. In these mutants, however, there was complete loss of the *Prdm16*⁺ cluster when we deconvolved biological conditions, despite a 43% overall contribution of total

transcriptomes from *Rorc(t)*^*ΔCD11c*^ mice (Fig. 2d, dotted line). Together, the results are most consistent with a requirement for RORγt expression during development of PRDM16⁺ APCs and suggest that these cells are likely candidates for tolerance-inducing APCs in response to the microbiota.

## Characterization of PRDM16⁺ APCs as tolDCs

To further define the PRDM16⁺ APCs, we utilized transcriptomic readout paired with single-cell ATAC–seq (multiome scRNA-seq/scATAC–seq), resulting in 14,917 high-quality nuclei from mLN of *Rorc(t)* fate-mapped mice. The results were computationally integrated[31] with data from all previous control and mutant mice, generating the exact same populations as above (Extended Data Fig. 4f). A previous investigation had used a similar multiome analysis, albeit on nuclei enriched via a *Rorc*-reporter mouse[6]. We additionally integrated that raw data alongside our own, which proved to be a critical step before shared nearest-neighbour clustering, as this provided the context of all other MHCII-expressing APCs found within mLN beyond *Rorc*-enriched cells, which were still observed in the three distinct clusters, ILC3, Nrg1_Pos and PRDM16⁺, suggesting that our strategy for computation and annotation was robust (Extended Data Fig. 4g).

Consistent with the gene expression data, scATAC–seq revealed chromatin accessibility at both *Rorc* and *Prdm16* loci in *Prdm16*⁺ APCs (Fig. 2e). Close inspection of the *Rorc(t)* + 7 kb CRE (green shaded region in Fig. 2e), demonstrated prominent peaks in both *Prdm16*⁺ APCs and ILC3, as expected. Comparing chromatin accessibility of *Prdm16*⁺ APCs to that of ILC3, cDC1, cDC2A and cDC2B populations revealed a consistent epigenetic pattern. For instance, although *Prdm16*⁺ APCs did not transcribe *Clec9a*, the locus was accessible, with conspicuous peaks noticeably shared across cDC populations but absent in ILC3s. This pattern of chromatin landscape shared by *Prdm16*⁺ APCs and cDCs, but not ILC3s, was observed at multiple loci, including *Csf1r*, *Flt3*, *Clecl0a*, *Sirpa*, *Itgax*, *Itgam* and *Itgae* (Extended Data Fig. 5). We observed the converse for *Cxcr6*, with chromatin accessibility and gene expression in ILC3, but not in *Prdm16*⁺ APCs and cDC.

Using a flow cytometry gating strategy for PRDM16⁺ APCs, MHCII⁺ ILC3s and cDCs, we confirmed that PRDM16⁺ APCs co-expressed PRDM16 and RORγt at high levels (Fig. 2f,g). PRDM16⁺ APCs also expressed CD40 at levels similar to cDCs, supporting their involvement in T cell priming. In contrast to MHCII⁺ ILC3s, PRDM16⁺ APCs and cDCs lacked expression of IL7R, CXCR6 and CD90 (Fig. 2g and Extended Data Fig. 6a). Notably, expression of CD11c and CD11b was limited to a portion of PRDM16⁺ APCs, suggesting that there is partial downregulation of CD11c during lineage progression. In terms of side scatter, PRDM16⁺ APCs resembled cDCs, whereas their forward scatter was intermediate between those of MHCII⁺ ILC3s and cDCs (Extended Data Fig. 6b). These results indicate that PRDM16⁺ APCs and MHCII⁺ ILC3s form distinct populations, with PRDM16⁺ APCs sharing more similarities with cDCs. In addition, our analysis of PRDM16⁺ APC abundance across a developmental time frame—in mice from 1 to 12 weeks of age—revealed a slight increase in their numbers with age, followed by stabilization in adulthood (Extended Data Fig. 6c). This pattern aligns with the observed ability of adult mice to induce tolerance to gut microbiota[5,32], suggesting that these APCs maintain functional persistence beyond early development.

Further validation by flow cytometry analysis revealed that whereas the numbers and RORγt MFI of PRDM16⁺ APCs and MHCII⁺ ILC3s were decreased in the mLN of Δ+7 kb mice, the number of MHCII⁺ ILC3s remained unchanged and PRDM16⁺ APCs were depleted in *Rorc(t)*^*ΔCD11c*^ mice (Fig. 2h,i and Extended Data Fig. 6d,e), indicating that PRDM16⁺ APCs but not ILC3s are potential pT_reg-inducing APCs. To further test this, we conditionally inactivated PRDM16 in RORγt-expressing cells (*Prdm16*^*ΔRORγt*^ mice) and assessed the fate of *Hh*-specific T cells. In these mice, numbers of MHCII⁺ ILC3s were unchanged in the mLN and

intestines, whereas PRDM16⁺ APCs were barely detectable when identified using PRDM16 as a marker (Extended Data Fig. 7a,b). In accordance, the differentiation of adoptively transferred *Hh*-specific pT_reg cells was abolished in the LILP and mLN of *Hh*-colonized *Prdm16*^*ΔRORγt*^ mice, accompanied by an increase in inflammatory T_H17 and T_H1 cells (Fig. 3a and Extended Data Fig. 7c,d), suggesting at least a loss of tolerance-inducing function in PRDM16⁺ APCs. Collectively, these findings establish PRDM16⁺ APCs as a distinct population that closely resembles cDCs, yet is uniquely capable of inducing microbiota-specific pT_reg cells, supporting their designation as PRDM16⁺RORγt⁺ tolDCs.

## tolDCs are required for oral tolerance

When Δ+7 kb, *Rorc(t)*^*ΔCD11c*^ and *Prdm16*^*ΔRORγt*^ mice were examined in the absence of *Hh* colonization, we observed the expected reduction, compared with control mice, of RORγt⁺ pT_reg cells in both SILP and LILP (Extended Data Fig. 7e,g,h). Of note, this decrease did not coincide with an increase in T_H17 cell proportions but was associated with a substantial increase in the number of intestinal T helper 2 (T_H2) cells (Extended Data Fig. 7e–h). Notably, in 40-week-old Δ+7 kb mice, no overt intestinal inflammation was observed by haematoxylin and eosin (H&E) staining. Nevertheless, these mice displayed features indicative of spontaneous type 2 gastrointestinal pathology[33,34], including muscularis propria hypertrophy, increased small intestine length and increased serum total IgE levels (Extended Data Fig. 7i–l). These observations in the mutant mice suggested that tolDCs induce pT_reg cells that are specific not only for microbiota, but also for food antigens with potential to induce allergy-related T_H2 cells. To test this idea, we transferred naive ovalbumin (OVA)-specific CD4⁺ T cells from OT-II TCR-transgenic mice into control mice and mutant mice that were deficient for microbiota-specific pT_reg induction, and administered OVA either via oral gavage or in the drinking water (Fig. 3b). Notably, OVA-specific OT-II pT_reg cells in control mice consisted of both RORγt⁻ and RORγt⁺ phenotypes (Fig. 3c–f). At five days post-transfer, there were few OT-II pT_reg cells in the mLN of *Prdm16*^*ΔRORγt*^, Δ+7 kb and *Rorc(t)*^*ΔCD11c*^ mice, and instead, the OT-II T cells displayed T_H2 and T follicular helper (T_FH) phenotypes, with no notable changes in T_H17 and T_H1 profiles (Fig. 3c and Extended Data Fig. 8a–c). By the 12th day post-transfer, the reduction in OT-II pT_reg cells persisted in the intestines of *Prdm16*^*ΔRORγt*^, Δ+7 kb and *MHCII*^*ΔRORγt*^ (*Rorc(t)*^*cre*^*I-AB*^*fl/fl*^) mice, coinciding with variable increases across all OT-II T helper cell subsets (Fig. 3d–f and Extended Data Fig. 8d–f). These results highlight the critical role of tolDCs in promoting food antigen-specific pT_reg cell differentiation. Dysfunction of these APCs correlates with intensified effector T helper cell responses, although the specific T helper cell subset favored depends on the local tissue environment.

Peripheral pT_reg cells are pivotal in the induction and maintenance of oral tolerance, a critical mechanism that suppresses diverse immune responses not only in the gastrointestinal tract but also systemically[35,36]. We therefore aimed to explore whether tolDCs are essential for directing food antigen-specific pT_reg cells to mediate oral tolerance. We first used an allergic lung response model, in which oral administration of antigen prior to sensitization results in pT_reg-mediated inhibition of the inflammatory process. Mice were not pre-treated or were administered OVA intragastrically before being primed with OVA in alum and subsequently exposed to intranasal OVA challenge (Fig. 4a). Wild-type mice that had previously been fed OVA exhibited significant resistance to allergic lung inflammation, demonstrated by lower lung inflammation score, diminished eosinophil numbers in the bronchoalveolar lavage fluid and lungs, reduced numbers of lung T_H2 cells and decreased levels of serum OVA-specific IgE and IgG1 compared with non-tolerized controls (Fig. 4b–d and Extended Data Fig. 9a–f). However, the same pre-feeding strategy did not induce oral tolerance in Δ+7 kb mice. These knockout mice showed similar increases in lung inflammation score, eosinophils, T_H2 cells and OVA-specific IgE and IgG1 production as

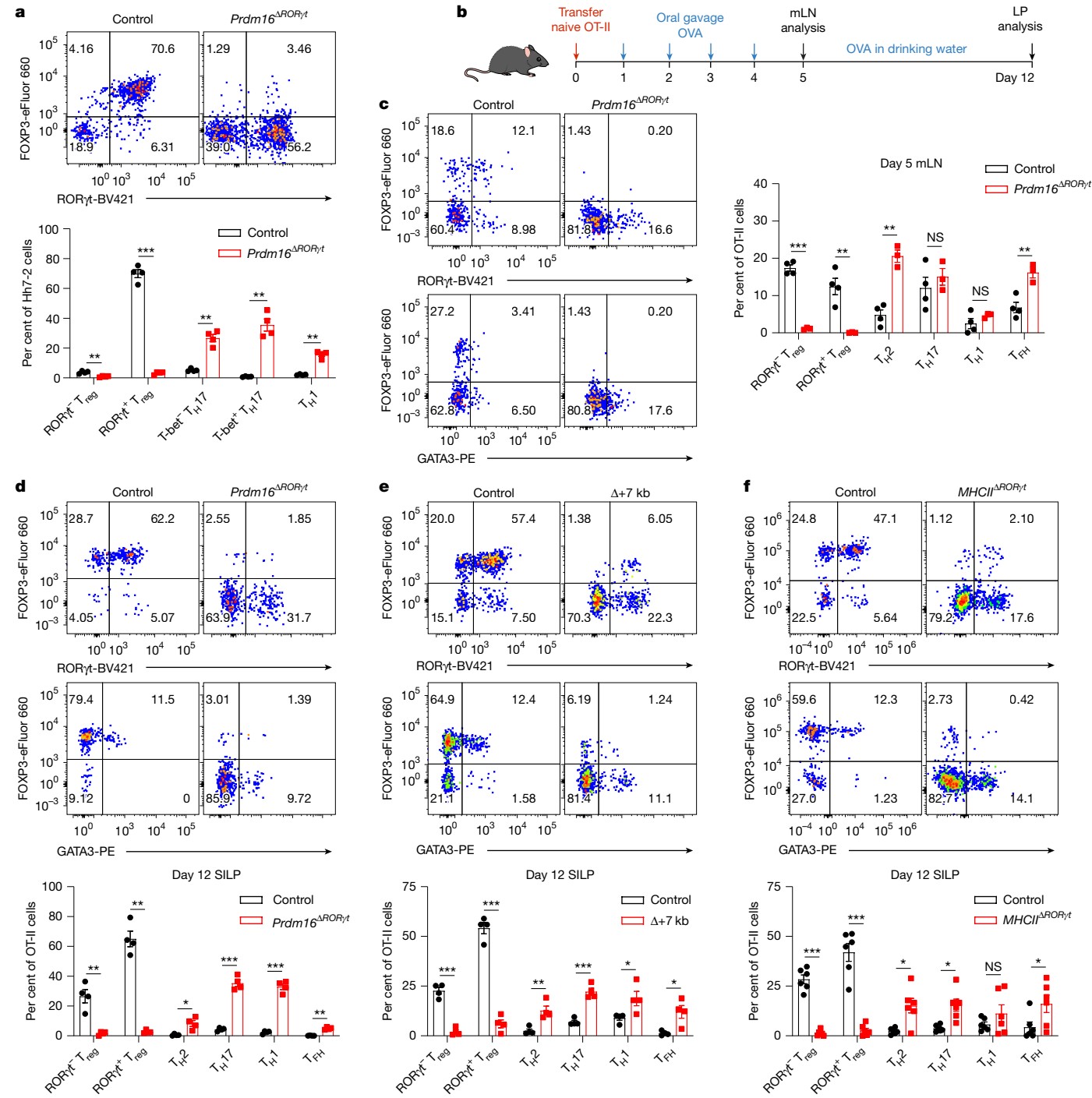

**Fig. 3 | PRDM16-dependent tolDCs promote microbiota-specific and food antigen-specific pT$_{reg}$ cell differentiation. a**, Phenotype of *Hh*-specific T cells in the LILP of *Hh*-colonized control (*Prdm16*^fl/fl^ and *Prdm16*^fl/wt^; n = 4) and *Prdm16*^ΔRORγt^ (*Rorc(t)*^cre^*Prdm16*^fl/fl^; n = 4) mice at 14 days after adoptive transfer of naive Hh7-2tg CD4⁺ T cells. **b**, Experimental design for the experiments in **c–f**. Created in BioRender; Fu, L. (2025) https://BioRender.com/o19q348. LP, lamina propria. **c–f**, Representative flow cytometry plots and frequencies of OT-II pT$_{reg}$ (FOXP3⁺RORγt^+/−^), T$_H$2 (FOXP3⁻GATA3⁺), T$_H$17 (FOXP3⁻GATA3⁻RORγt⁺), T$_H$1 (FOXP3⁻GATA3⁻RORγt⁻T-bet⁺) and T$_{FH}$ (FOXP3⁻GATA3⁻RORγt⁻T-bet⁻BCL6⁺) cells in the mLN of OVA-treated control and *Prdm16*^ΔRORγt^ mice (**c**) and SILP of

OVA-treated control and *Prdm16*^ΔRORγt^ (**d**), Δ+7 kb (**e**) or *MHCII*^ΔRORγt^ (*Rorc(t)*^cre^*I-AB*^fl/fl^) (**f**) mice at 5 and 12 days post-adoptive transfer of naive OT-II CD4⁺ T cells. The flow cytometry plots are gated on total OT-II cells (CD45⁺B220⁻TCRγδ⁻TCRβ⁺ CD4⁺Vα2⁺Vβ5.1/5.2⁺GFP⁺). **c**, mLN: control mice, n = 4; *Prdm16*^ΔRORγt^ mice, n = 3. **d**, SILP: control mice, n = 4; *Prdm16*^ΔRORγt^ mice, n = 4. **e**, SILP: control mice, n = 4; Δ+7 kb mice, n = 4. **f**, SILP: control (*I-AB*^fl/fl^) mice, n = 6; *MHCII*^ΔRORγt^ mice, n = 6. Data in **f** are pooled from two independent experiments; data in **a**,**c**–**e** are representative of two (**a**,**c**,**d**) or three (**e**) independent experiments. Data are mean ± s.e.m. Unpaired two-sided *t*-test.

non-tolerized Δ+7 kb mice (Fig. 4b–d and Extended Data Fig. 9a–f). We then focused on OVA-specific T cell responses in these mice, using OVA:I-A^b^ tetramers to identify those cells. In tolerized wild-type mice, tetramer-positive T cells were significantly fewer compared with

non-tolerized animals, and most cells were GATA3⁺FOXP3⁺ T$_{reg}$ cells, with limited RORγt expression (Fig. 4e,f and Extended Data Fig. 9g,h). By contrast, in both tolerized and non-tolerized Δ+7 kb mice, there was loss of OVA:I-A^b^ tetramer-binding pT$_{reg}$ cells in the lung, accompanied

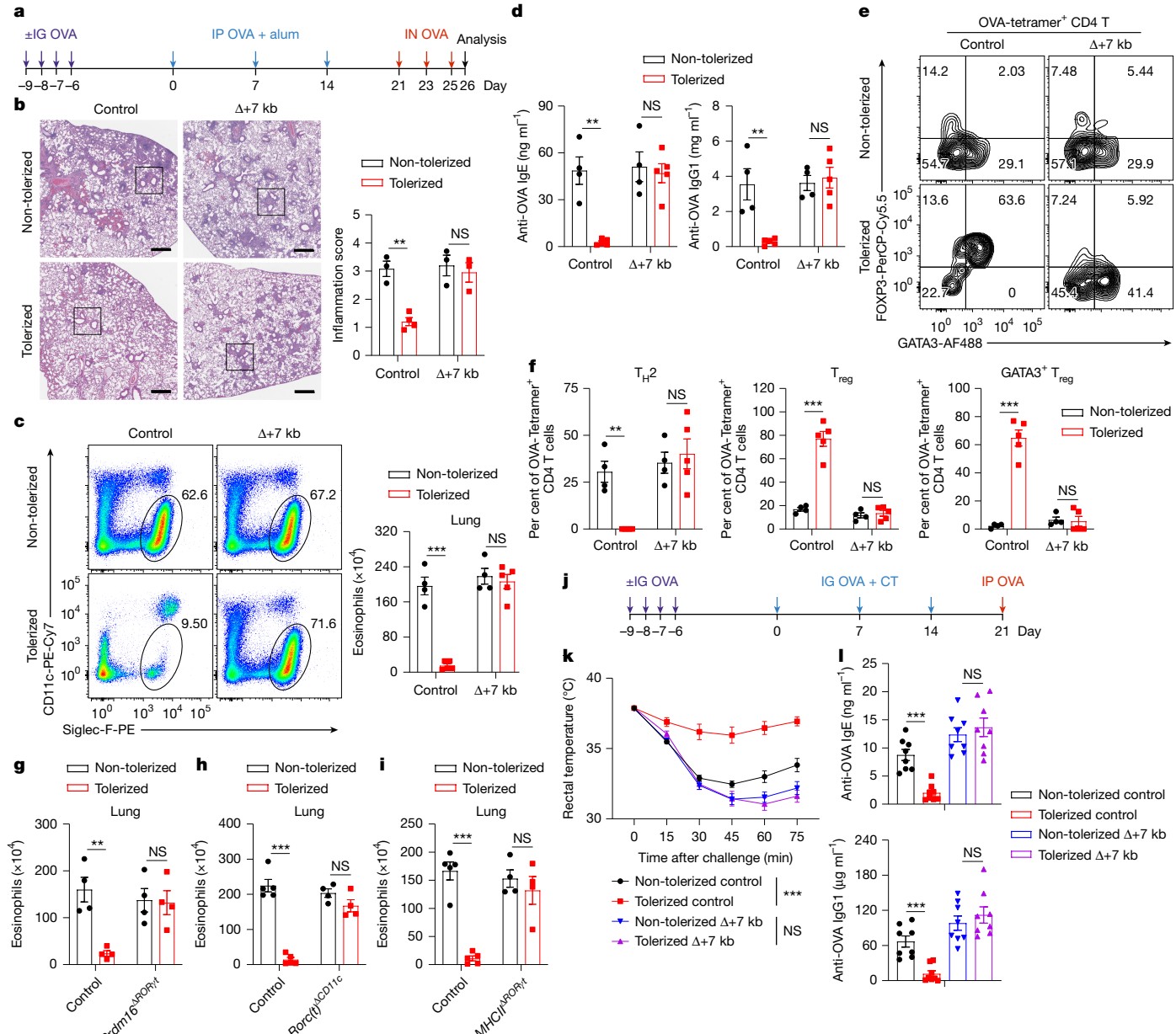

**Fig. 4 | tolDCs are required to develop oral tolerance. a**, Experimental design for the airway allergy experiments in **b**–**i**. IG, intragastric; IN, intranasal; IP, intraperitoneal. **b**, H&E staining and inflammation score of lung sections from control and Δ+7 kb mice. Scale bars, 500 μm. Non-tolerized control mice, $n = 3$; tolerized control mice, $n = 4$; non-tolerized Δ+7 kb mice, $n = 3$; tolerized Δ+7 kb mice, $n = 3$. **c**–**f**, Eosinophil (CD45$^+$CD11b$^+$CD11c$^{low/-}$Siglec-F$^+$) numbers in the lung (**c**), OVA-specific IgE and IgG1 levels in the serum (**d**) and phenotype (**e**) and quantification (**f**) of OVA:I-A$^b$ tetramer$^+$ CD4 T cells in the lung of control and Δ+7 kb mice. Flow cytometry plots in **e** were generated by concatenating the samples from each group. Non-tolerized mice, $n = 4$ per group; tolerized mice, $n = 5$ per group. **g**, Eosinophil numbers in the lung of control and

$Prdm16^{ΔRORγt}$ mice. $n = 4$ per group. **h**, Eosinophil numbers in the lung of control ($n = 5$ per group) and $Rorc(t)^{ΔCD11c}$ ($n = 4$ per group) mice. **i**, Eosinophil numbers in the lung of control ($n = 5$ per group) and $MHCII^{ΔRORγt}$ ($n = 4$ per group) mice. **j**, Experimental design for the food allergy experiments in **k**,**l**. CT, cholera toxin. **k**,**l**, Changes in rectal temperature (**k**) and OVA-specific IgE and IgG1 levels in the serum (**l**) of control and Δ+7 kb mice. $n = 8$ per group. Data in **k**,**l** are pooled from two independent experiments. Data in **b**–**f** are representative of two independent experiments. Data are mean ± s.e.m. Unpaired two-sided $t$-test (**b**–**d**,**f**–**i**,**l**) and two-stage step-up method of Benjamini, Krieger and Yekutieli (**k**). Schematics in **a**,**j** Created in BioRender; Fu, L. (2025) https://BioRender.com/o19q348.

by an increase in tetramer-positive $T_H2$, $T_H17$ and $T_H1$ cells, with a predominant increase in $T_H2$ cells (Fig. 4e,f and Extended Data Fig. 9g,h), consistent with loss of tolerance. Similarly, OVA-pre-fed $Prdm16^{ΔRORγt}$, $Rorc(t)^{ΔCD11c}$ and $MHCII^{ΔRORγt}$ mice exhibited similar increases in lung eosinophils and $T_H2$ cells to those observed in non-tolerized mutant mice (Fig. 4g–i and Extended Data Fig. 9i–k).

To investigate whether tolDC-induced $pT_{reg}$ cells are broadly required for tolerance to food antigens, we used a food allergy model in which mice were either pre-fed or not fed OVA prior to sensitization

intragastrically with OVA and cholera toxin, followed by intraperitoneal OVA challenge (Fig. 4j). Systemic OVA challenge resulted in similar anaphylactic responses—assessed as rapid reduction in core body temperature and increased levels of serum OVA-specific IgE and IgG1—in tolerized Δ+7 kb mice and non-tolerized Δ+7 kb mice, whereas tolerized wild-type mice were protected (Fig. 4k,l). These findings indicate that tolDCs are crucial for the development of oral tolerance, highlighting their essential role in regulating immune responses to dietary antigens and preventing allergic responses.

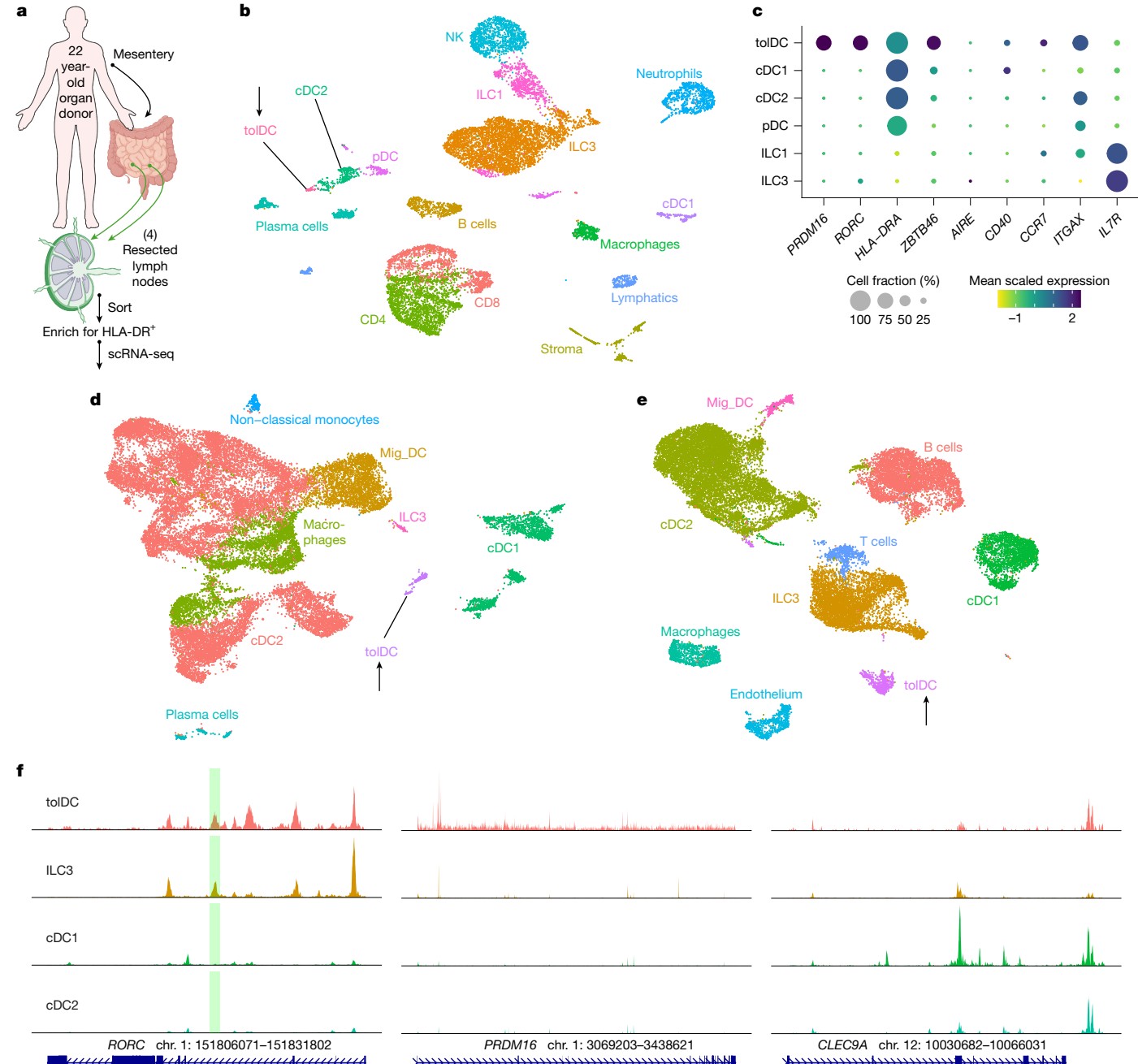

**Fig. 5 | PRDM16-expressing tolDCs are conserved in humans. a**, Input scheme for human scRNA-seq, with four mLNs resected and enriched for APCs. Created in BioRender; Fu, L. (2025) https://BioRender.com/o19q348. **b**, UMAP of 12,928 resultant human mLN transcriptomes. **c**, Dot plot of indicated clusters from **b** examining the expression of genes previously ascribed to RORγt-APC subsets. **d**, UMAP derived from public datasets of human lamina propria (ileum and colon) resected from six donors, enriched for APCs. **e**, UMAP derived from public datasets of human tonsil from nine donors, enriched for APCs. Black arrows in **b**,**d**,**e** highlight populations of tolDCs. **f**, Chromatin accessibility profiles for *RORC*, *PRDM16* and *CLEC9A* loci across the indicated cell types. Green shaded region demarcates the human orthologue of mouse *Rorc(t)* +7 kb CRE, part of conserved non-coding sequence 9 (CNS9).

## Orthologous tolDCs across human tissues

To determine whether the tolDCs identified in mice are also present in humans, we performed scRNA-seq on HLA-DR-enriched cells from freshly isolated mLNs of a 22-year-old trauma patient with no known history of atopy or chronic disease (Fig. 5a). Following the same unsupervised computation as for the mouse experiments, we clustered the resulting single-cell dataset (Fig. 5b). Established biomarkers (Methods) enabled annotation of multiple ILC and dendritic cell subtypes (Fig. 5c and Extended Data Fig. 10a).

On querying for *RORC* in this human dataset, we observed non-ILC3 expression in a single set of cells that clustered at the extremity of the cDC2 annotation (arrow, Fig. 5b,c). Notably, this same population exclusively displayed a high *PRDM16* signal. Unlike in the mouse scRNA-seq analysis, this population was not spontaneously demarcated by an unsupervised workflow, so we manually sub-clustered and similarly annotated it as tolDC. This human cluster was also positive for *ZBTB46*, *CD40*, *CCR7* and *ITGAX*, although *AIRE* was not detected in this or any other HLA-DR+ mLN population.

To further query for human tolDCs, we analysed public datasets derived from other tissues. We assembled raw sequencing data from

6 colonic lamina propria and 4 paired ileal lamina propria digests (all resected more than 10 cm from sites of colorectal cancer)[37], as well as 9 elective tonsillectomies[38,39]. Following our previous analysis, we annotated clusters with special attention towards distinct populations with high co-expression of *PRDM16* and *RORC*. This revealed a clear tolDC population in lamina propria (with equal contribution from ileum and colon, Fig. 5d), as well as tolDC in tonsil (Fig. 5e). We examined multiple human bone marrow datasets but could not identify any cells that co-expressed *PRDM16* and *RORC*. It therefore remains unclear when or where final tolDC specification takes place.

We also examined available scATAC–seq data, especially since mouse *Rorc(t)* +7 kb CRE has been demonstrated to have an orthologue contained within conserved non-coding sequence 9 of human *RORC*[22]. As in mouse, there was a prominent peak at this locus (Fig. 5f, green shaded region) for human ILC3 and tolDC. Similarly, chromatin at the *PRDM16* locus was accessible only in tolDCs, whereas chromatin at *CLEC9A* was accessible in tolDCs and classical dendritic cell populations.

Aggregating all available data totalled 62,952 mouse cells (including 1,271 tolDCs) and 67,638 human cells (including 988 tolDCs). By setting a relatively stringent definition of differentially expressed genes (DEGs) for each cell type within each dataset (Methods), we queried for DEGs shared across species. Conserved genes (Extended Data Fig. 10b and Supplementary Table 1) included mostly expected results for cDC1s (such as *CLEC9A*, *CADM1* and *XCR1*) and ILC3s (such as *IL23R*, *IL22* and *RORC*). This same analysis revealed eight candidate genes that were conserved as DEGs across human and mouse tolDCs (Extended Data Fig. 10b,c), notably including *PRDM16*, *PIGR*, *IDO2*, *DLGAP1*, *MACC1* and *CDH1*, largely corroborated by broader cross-species analyses of RORγt+ APCs[39,40].

Genome-wide association studies have identified many single nucleotide polymorphisms (SNPs) in *PRDM16* that have strong statistical association with autoimmune and inflammatory diseases, including asthma in African Americans (mostly paediatric)[41], allergic rhinitis (IgE-mediated inflammation of the upper airway)[42], rheumatoid arthritis[43] and inflammatory bowel disease in the Basque population[44].

## Discussion

In maintaining intestinal homeostasis, the gut immune system is tasked with tolerating a complex array of dietary and microbial antigens, largely through the action of pT$_{reg}$ cells. Three independent studies have demonstrated that RORγt-APCs, but not cDCs, promote the differentiation of gut microbiota-specific pT$_{reg}$ cells. However, the precise identity of the pT$_{reg}$-inducing RORγt-APCs remained unresolved[45]. Candidates include MHCII+ ILC3s and various non-ILC RORγt+ APCs—such as Janus cells, Thetis cells and RORγt+ dendritic cells[5–7,39,40]. Notably, the non-ILC populations are likely to overlap and may represent a heterogeneous mix of APC subsets. Here we performed an in-depth characterization of tolDCs and identified them, rather than ILC3s, as the tolerance-inducing APCs, using the transcription factor PRDM16 as both a phenotypic and genetic discriminator. The tolDCs represent a well-defined subset within the broader group of non-ILC RORγt+ APCs. Although MHCII expression in RORγt+ APCs, most probably the tolDCs, results in exclusive programming of pT$_{reg}$ cells[5], it remains possible that under some conditions the tolDCs exhibit functional plasticity and adopt effector functions. However, the suggestion that RORγt+ dendritic cells can induce effector T cells was based on in vitro assays with potentially heterogeneous cell populations[40], and thus this hypothesis warrants further investigation. Furthermore, our study shows that tolDCs direct both microbiota-specific and dietary antigen-specific pT$_{reg}$ cell differentiation, thus facilitating oral tolerance to food antigens. Oral tolerance can be enhanced by oral immunotherapy for food allergies[46,47] and has been demonstrated to be effective in animal models to control antigen-specific autoimmune diseases[48,49], raising the prospect that

tolDCs could be harnessed to modulate autoimmune diseases and transplantation tolerance.

Our findings indicate that the competency of tolDCs in establishing mucosal tolerance crucially relies on PRDM16 and RORγt, as well as a specific CRE within the *Rorc* locus. The transcriptional regulatory network in which PRDM16 and RORγt participate to influence development and function of these APCs remains to be identified. A comprehensive understanding of the key components will probably reveal human genetic variants that can predispose to inflammatory and allergic conditions. An understanding of the ontogeny of tolDCs may also provide critical insights into inflammatory diseases and potential therapeutic avenues. Crucially, these tolerance-inducing APCs possess a remarkable ability to induce pT$_{reg}$ cells despite their low abundance in the intestinal secondary lymphoid organs. It is notable that multiple genes whose expression is conserved across human and mouse tolDCs are known for specialized cell adhesion and migration functions (*CDH1*, *MACC1*, *KRT8* and *DLGAP1*). Unravelling the mechanisms behind the potent immunomodulatory effect of tolDCs is essential and will probably require determination of the spatial distribution of the APCs and temporal dynamics of differentiation of T cell subsets that are specific for antigens encountered in the alimentary tract. Notably, the abundance of these APCs is maintained from birth to adulthood, consistent with previous[5,12,32] and current findings that adult mice retain the capacity to induce tolerance to microbiota and oral antigens. tolDCs may therefore serve as a valuable therapeutic target to enhance T$_{reg}$ cell functions in multiple immune-related diseases.

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

## Methods

### Mice

C57BL/6 mice (JAX 000664), CD45.1 mice (B6.SJL-*Ptprc^a^ Pepc^b*/BoyJ, JAX 002014), CD90.1 mice (B6.PL-*Thy1^a*/CyJ, JAX 000406), *Cd11c^cre^* mice (B6.Cg-Tg(*Itgax-cre*)1-1Reiz/J, JAX 008068), *I-AB^fl/fl^* mice (B6.129×1-*H2-Ab1^b-tm1Koni*/J, JAX 013181) and *ROSA26^lsl-tdTomato/lsl-tdTomato^* mice (B6.Cg-*Gt(ROSA)26Sor^tm14(CAG-tdTomato)Hze*/J, JAX 007914) were purchased from the Jackson Laboratories. *Rorc^fl/fl^*, *Rorc(t)^gfp/gfp^*, *Rorc(t)^cre^* and Hh7-2tg mice were generated in our laboratory and have been described[16,32,50,51]. *Il23r^gfp^* mice[52] were provided by M. Oukka. OT-II;UBC-GFP;*Rag1^−/−^* mice[53–55] were provided by S. R. Schwab. *Prdm16^fl/fl^* mice[56] were provided by B. Spiegelman. Tg (Δ+3 kb *Rorc(t)*-mCherry), *Rorc(t)* +6 kb^−/−^ and *Rorc(t)* +7 kb^−/−^ mice were generated as described in 'Generation of BAC transgenic reporter and CRISPR knockout mice'. Mice were bred and maintained in the Alexandria Center for the Life Sciences animal facility of the New York University School of Medicine, in specific pathogen-free conditions. Sex- and age-matched mice used in all experiments were 6–12 weeks old at the start of treatment if not otherwise indicated. Sample sizes were not predetermined but are listed with each result, and randomization occurred across littermates. All animal procedures were performed in accordance with protocols approved by the Institutional Animal Care and Usage Committee of New York University School of Medicine.

### Generation of BAC transgenic reporter and CRISPR knockout mice

Tg (Δ+3 kb *Rorc(t)*-mCherry) mice were generated as previously described[23]. The following primers were used for generating amplicons for GalK recombineering, and screening for correct insertion and later removal of the GalK cassette: Galk Rec +3 kb F: CTGCCTCCCACGTGCTAGGATTGTAATATAGAGCATCAGGCCCTGCTCC ACCTGTTGACAATTAATCATCGGCA; Galk Rec +3 kb R: ACAGACA GATACCATTCCTTGGGCCTGGCTTCCCTCAGTGGTCCTGGCTGTCAGCA CTGTCCTGCTCCTT; +3 kb HA F: CTGCCTCCCACGTGCTAGGATTG TAATATAGAGCATCAGGCCCTGCTCCA; +3 kb HA R: ACAGACAGAT ACCATTCCTTGGGCCTGGCTTCCCTCAGTGGTCCTGGCTG; +3 kb screen F: CTGCCTCCCACGTGCTAGGAT; +3 kb screen R: ACAGAC AGATACCATTCCTTGGG. The following primers were used for the recombineering that led to scarless deletion of *cis* element: deletion template +3 kb HA F: CTGCCTCCCACGTGCTAGGATTGTAATATAGAGC ATCAGGCCCTGCTCCACAGCCAGGACCACTGAGGGAAGCCAGGCCCA AGGAATGGTATCTGTCTGT; deletion template +3 kb HA R: ACAGACAG ATACCATTCCTTGGGCCTGGCTTCCCTCAGTGGTCCTGGCTGTGGAGCA GGGCCTGATGCTCTATATTACAATCCTAGCACGTGGGAGGCAG.

Single guide RNAs (sgRNAs) were designed flanking conserved regions in the +6 kb and +7 kb region to be deleted using pairs of guides. sgRNAs were cloned into pX458 for use as a PCR template to generate a template for in vitro transcription. In vitro transcribed sgRNAs and Cas9 mRNA were microinjected into zygotes to generate founder animals that were screened by PCR for deletions. For mice containing expected or interesting deletions, PCR products were TA cloned and Sanger sequenced to determine location of deletion. Founder mice with clean deletions were chosen to breed to C57BL/6 mice to generate lines. The following primers were used to clone sgRNA sequences into a BbsI site (lowercase letters represent homology to BbsI): +6/7 kb sgRNA-1 F: caccg TTTTCTTTGTGATACCCTTC; +6/7 kb sgRNA-1 R: aaacGAAGGGTATC ACAAAGAAAAc; +6/7 kb sgRNA-2 F: caccgGGAGAGACAACTGAAATCGT; +6/7 kb sgRNA-2 R: aaacACGATTTCAGTTGTCTCTCC; +6/7 kb sgRNA-3 F: caccgTGGACCCAAGTGTTACTGCC; +6/7 kb sgRNA-3 R: aaacGGCAGT AACACTTGGGTCCAc.

### Mouse antibodies, intracellular staining and flow cytometry

The following monoclonal antibodies were purchased from Thermo Fisher, BD Biosciences, BioLegend or Abcam: CD3ε (145-2C11, 1:250), CD4 (RM4-5, 1:250), CD11b (M1/70, 1:250), CD11c (N418, 1:250), CD25 (PC61.5, 1:250), CD40 (3/23, 1:250), CD44 (IM7, 1:250), CD45 (30-F11, 1:100), CD45.1 (A20, 1:100), CD45.2 (104, 1:100), CD62L (MEL-14, 1:250), CD90.1 (HIS51, 1:250), CD90.2 (53-2.1, 1:250), IL7R (SB/199, 1:250), CXCR6 (SA051D1 and DANID2, 1:250), CCR6 (140706, 1:250), NCR1 (29A1.4, 1:250), MHCII I-A/I-E (M5/114.15.2, 1:100), Ly6G (1A8, 1:250), Siglec-F (E50-2440, 1:250), B220 (RA3-6B2, 1:250), TCR Vα2 (B20.1, 1:250), TCRβ (H57-597, 1:250), TCR Vβ5.1/5.2 (MR9-4, 1:250), TCR Vβ6 (RR4-7, 1:250), TCRγδ (GL3, 1:250), IL-22 (IL22JOP, 1:250), FOXP3 (FJK-16s, 1:250), RORγt (B2D and Q31-378, 1:250), GATA3 (TWAJ, 1:100), T-bet (O4-46, 1:100), BCL6 (K112-91, 1:100) and PRDM16 (EPR24315-59, 1:250). PE-conjugated F(ab')2-donkey anti-rabbit IgG (H + L) (Thermo Fisher, 12-4739-81, 1:250) was used as the secondary antibody to detect PRDM16. Anti-mouse CD16/32 (Bio X Cell clone 2.4G2, 1:500) was used to block Fc receptors. Live/dead fixable blue (Thermo Fisher, L34962, 1:1,000) was used to exclude dead cells. I-A^b^ OVA$_{328-337}$ tetramers (HAAHAEINEA) were provided by the NIH Tetramer Core Facility.

For labelling OVA-specific T cells, cells were incubated with tetramers (1:100) for 60 min at 37 °C prior to surface staining. For intracellular staining, cells were stained for surface markers, followed by fixation and permeabilization before intracellular staining according to the manufacturer's protocol (FOXP3 staining buffer set from Thermo Fisher). For cytokine analysis, cells were stimulated ex vivo for 3 h with IL-23 (10 ng ml^−1^; R&D systems) and GolgiStop (BD Biosciences) in complete RPMI-1640 culture medium (RPMI-1640 with 10% FBS, 1% GlutaMAX, 1% penicillin−streptomycin, 10 mM HEPES, and 1 mM sodium pyruvate). Flow cytometry was performed on LSR II and Aria using FACSDiva v8.0.1 (BD Biosciences) or a Cytek Aurora using Spec-troFLo v3.03 (Cytek Biosciences) and analysed using FlowJo software (Tree Star).

### Isolation of lymphocytes

For isolation of cells from lymph nodes and spleens, tissues were mechanically disrupted with the plunger of a 1-ml syringe and passed through 70-µm cell strainers. Bone marrow cells were isolated by flushing the marrow from femur bones using a syringe with RPMI-1640 wash medium (RPMI-1640 with 3% FBS, 10 mM HEPES, 1% Gluta-MAX, 1 mM sodium pyruvate, and 1% penicillin−streptomycin). Red blood cells were lysed with ACK lysing buffer (Thermo Fisher). Bronchoalveolar lavage fluid cells were collected by flushing the lungs twice with 0.75 ml PBS through a catheter inserted into the trachea.

Lung tissues were minced into small fragments and digested at 37 °C for 45 min with shaking in RPMI-1640 wash medium containing 0.5 mg ml^−1^ collagenase D (Sigma) and 0.25 mg ml^−1^ DNase I (Sigma). After removal of Peyer's patches and caecal patches, the intestines were opened longitudinally, cut into 0.5-cm pieces, and washed twice with PBS. Intestine tissues were then shaken in HBSS wash medium (without Ca^2+^ and Mg^2+^, containing 3% FBS and 10 mM HEPES) with 1 mM DTT and 5 mM EDTA at 37 °C for 20 min, repeated twice. After washing with HBSS wash medium (without DTT and EDTA), the tissues were digested in RPMI-1640 wash medium containing 1 mg ml^−1^ collagenase D (Sigma), 0.25 mg ml^−1^ DNase I (Sigma), and 0.1 U ml^−1^ Dispase (Worthington) at 37 °C with shaking for 35 min (small intestines) or 55 min (large intestines). Leukocytes were collected at the interface of a 40%/80% Percoll gradient (GE Healthcare).

### C. rodentium-mediated colon inflammation

*C. rodentium* strain DBS100 (ATCC51459; American Type Culture Collection) was grown at 37 °C in LB broth. Mice were inoculated with 0.2 ml of a bacterial suspension ($2 \times 10^9$ CFU) by oral gavage. Mice were followed for the next 14 days to measure body weight change. Faecal pellets were collected and used to measure *C. rodentium* burden with serial dilutions on MacConkey agar plates.

### *Hh* culture and oral infection

*Hh* was provided by J. Fox. *Hh* was cultured and administered as previously described[32]. Frozen stock aliquots of *Hh* were stored in Brucella broth with 20% glycerol and frozen at −80 °C. The bacteria were grown on blood agar plates (TSA with 5% sheep blood, Thermo Fisher). Inoculated plates were placed into a hypoxia chamber (Billups-Rothenberg), and anaerobic gas mixture consisting of 80% nitrogen, 10% hydrogen and 10% carbon dioxide (Airgas) was added to create a micro-aerobic atmosphere, in which the oxygen concentration was 3–5%. The micro-aerobic jars containing bacterial plates were left at 37 °C for 4 days before animal inoculation. For oral infection, *Hh* was resuspended in Brucella broth by application of a pre-moistened sterile cotton swab applicator tip to the colony surface. 0.2 ml bacterial suspension was administered to each mouse by oral gavage. Mice were inoculated for a second dose after 4 days.

### Adoptive transfer of TCR-transgenic cells

Adoptive transfer of Hh7-2tg CD4⁺ T cells was performed as previously described[32], with minor modifications. Recipient mice were colonized with *Hh* by oral gavage 8 days before adoptive transfer. Lymph nodes from CD90.1;Hh7-2 TCR-transgenic mice were collected and mechanically disassociated. Naive Hh7-2tg CD4⁺ T cells were sorted as CD4⁺TCRβ⁺CD44$^{low/-}$CD62L⁺CD25⁻Vβ6⁺CD90.1⁺ (Hh7-2tg), on an Aria II (BD Biosciences). Cells were resuspended in PBS on ice and 100,000 cells were then transferred into congenic isotype-labelled recipient mice by retro-orbital injection. Cells were analysed 14 days after transfer.

Adoptive transfer of OT-II CD4⁺ T cells was performed as previously described[57], with minor modifications. Lymph nodes from OT-II;UBC-GFP;*Rag1*$^{-/-}$ mice were collected and mechanically dissociated. Naive OT-II CD4⁺ T cells were sorted as CD4⁺TCRβ⁺CD44$^{low/-}$CD62L⁺CD25⁻Vα2⁺Vβ5.1/5.2⁺GFP⁺ (OT-II), on the Aria II (BD Biosciences). Cells were resuspended in PBS on ice and 100,000 cells were then transferred into recipient mice by retro-orbital injection. Recipient mice received OVA by oral gavage (50 mg; A5378; Sigma) for 4 consecutive days, followed by drinking water containing OVA (2.5 mg ml⁻¹) for an additional 7 days after transfer. Cells from mLNs were analysed 5 days after transfer and cells from intestines were analysed 12 days after transfer.

### Generation of bone marrow chimeric reconstituted mice

To generate chimeric mice, 4- to 5-week-old wild-type CD45.1 mice were irradiated twice with 500 rad per mouse at an interval of 2–5 h (X-RAD 320 X-Ray Irradiator). One day later, mice were reconstituted with bone marrow cells (3 to 4 × 10⁶) obtained from wild-type CD45.1/2 mice mixed with either CD45.2 Δ+7 kb mutant or littermate control bone marrow cells to achieve a ~50:50 chimera. Mice were kept for a week on broad spectrum antibiotics (0.8 mg ml⁻¹ sulfamethoxazole and 0.16 mg ml⁻¹ trimethoprim), followed by microbiome reconstitution. After at least 8 weeks, tissues were collected for analysis.

### Induction of allergic airway inflammation

Mice were tolerized by oral gavage with 50 mg of OVA for 4 consecutive days. Six days after the last gavage, the mice were sensitized 3 times, 1 week apart, with intraperitoneal injections of 100 μg of OVA mixed with 1 mg of Alum (1:1; Alhydrogel adjuvant 2%; vac-alu-50; InvivoGen). Seven days after the final sensitization, the mice were challenged intranasally with 40 μg of OVA every other day, for a total of three challenges. Mice were analysed 24 h after the last intranasal administration.

### OVA and cholera toxin allergy mouse model

To induce tolerance, mice were given 50 mg of OVA by oral gavage once daily for four days. Six days after the final gavage, the mice were sensitized by oral administration of 5 mg of OVA combined with 10 μg of cholera toxin (C8052; Sigma), administered three times at weekly intervals. Seven days after the last sensitization, an intraperitoneal

challenge of 200 μg of OVA was administered. Rectal temperature was measured every 15 min for a total of 75 min after the challenge, using a Type J/K/T thermocouple thermometer (Kent Scientific). Serum was collected the following day to measure anti-OVA IgE and IgG1 levels.

### Histology analysis

Lung tissues were perfused and fixed with 10% buffered formalin phosphate, embedded in paraffin, and sectioned. Lung sections were stained with H&E and scored for histopathology in a blinded fashion as previously described[58], with minor modifications. In brief, lung inflammation was assessed by scoring cellular infiltration around the airways and vessels: 0, no infiltrates; 1, a few inflammatory cells; 2, a ring of inflammatory cells 1-cell-layer deep; 3, a ring of inflammatory cells 2–3 cells deep; 4, a ring of inflammatory cells 4–5 cells deep; and 5, a ring of inflammatory cells greater than 5 cells deep. The inflammation score for each mouse was calculated as the average of the scores from five lung lobes.

Intestine tissues were fixed with 4% paraformaldehyde, embedded in paraffin, and sectioned. Intestine sections were stained with H&E and scored for histopathology in a blinded fashion as previously described[59].

### Enzyme-linked immunosorbent assay

Serum levels of total IgE (432404; BioLegend), anti-OVA IgE (439807; BioLegend) and anti-OVA IgG1 (500830; Cayman Chemical) were measured following the manufacturer's recommendations.

### Bulk ATAC–seq

Samples were prepared as previously described[23,60]. In brief, 20–50,000 sort-purified cells were pelleted in a fixed rotor centrifuge at 500*g* for 5 min, washed once with 50 μl of cold PBS buffer, then spun down again at 500*g* for 5 min. Cells were gently pipetted to resuspend the cell pellet in 50 μl of cold lysis buffer (10 mM Tris-HCl, pH7.4, 10 mM NaCl, 3 mM MgCl₂, 0.1% IGEPAL CA-630) for 10 min. Cells were then spun down immediately at 500*g* for 10 min and 4 °C after which the supernatant was discarded and proceeded immediately to the Tn5 transposition reaction. Gently pipette to resuspend nuclei in the transposition reaction mix. Incubate the transposition reaction at 37 °C for 30 min. Immediately following transposition, purify using a Qiagen MinElute Kit. Elute transposed DNA in 10 μl Elution Buffer (10 mM Tris buffer, pH 8.0). The transposed nuclei were then amplified using NEBNext High-fidelity 2× PCR master mix for 5 cycles. In order to reduce GC and size bias in PCR, the PCR reaction was monitored using quantitative PCR (qPCR) to stop amplification prior to saturation using a qPCR side reaction. The additional number of cycles needed for the remaining 45 μl PCR reaction was determined as following: (1) plot linear Rn versus cycle; (2) set a 5,000 RF threshold; (3) calculate the no. of cycles that correspond to 1/4 of maximum fluorescent intensity. The amplified library was purified using Qiagen PCR Cleanup Kit and eluted in 20 μl Elution Buffer (10 mM Tris Buffer, pH 8). The purified libraries were then run on a high sensitivity Tapestation to determine whether proper tagmentation was achieved (band pattern, not too much large untagmented DNA or small overtagmented DNA at the top or bottom of gel). Paired-end 50 bp sequences were generated from samples on an Illumina HiSeq2500. Sequences were mapped to the mouse genome (mm10) with bowtie2 (2.2.3), filtered based on mapping score (MAPQ > 30, Samtools (0.1.19)), and duplicates were removed (Picard).

### Mouse mLN collection, cytometric sorting and scRNA-seq

Live, CD45⁺Lin⁻MHCII⁺ cells were sorted (Fig. 2a–d) from the mLN of 3-week-old Δ+7 kb mutants (*n* = 7) and their littermate controls (*n* = 7), as well as from 17- to 22-day-old *Rorc(t)*$^{ΔCD11c}$ mutants (*n* = 4) and their littermate controls (*n* = 4). Lineage markers comprised TCRβ, TCRγδ, B220 and Ly6G. All libraries were prepared using Chromium Next GEM

Single Cell 3-prime Kit v3.1 (10x Genomics), following the vendor's protocol and sequenced on an Illumina NovaSeq X+ sequencer.

## Mouse multiome scRNA-seq/scATAC−seq

Live, CD45⁺Lin⁻MHCII⁺tdTomato⁺ cells, along with one-eighth of the CD45⁺Lin⁻MHCII⁺tdTomato⁻CD11c⁺ cells were sorted (Fig. 2e) from the mLN of 3-week-old *Rorc(t)*$^{cre}$*ROSA26*$^{wt/lsl-tdTomato}$ mice (*n* = 4) and combined. Nuclear isolation was performed using a demonstrated protocol for single-cell multiome ATAC and gene expression sequencing (CG000365; 10x Genomics). All libraries were prepared using Chromium Next GEM Single Cell Multiome ATAC + Gene Expression Kit (10x Genomics), following the vendor's protocol, and sequenced on an Illumina NovaSeq X+ sequencer.

## Human mLN collection, cytometric sorting and scRNA-seq

We coordinated with a transplant surgery operating room to be notified immediately once a deceased (brain dead) donor was identified for life-saving clinical organ collection. We procured four mLNs from a 22-year-old patient maintained on life support, with minimal time from surgical resection to digestion of tissue at the benchside, resulting in >92% viable cells. The donor was free of chronic diseases and cancer, and negative for hepatitis B, hepatitis C, and HIV. This study does not qualify as human subjects research, as confirmed by NYU Langone Institutional Review Board, because tissues were obtained from a de-identified deceased individual.

The nodes were transported in Belzer UW Cold Storage Solution, and resected for excess adipose. Digestion buffer comprised RPMI media with 0.2 mg ml⁻¹ Liberase Thermolysin Medium and 2.5 mg ml⁻¹ collagenase D. Each node was pierced with a 31G syringe, injected with digestion buffer until plump, and submerged in the same for 30 min at 37 °C under agitation. Placing onto 100-µm filters, nodes were smashed open with syringe plungers, washing with excess RPMI to maximally recover cells. The suspension was stained with Live/dead eFluor780 (Thermo Fisher), CD19 and CD3 (Biolegend HIB19 and UCHT1) and HLA-DR (BD Biosciences L243). Cells were initially gated to enrich viable non-lymphocyte singlets (Fig. 5a). Not knowing MHCII expression levels within human nodes a priori, we chose to sort a wide range of HLA-DR⁺ stained cells, omitting only the very bottom quartile. Sorting 61,232 cells ultimately allowed loading of 26,000 across 2 lanes of a 10X microfluidics chip, followed by the same library preparation and sequencing as for mouse experiments.

## Computational analysis

RNA-sequencing data were aligned to reference genomes mm10-2020-A (mouse) or GRCh38-2020-A (human) and counted by Cell Ranger (v7.1.0) with default quality control parameters. Each dataset was filtered, removing cells with fewer than 500 detected genes, those with an aberrantly high number of genes (more than 10,000), and those with a high percentage of mitochondrial genes (more than 5%). We computed cell cycle scores for known S phase and G2/M phase marker genes, regressing out these variables, along with mitochondrial and ribosomal protein genes.

We utilized a completely unsupervised computational workflow that analysed all cells in aggregate, utilizing the most recent methods[61] within Seurat version 5.1 (including the sctransform function) for normalization of gene expression, anchor-based integration (based on 3,000 features), and shared nearest-neighbour cluster identification. We performed dimensional reduction using a principal component analysis that retained 50 dimensions, and clustering with the Leiden algorithm set to resolution = 1.0. This matched together cells with shared biological states across mutant and control animals, ensuring the final clusters were not driven by effects within either condition. The *Rorc(t)*$^{ΔCD11c}$ mouse scRNA-seq dataset was appended to the Δ+7 kb dataset, also clustered in an unsupervised manner, and then analysed for the same populations as before. All available raw public datasets from Gene Expression Omnibus and Chan Zuckerberg CELLxGENE repositories were filtered with standard quality controls only (no cell types were removed a priori), and similarly integrated alongside previous data. The human scRNA-seq dataset also underwent a predominantly unsupervised pipeline, but the tolDC population was subsequently manually sub-clustered from a juxtaposed cDC2 population.

For mouse cell-type annotation, juxtaposed ILCs were partitioned based on canonical expression of *Ncr1* and *Tbx21* (type 1), *Il1rl1* and *Il4* (type 2) or *Cxcr6*, *Ccr6* and *Nrp1* (type 3), versus a contiguous population of natural killer cells positive for *Eomes* and *Gzma*. cDC1 occupied a distinct cluster positive for *Xcr1* and *Clec9a*. While subsets of cDC2 (all positive for *Sirpa*) were contiguous, they could be separated based on expression of *Clec4a4*, *Cd4* and *Dtx1* (cDC2A) versus *Cd209a*, *Cx3cr1* and *Mgl2* (cDC2B), with all *Tbx21* expression contained within the former, as previously reported[62]. We observed two sizable populations of *Ccr7*⁺ migratory dendritic cells that were denoted Mig_DC_1 and Mig_DC_2, both consistent with a signature known to include *Socs2*, *Fas* and *Cxcl16*[5,63].

For human cell-type annotation, a population of ILC3s was demarcated by expression of *IL7R*, *KIT* and *NRP1*. This was juxtaposed by an ILC1 cluster expressing *NCR1* and *TBX21*, as well as NK cells expressing *EOMES*, all consistent with prior literature[64]. A cluster positive for *XCR1* and *CLEC9A* was readily distinguished as cDC1. Compared with mouse cDC2, there is less known about human cDC2s that allows for confident subset assignment. Nevertheless, we could identify a *SIRPA*⁺ population overlapping with *CD1C* expression, as would be expected for human cDC2s[65]. We annotated a plasmacytoid dendritic cell cluster adjacent to this, as these cells were negative for *CD1C* while positive for *IL3RA* (CD123) and included rare cells positive for *TLR7* and *TLR9* expression[66].

All scATAC−seq analysis was performed with Signac version 1.14, following a standard workflow[67]. In brief, mouse data were annotated with EnsDb.Mmusculus.v79 and converted to UCSC mm10 annotation where appropriate, and human data were annotated with EnsDb.Hsapiens.v98 and converted to UCSC hg38 annotation. Density scatter plots established rational thresholds for quality control, filtering data based on transcription start site scores, nucleosome banding patterns, features per cell, and mitochondrial genes. Normalization was performed with term frequency-inverse document frequency (TF-IDF). Dimensional reduction was performed with singular value decomposition on the TF-IDF matrix.

DEGs were identified with the Seurat FindMarkers algorithm using the Wilcoxon rank-sum test, computing the fold change gene expression within each indicated cell type (tolDC, cDC1 and ILC3) as compared to all other cell types, and this was performed separately within mouse and human datasets. Testing was limited to genes demonstrating $\log_2$(fold change) = 3.2 upregulated genes. The DEGs that defined each cell type according to this threshold were compared to the other species, as well as compared across indicated cell types, to generate the Venn diagrams in Extended Data Fig. 10b. All 141 DEGs for human tolDC are illustrated in the volcano plot in Extended Data Fig. 10c, with the 8 DEGs that are shared by mouse tolDCs specifically annotated.

## Statistical analysis

Unpaired two-sided *t*-test, paired two-sided *t*-test and the two-stage step-up method of Benjamini, Krieger and Yekutieli were performed to compare the results using GraphPad Prism, version 10 (GraphPad Software). No samples were excluded from analysis. We treated *P* values below 0.05 as significant differences. *$P < 0.05$, **$P < 0.01$ and ***$P < 0.001$. Details regarding number of replicates and representative data can be found in figure legends.

## Reporting summary

Further information on research design is available in the Nature Portfolio Reporting Summary linked to this article.

## Data availability

All mouse and human sequencing data generated and assembled for this project have been made available for open-access download at https://doi.org/10.5281/zenodo.15032578 (ref. 68) and https://doi.org/10.5281/zenodo.15115243 (ref. 69). Reference genomes mm10-2020-A and GRCh38-2020-A were used for mapping. Source data are provided with this paper.

## Code availability

All code used for computational analysis, and detailed methods on how to generate each figure panel starting with the raw data, are made available at https://doi.org/10.5281/zenodo.15032578 (ref. 68) and https://doi.org/10.5281/zenodo.15115243 (ref. 69).

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

**Acknowledgements** The authors thank members of the Littman laboratory for valuable discussion and critical reading of the manuscript; S. R. Schwab and M. Okuniewska for providing OT-II;UBC-GFP;*Rag1*⁻/⁻ mice; B. Spiegelman for providing *Prdm16*ᶠˡ/ᶠˡ mice; M. Oukka for providing *Il23r*ᵍᶠᵖ mice; S. Y. Kim and the NYU Rodent Genetic Engineering Laboratory (RRID:SCR_017925) for assistance with generation and rederivation of mutant mice; the NYU Genome Technology Core for scRNA-seq (RRID: SCR_017929); C. Loomis and the NYU Experimental Pathology Research Laboratory (RRID:SCR_017928) for histology of tissue samples; and the NIH Tetramer Core Facility for providing MHC class II tetramers. NYU core facilities are partially supported by NYU Cancer Center Support Grant P30CA016087 at the Laura and Isaac Perlmutter Cancer Center. L.F. is supported by a Cancer Research Institute Irvington Postdoctoral Fellowship (CRI2690). R.U. is supported by a Clinical Scientist Career Development Award (K08CA283272), Stand Up to Cancer Phillip A. Sharp Innovation in Collaboration Award (SU2C-AACR-PS-36), and by the Rosenfield and Glassman Foundation. F.M.C. is supported by the National Institutes of Health (T32AL100853, F32AI181496). This work was supported by R01AI158687 (D.R.L.) and the Howard Hughes Medical Institute (D.R.L.).

**Author contributions** L.F. designed and performed most mouse experiments and analysed the data. M.P. and F.M.C. performed mouse experiments and analysed the data. G.R.-M. assisted with mouse experiments. M.P. performed bulk ATAC–seq and analysed the data. M.P. generated *Rorc(t)* CRE mutant mice. L.F., R.U. and G.R.-M. analysed histology. L.F. performed scRNA-seq and multiome scRNA-seq/scATAC–seq of mouse samples and contributed to computational analysis. R.U. performed scRNA-seq of human samples. A.G. arranged and acquired human tissues. R.U. performed scRNA-seq and scATAC–seq computational analysis. L.F., R.U. and D.R.L. wrote the manuscript, with input from the other authors. D.R.L. supervised the research and contributed to experimental design.

**Competing interests** D.R.L. is a cofounder of Vedanta Biosciences and ImmunAI, on the advisory boards of IMIDomics, Sonoma Biotherapeutics, NILO Therapeutics and Evommune, and on the board of directors of Pfizer Inc. All other authors declare no competing interests.

**Additional information**
**Correspondence and requests for materials** should be addressed to Dan R. Littman.

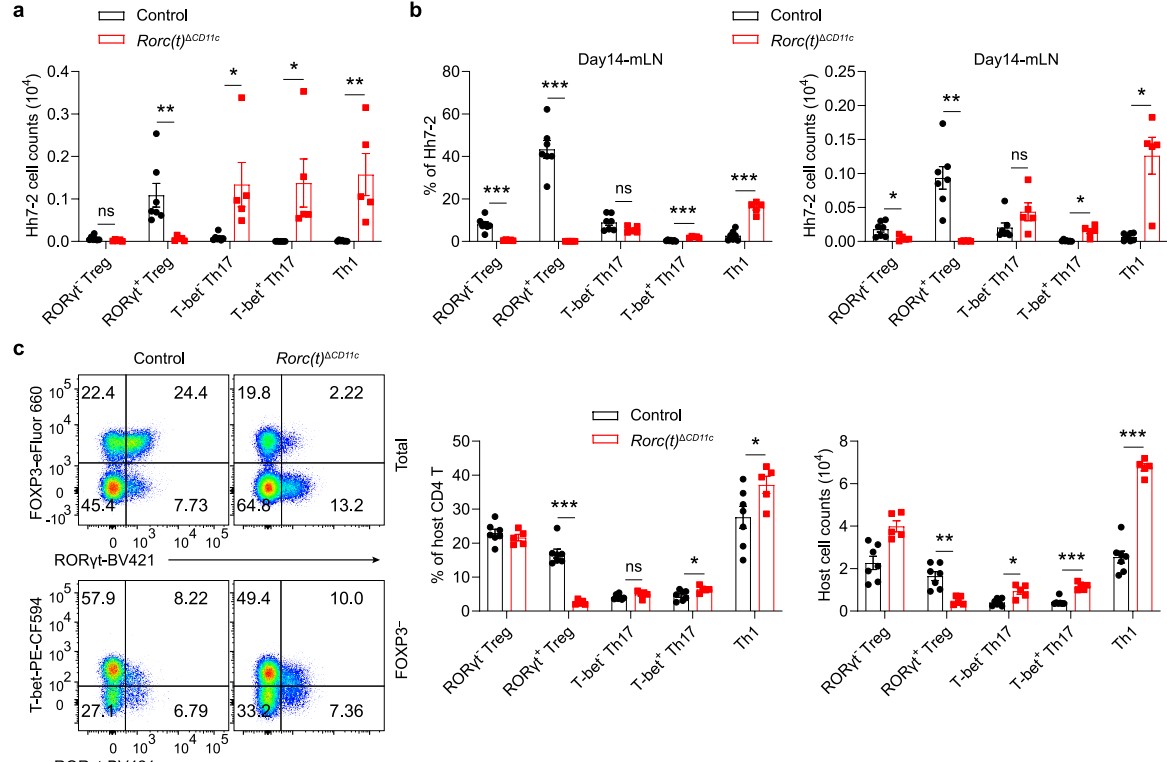

**Extended Data Fig. 1 | RORγt expression in CD11c lineage APCs is necessary for directing the differentiation of gut microbiota-specific pTregs.** **a**, Numbers of *Hh*-specific pTreg, Th17 and Th1 cells in the LILP of *Hh*-colonized control (n = 7) and *Rorc(t)^ΔCD11c* (n = 5) mice at 14 days after adoptive transfer of naïve Hh7-2tg CD4⁺ T cells. **b**, Phenotype of *Hh*-specific T cells in the mLN of mice shown in **a**. **c**, Phenotype of host CD4 T cells in the LILP of mice shown in **a**,

with representative flow cytometry profiles (left) and aggregate quantitative data (right). The flow cytometry plots are gated on total (upper) and FOXP3⁻ (lower) host CD4 T cells (CD45⁺B220⁻TCRγδ⁻TCRβ⁺CD4⁺CD90.1⁻). Data are pooled from two independent experiments. Data are means ± s.e.m.; ns, not significant; *P < 0.05, **P < 0.01 and ***P < 0.001; statistics were calculated by unpaired two-sided t-test.

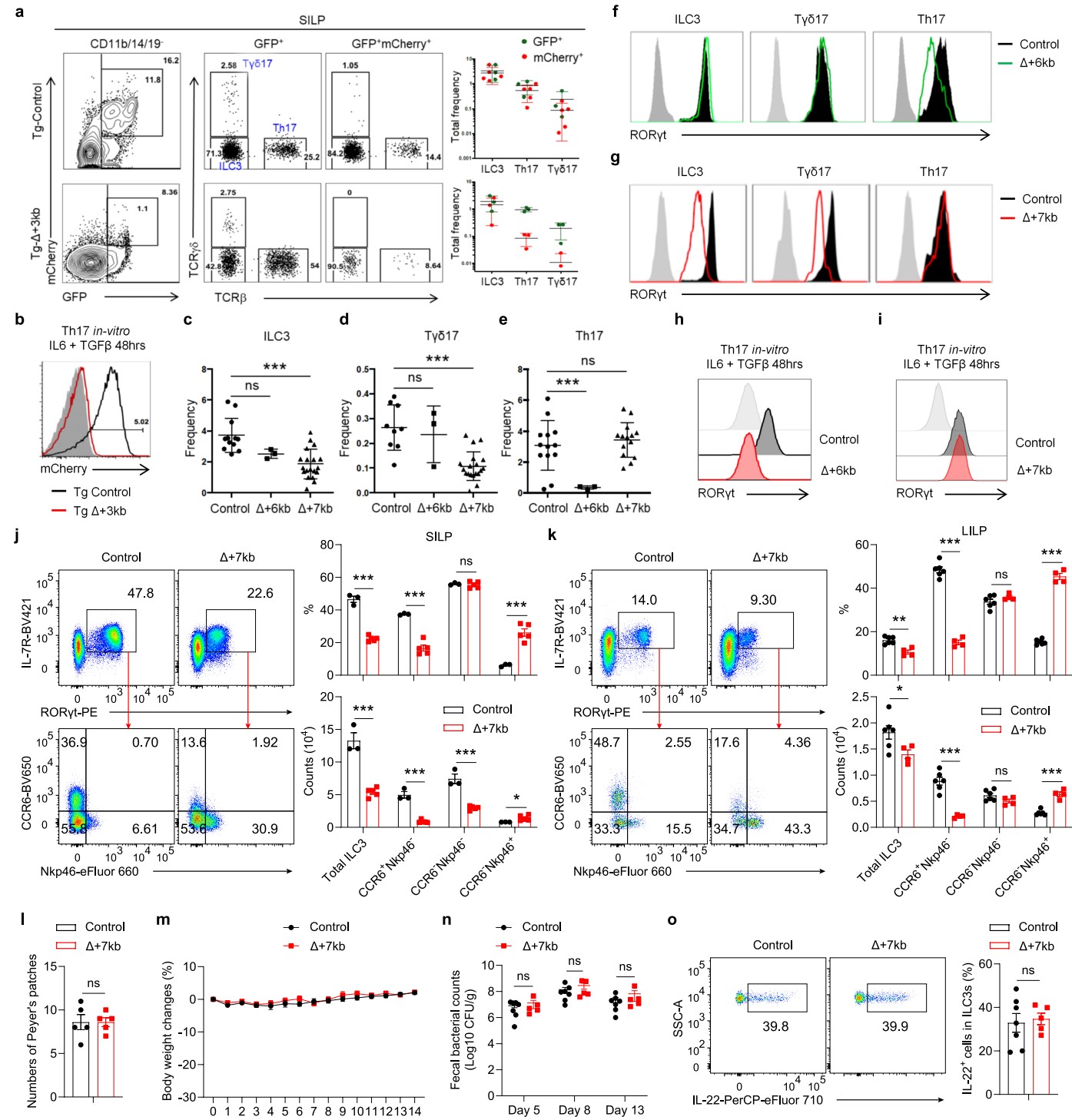

**Extended Data Fig. 2 | Characterization of lineage-specific *Rorc(t)* cis-regulatory elements. a,b,** Comparison of SILP GFP+ and mCherry+ populations (**a**) as well as mCherry expression of in vitro differentiated Th17 cells (**b**) in BAC transgenic mice bred to RORγt-GFP knock-in/knockout reporter mice. **a,** Tg (Control *Rorc(t)*-mCherry);*Rorc(t)*^wt/gfp^ mice, n = 3; Tg (Control *Rorc(t)*-mCherry) mice, n = 2; Tg (Δ+3 kb *Rorc(t)*-mCherry);*Rorc(t)*^wt/gfp^ mice, n = 3. **c,** Frequencies of SILP ILC3 in control (n = 12),Δ+6 kb (n = 3) and Δ+7 kb (n = 19) mice. **d,** Frequencies of SILP Tγδ17 in control (n = 9), Δ+6 kb (n = 3) and Δ+7 kb (n = 19) mice. **e,** Frequencies of SILP Th17 in control (n = 13),Δ+6 kb (n = 3) and Δ+7 kb (n = 14) mice. **f-i,** RORγt expression in the SILP RORγt+ populations (**f,g**) and in vitro differentiated Th17 cells (**h,i**) from mice shown in **c-e.j,k,** Phenotype of ILC3 (CD45+Lin−CD127+RORγt+) subsets in the SILP (**j**) and LILP (**k**) of control and Δ+7 kb mice. SILP: control mice, n = 3; Δ+7 kb mice, n = 5. LILP: control mice, n = 6; Δ+7 kb mice, n = 4. **l,** Numbers of Peyer's patches in control (n = 5) and Δ+7 kb (n = 5) mice. **m-o,** Body weight changes (**m**), fecal *C. rodentium* counts (**l**) and frequencies of LILP IL-22+ ILC3 (**o**) (day 14, ex vivo stimulation with IL-23) in control (n = 7) and Δ+7 kb (n = 5) mice post infection. Data in **m-o** are pooled from two independent experiments. Data in **j,k** are representative of two (**j,k**) or three (**l**) independent experiments. Data are means ± s.d. (**a,c-e**) or means ± s.e.m. (**j-o**); ns, not significant; *P < 0.05, **P < 0.01 and ***P < 0.001; statistics were calculated by unpaired two-sided t-test.

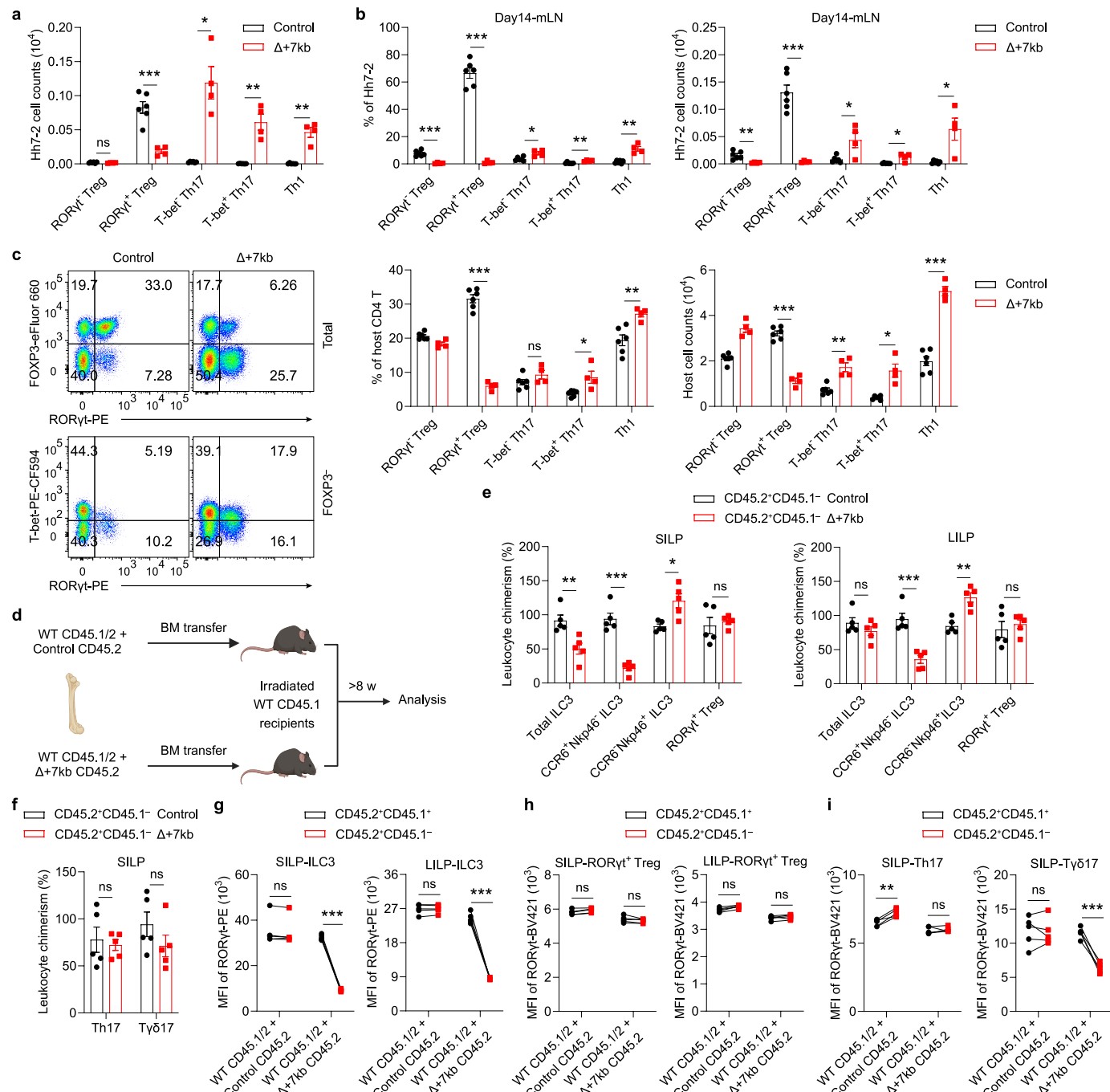

**Extended Data Fig. 3 | *Rorc(t)* +7 kb regulates RORγt⁺ Treg in a cell-extrinsic manner. a**, Numbers of *Hh*-specific T cells in the LILP of *Hh*-colonized control (n = 6) and Δ+7 kb (n = 4) mice at 14 days after adoptive transfer of naïve Hh7-2tg CD4⁺ T cells. **b**, Phenotype of *Hh*-specific T cells in the mLN of mice shown in **a**. **c**, Phenotype of host CD4 T cells in the LILP of mice shown in **a**. **d**, Experimental design for the bone marrow (BM) chimeric experiments in **e-i**. Created in BioRender; Fu, L. (2025) https://BioRender.com/o19q348. **e,f**, Relative CD45.2⁺CD45.1⁻ leukocyte chimerism normalized to CD45.2⁺CD45.1⁻ splenic B cells. n = 5 per group. **g-i**, RORγt mean fluorescence intensity (MFI) of CD45.2⁺CD45.1⁺ and CD45.2⁺CD45.1⁻ RORγt⁺ cells in the SILP and LILP. n = 5 per group. Data are representative of two independent experiments. Data are means ± s.e.m.; ns, not significant; *P < 0.05, **P < 0.01 and ***P < 0.001; statistics were calculated by unpaired two-sided t-test (**a-c,e,f**) and paired two-sided t-test (**g-i**).

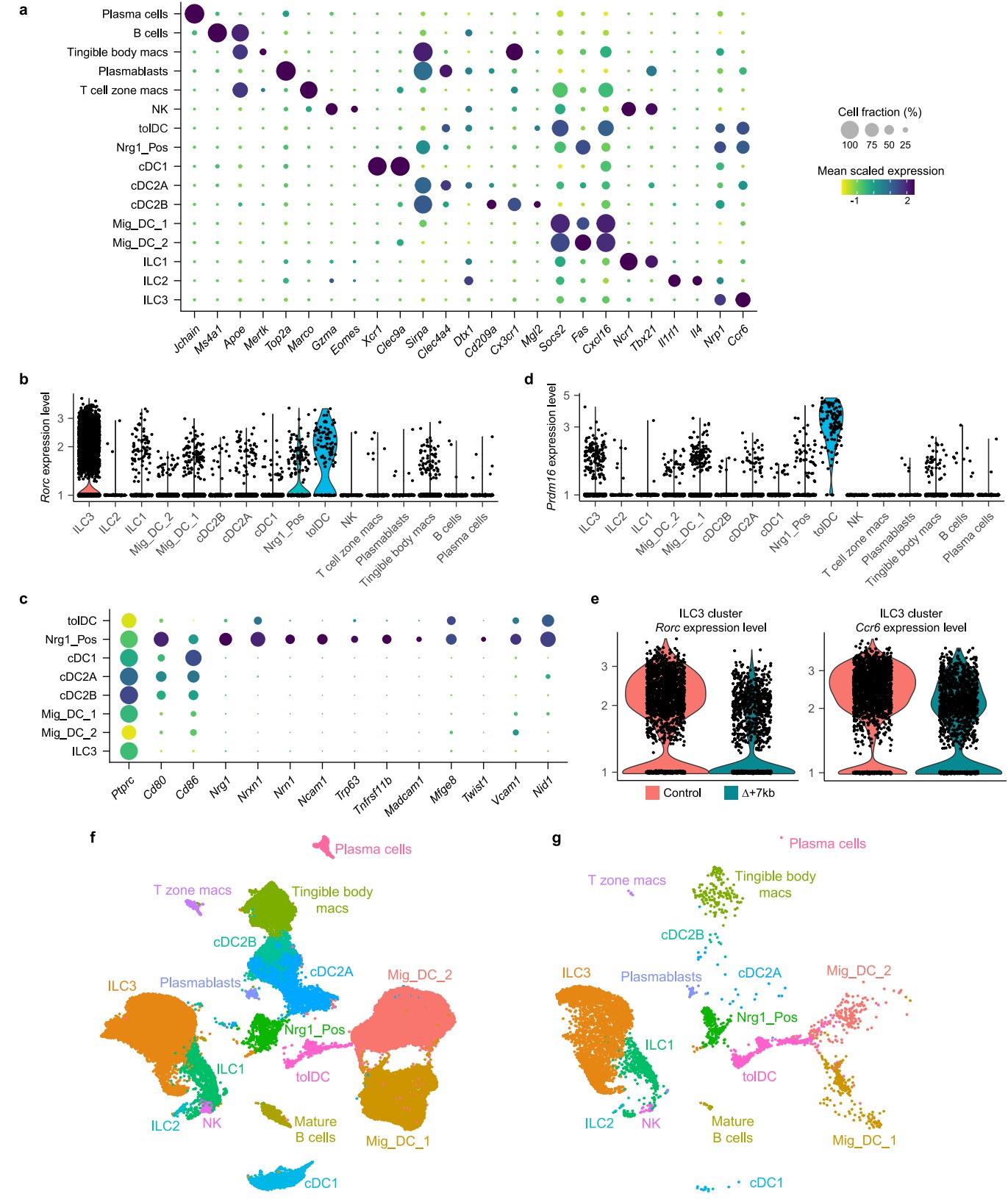

**Extended Data Fig. 4 | Sequencing annotation and analysis of Δ+7 kb mouse model mLN and multiome datasets. a**, Dot plot of all 16 cell types from Δ+7 kb mouse model mLN (mutant and control mice combined), demonstrating canonical genes used to annotate each cluster. **b**, Violin plot of *Rorc* expression across all clusters. **c**, Dot plot of select APC clusters, examining genes described for Nrg1_Pos as well as FRC/mTEC cell types. **d**, Violin plot of *Prdm16* expression across all clusters. **e**, Violin plots of *Rorc* and *Ccr6* expression within the ILC3 cluster, comparing Δ+7 kb mutant versus control biological conditions. **f**, Annotated UMAP with datasets combined from all murine experiments (all mutant and control animals, as well subsequent multi-ome experiment). **g**, Re-analysis of raw data from Akagbosu *et al.*, which was computationally integrated alongside data in **f**.

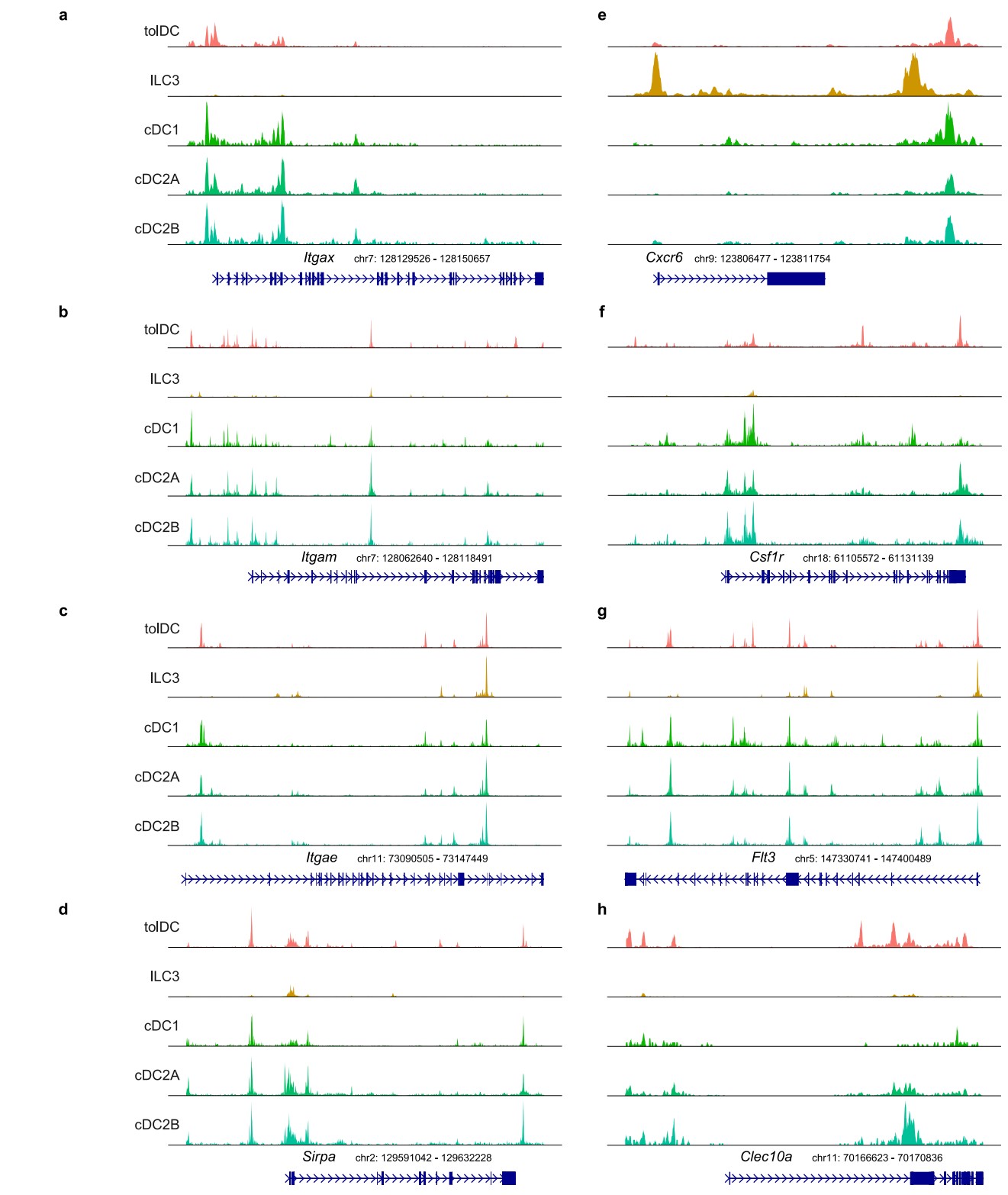

**Extended Data Fig. 5 | Comparing the epigenetic landscape across APC populations. a-h**, Chromatin accessibility profiles for *Itgax*, *Itgam*, *Itgae*, *Sirpa*, *Cxcr6*, *Csf1r*, *Flt3*, and *Clec10a* loci across the indicated mouse APC populations.

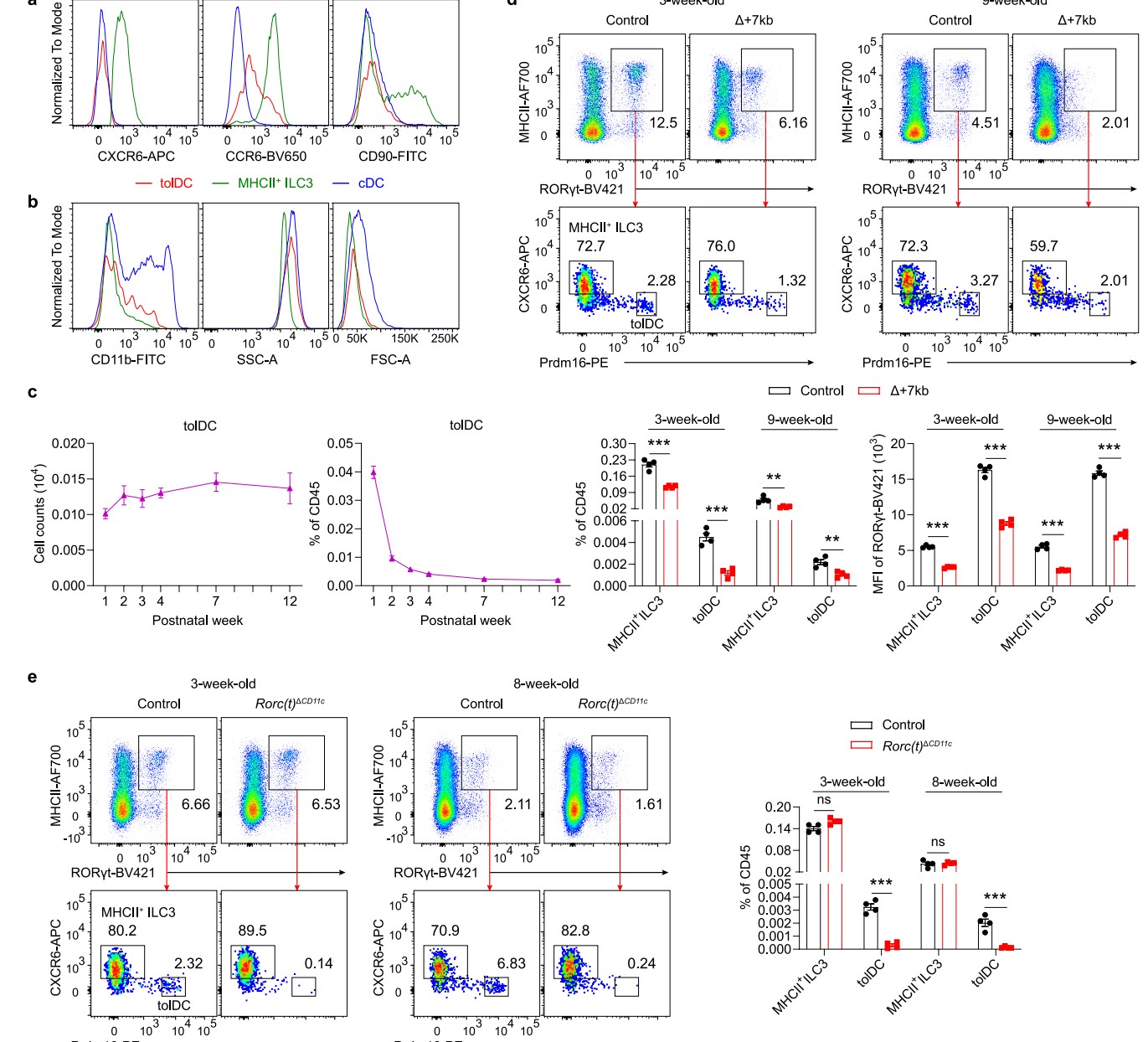

**Extended Data Fig. 6 | Selective loss of tolDC in *Rorc(t)^{ΔCD11c}* mice.**
**a**,**b**, Expression of the indicated proteins in tolDC, MHCII⁺ ILC3 and cDC, as gated in Fig. 2f. **c**, Numbers and frequencies of tolDC (CD45⁺Ly6G⁻B220⁻TCRγδ⁻TCRβ⁻ MHCII⁺RORγt⁺CXCR6⁻Prdm16^high) in mLN from week 1 to week 12. n = 6 (week 1); n = 5 (all other timepoints). **d**, Representative flow cytometry plots (top), frequencies (bottom left) and RORγt MFI (bottom right) of tolDC and MHCII⁺ ILC3 in the mLN of 3-week-old and 9-week-old control and Δ+7 kb mice.

n = 4 per group. **e**, Representative flow cytometry plots (left) and frequencies (right) of tolDC and MHCII⁺ ILC3 in the mLN of 3-week-old and 8-week-old control and *Rorc(t)^{ΔCD11c}* mice. n = 4 per group. The top flow cytometry plots are gated on CD45⁺Ly6G⁻B220⁻TCRγδ⁻TCRβ⁻. Data in **a**,**b**,**d**,**e** are representative of two (**d**,**e**) or three (**a**,**b**) independent experiments. Data are means ± s.e.m.; ns, not significant; **P < 0.01 and ***P < 0.001; statistics were calculated by unpaired two-sided t-test.

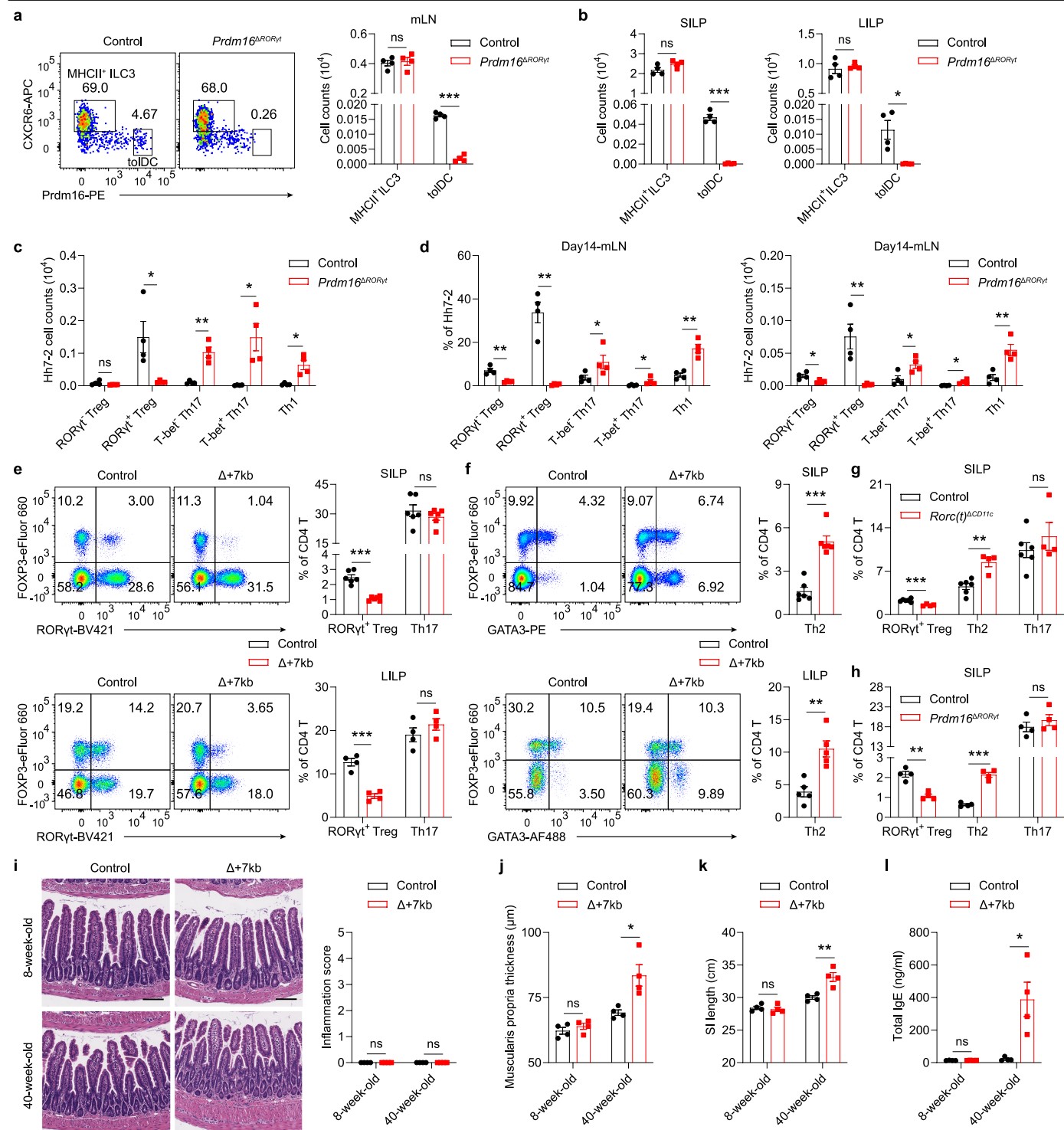

**Extended Data Fig. 7 | tolDC deficiency leads to type 2 gastrointestinal pathology. a,b**, Representative flow cytometry plots and numbers of tolDC and MHCII⁺ ILC3 in the mLN (**a**), SILP and LILP (**b**) of control (n = 4) and *Prdm16^ΔRORγt* (n = 4) mice. The flow cytometry plots are gated on CD45⁺Ly6G⁻B220⁻TCRγδ⁻ TCRβ⁻MHCII⁺RORγt⁺. **c**, Numbers of *Hh*-specific T cells in the LILP of *Hh*-colonized control (n = 4) and *Prdm16^ΔRORγt* (n = 4) mice at 14 days after adoptive transfer of naïve Hh7-2tg CD4⁺ T cells. **d**, Phenotype of *Hh*-specific T cells in the mLN of mice shown in **c**. **e,f**, Representative flow cytometry plots and frequencies of RORγt⁺ Treg, Th17 and Th2 cells in the SILP and LILP of control and Δ+7 kb mice. **e**: n = 6 per group (SILP), n = 4 per group (LILP); **f**: n = 6 per group (SILP), n = 5 per

group (LILP), per group. **g,h**, Frequencies of RORγt⁺ Treg, Th2 and Th17 cells in the SILP of control and mutant mice. **g**, control mice, n = 6; *Rorc(t)^ΔCD11c* mice, n = 4. **h**, control mice, n = 4; *Prdm16^ΔRORγt* mice, n = 4. **i,j**, H&E staining and inflammation score (**i**), and average muscularis propria thickness (**j**) of the distal small intestine sections in 8-week-old and 40-week-old control (n = 4) and Δ+7 kb (n = 4) mice. Scale bars, 100 μm. **k,l**, Small intestine length (**k**) and serum total IgE levels (**l**) of 8-week-old and 40-week-old control (n = 4) and Δ+7 kb (n = 4) mice. Data are representative of two (**a-d,g,h**) or three (**e,f**) independent experiments. Data are means ± s.e.m.; ns, not significant; *P < 0.05, **P < 0.01 and ***P < 0.001; statistics were calculated by unpaired two-sided t-test.

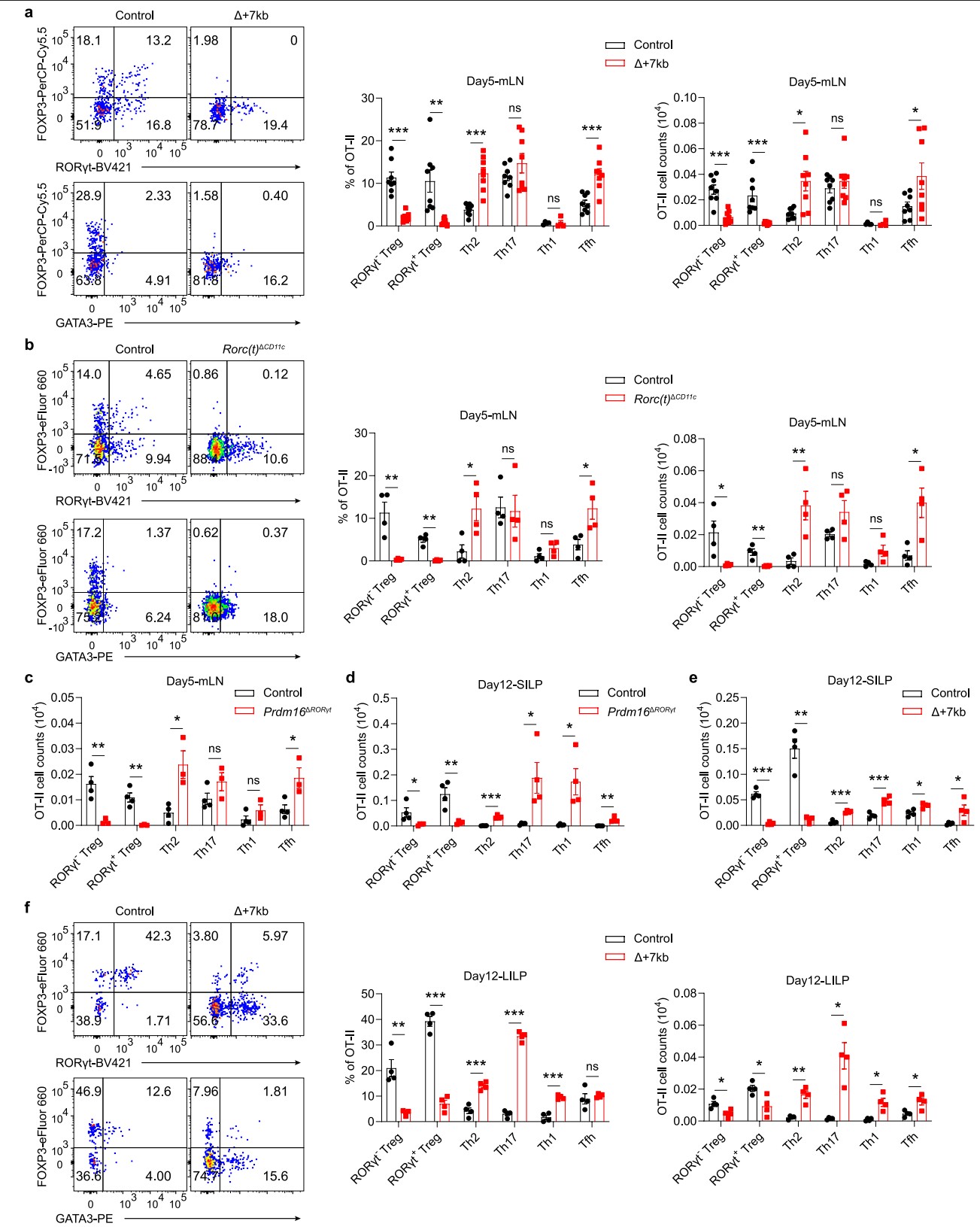

**Extended Data Fig. 8 | tolDC are essential for the differentiation of food antigen-specific pTregs. a-f**, Phenotype of OT-II pTreg, Th2, Th17, Th1 and Tfh cells in the mLN, SILP and LILP of OVA-fed control and mutant mice at 5 and 12 days post-adoptive transfer of naïve OT-II CD4⁺ T cells. **a**, mLN: control mice, n = 8; Δ+7 kb mice, n = 8. **b**, mLN: control mice, n = 4; *Rorc(t)^ΔCD11c* mice, n = 4. **c**, mLN: control mice, n = 4; *Prdm16^ΔRORγt* mice, n = 3. **d**, SILP: control mice, n = 4;

*Prdm16^ΔRORγt* mice, n = 4. **e**, SILP: control mice, n = 4; Δ+7 kb mice, n = 4. **f**, LILP: control mice, n = 4; Δ+7 kb mice, n = 4. Data in **a** are pooled from two independent experiments. Data in **b-f** are representative of two (**b-d**) or three (**e**,**f**) independent experiments. Data are means ± s.e.m.; ns, not significant; *P < 0.05, **P < 0.01 and ***P < 0.001; statistics were calculated by unpaired two-sided t-test.

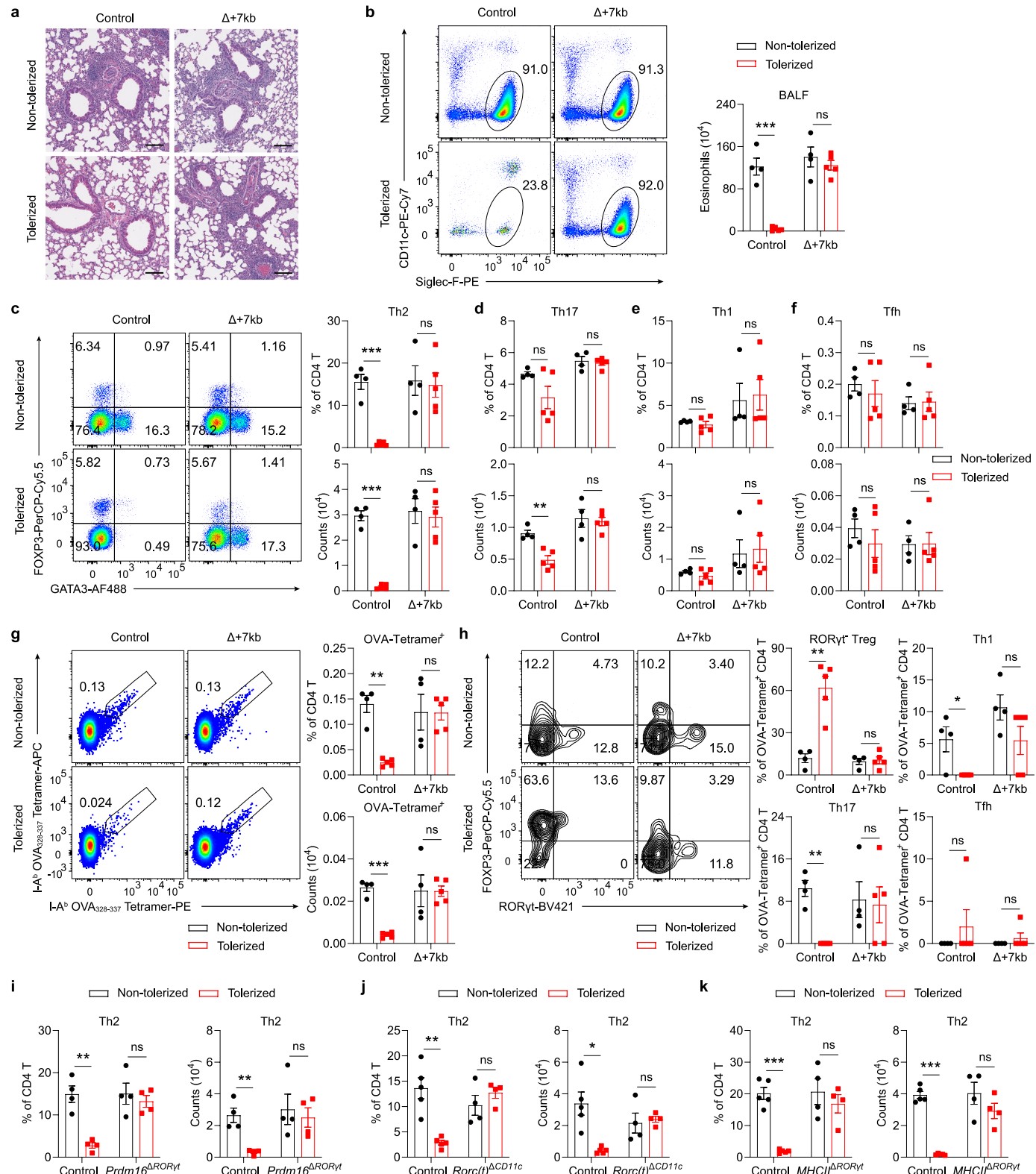

**Extended Data Fig. 9 | tolDC are required for establishing oral tolerance against allergic airway responses. a**, Magnified images of the lung sections in Fig. 4b. Scale bars, 100 μm. **b-h**, Eosinophil numbers in the BALF (bronchoalveolar lavage fluid) (**b**), phenotype of total CD4 T cells (**c-f**) and OVA:I-A^b tetramer^+ CD4 T cells (**g,h**) in the lung of control and Δ+7 kb mice shown in Fig. 4c–f. Flow cytometry plots in **g,h** were generated by concatenating the samples from each group. Non-tolerized mice, n = 4 per group; tolerized mice, n = 5 per group.

**i**, Phenotype of total Th2 cells in the lung of control and *Prdm16*^ΔRORγt shown in Fig. 4g. n = 4 per group. **j**, Phenotype of total Th2 cells in the lung of control (n = 5 per group) and *Rorc(t)*^ΔCD11c (n = 4 per group) shown in Fig. 4h. **k**, Phenotype of total Th2 cells in the lung of control (n = 5 per group) and *MHCII*^ΔRORγt (n = 4 per group) shown in Fig. 4i. Data in **b-h** are representative of two independent experiments. Data are means ± s.e.m.; ns, not significant; *P < 0.05, **P < 0.01 and ***P < 0.001; statistics were calculated by unpaired two-sided t-test.

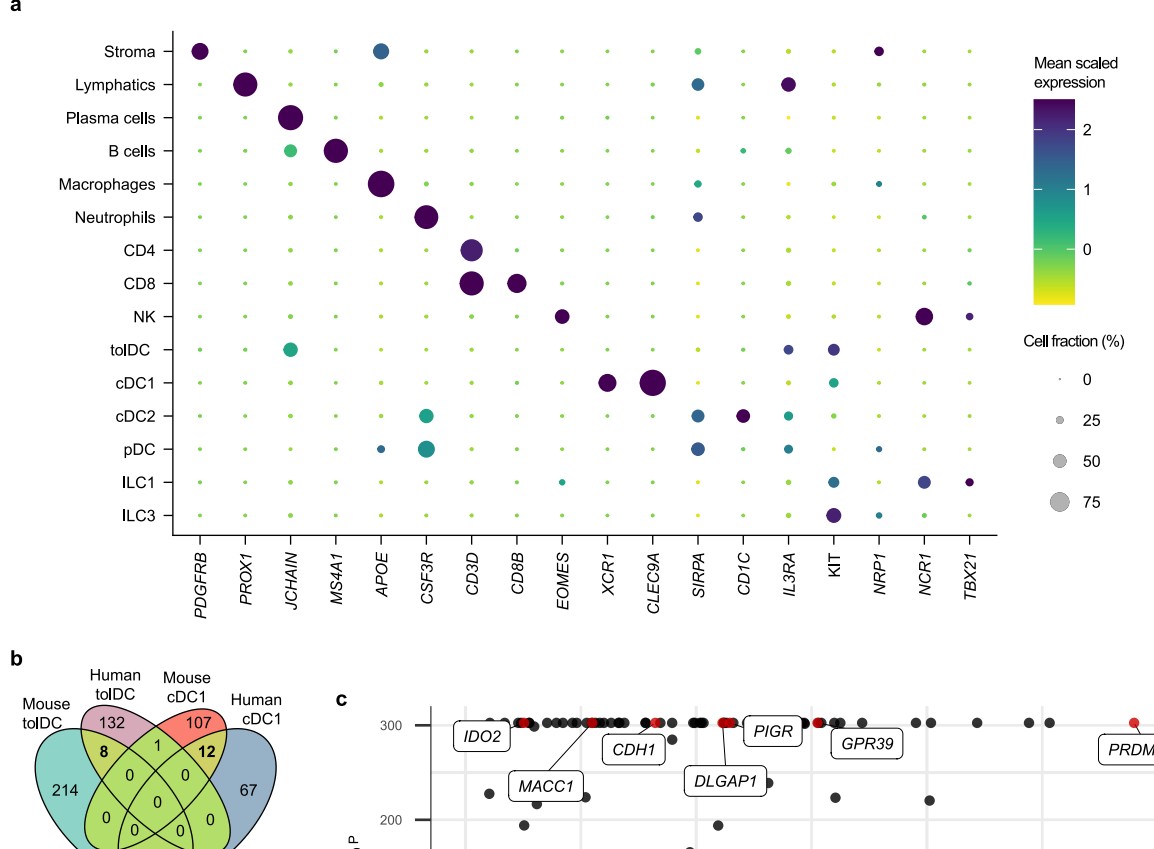

**Extended Data Fig. 10 | Analysis of differentially upregulated genes shared by human and mouse tolDC. a**, Dot plot of all 15 cell types from human mLN sc-RNA-seq experiment, demonstrating canonical genes used to annotate each cluster. **b**, Enumeration of differentially upregulated genes above a threshold of $Log_2$(Fold Change) = 3.2 after integrating all available data, from all tissues, for indicated murine and human APC populations. Overlaps within the Venn diagrams demonstrate conserved genes. **c**, Volcano plot of all 141 genes differentially upregulated in human tolDC, with explicit annotation of the 8 genes shared with mouse tolDC, as seen in **b**. P values (computed via FindMarkers Seurat algorithm) are two-sided and adjusted by Bonferroni correction using all genes in the dataset.

# Reporting Summary

## Statistics

For all statistical analyses, confirm that the following items are present in the figure legend, table legend, main text, or Methods section.

| n/a | Confirmed | |
|---|---|---|
| ☐ | ☒ | The exact sample size (*n*) for each experimental group/condition, given as a discrete number and unit of measurement |
| ☐ | ☒ | A statement on whether measurements were taken from distinct samples or whether the same sample was measured repeatedly |
| ☐ | ☒ | The statistical test(s) used AND whether they are one- or two-sided <br> *Only common tests should be described solely by name; describe more complex techniques in the Methods section.* |
| ☒ | ☐ | A description of all covariates tested |
| ☐ | ☒ | A description of any assumptions or corrections, such as tests of normality and adjustment for multiple comparisons |
| ☐ | ☒ | A full description of the statistical parameters including central tendency (e.g. means) or other basic estimates (e.g. regression coefficient) AND variation (e.g. standard deviation) or associated estimates of uncertainty (e.g. confidence intervals) |
| ☐ | ☒ | For null hypothesis testing, the test statistic (e.g. *F*, *t*, *r*) with confidence intervals, effect sizes, degrees of freedom and *P* value noted <br> *Give P values as exact values whenever suitable.* |
| ☒ | ☐ | For Bayesian analysis, information on the choice of priors and Markov chain Monte Carlo settings |
| ☒ | ☐ | For hierarchical and complex designs, identification of the appropriate level for tests and full reporting of outcomes |
| ☒ | ☐ | Estimates of effect sizes (e.g. Cohen's *d*, Pearson's *r*), indicating how they were calculated |

*Our web collection on statistics for biologists contains articles on many of the points above.*

## Software and code

Policy information about availability of computer code

| | |
|---|---|
| Data collection | Flow cytometry was performed on LSR II and Aria using FACSDiva v8.0.1 (BD Biosciences) or a Cytek Aurora using SpectroFLo v3.03 (Cytek Biosciences). Illumina NovaSeq X+ system was used for library sequencing. |
| Data analysis | BulkRNA-seq were mapped to the murine genome (mm10) with bowtie2 (2.2.3), filtered based on mapping score (MAPQ > 30, Samtools (0.1.19)), and duplicates removed (Picard). Cell Ranger (v7.1) was used to align genomes for single cell experiments. Seurat (v5.1) was used for sc-RNA-seq computational analysis. All code used for analysis in this manuscript is available at https://doi.org/10.5281/zenodo.15032578 and https://doi.org/10.5281/zenodo.15115243. |

For manuscripts utilizing custom algorithms or software that are central to the research but not yet described in published literature, software must be made available to editors and reviewers. We strongly encourage code deposition in a community repository (e.g. GitHub). See the Nature Portfolio guidelines for submitting code & software for further information.

## Data

Policy information about availability of data

All manuscripts must include a data availability statement. This statement should provide the following information, where applicable:
- Accession codes, unique identifiers, or web links for publicly available datasets
- A description of any restrictions on data availability
- For clinical datasets or third party data, please ensure that the statement adheres to our policy

All mouse and human sequencing data generated and assembled for this project are made available for open-access download at https://doi.org/10.5281/zenodo.15032578 and https://doi.org/10.5281/zenodo.15115243. Reference genomes mm10-2020-A (mouse) and GRCh38-2020-A (human) were used for mapping.

## Research involving human participants, their data, or biological material

Policy information about studies with human participants or human data. See also policy information about sex, gender (identity/presentation), and sexual orientation and race, ethnicity and racism.

| | |
|---|---|
| Reporting on sex and gender | Our limited human biospecimens and public datasets were not designed to test sex or gender, and therefore our findings do not specifically apply to only one sex or gender. |
| Reporting on race, ethnicity, or other socially relevant groupings | *Please specify the socially constructed or socially relevant categorization variable(s) used in your manuscript and explain why they were used. Please note that such variables should not be used as proxies for other socially constructed/relevant variables (for example, race or ethnicity should not be used as a proxy for socioeconomic status).* *Provide clear definitions of the relevant terms used, how they were provided (by the participants/respondents, the researchers, or third parties), and the method(s) used to classify people into the different categories (e.g. self-report, census or administrative data, social media data, etc.)* *Please provide details about how you controlled for confounding variables in your analyses.* |
| Population characteristics | Lymph node tissue (our primary human data) was obtained from a de-identified male organ donor at age 22, with no known history of atopy or chronic disease. We utilize public datasets from human intestine (https://doi.org/10.1101/2021.03.28.437379), with male and female donors across ages 63-83 but otherwise de-identified. We utilize public datasets from adult human tonsil (https://doi.org/10.1073/pnas.2318710120) that was otherwise completely de-identified for age and gender. |
| Recruitment | *Describe how participants were recruited. Outline any potential self-selection bias or other biases that may be present and how these are likely to impact results.* |
| Ethics oversight | This study does not qualify as human subjects research, as confirmed by NYU Langone Institutional Review Board, because tissues were obtained from a de-identified deceased individual. |

Note that full information on the approval of the study protocol must also be provided in the manuscript.

# Field-specific reporting

Please select the one below that is the best fit for your research. If you are not sure, read the appropriate sections before making your selection.

☒ Life sciences          ☐ Behavioural & social sciences          ☐ Ecological, evolutionary & environmental sciences

For a reference copy of the document with all sections, see nature.com/documents/nr-reporting-summary-flat.pdf

# Life sciences study design

All studies must disclose on these points even when the disclosure is negative.

| | |
|---|---|
| Sample size | Three or more mice were used in each experiment. The precise number of animals for each experiment are indicated within each figure legend. These sample sizes were determined from our previous experience in evaluating T cell driven inflammatory responses, and from what is generally accepted in this filed (e.g., PMID 36071167, PMID 36070798). |
| Data exclusions | No samples were excluded from analysis. |
| Replication | All the findings on the main figures were replicated at least twice. The precise number of repeats are provided in the figure legend. All attempts were successful. |
| Randomization | Allocation into sample groups was random. In addition, all control mice were from the same litter. Both males and females were used. |
| Blinding | Histological analysis was a fully blinded process. However, the remaining experiments were not blinded, since the induction of inflammatory responses versus control experiments, and their serial monitoring, requires re-visiting the same animals in the same cages each day. |

# Reporting for specific materials, systems and methods

We require information from authors about some types of materials, experimental systems and methods used in many studies. Here, indicate whether each material, system or method listed is relevant to your study. If you are not sure if a list item applies to your research, read the appropriate section before selecting a response.

## Materials & experimental systems

| n/a | Involved in the study |
|---|---|
| ☐ | ☒ Antibodies |
| ☒ | ☐ Eukaryotic cell lines |
| ☒ | ☐ Palaeontology and archaeology |
| ☐ | ☒ Animals and other organisms |
| ☒ | ☐ Clinical data |
| ☒ | ☐ Dual use research of concern |
| ☒ | ☐ Plants |

## Methods

| n/a | Involved in the study |
|---|---|
| ☒ | ☐ ChIP-seq |
| ☐ | ☒ Flow cytometry |
| ☒ | ☐ MRI-based neuroimaging |

## Antibodies

| | |
|---|---|
| Antibodies used | The following monoclonal antibodies were purchased from Abcam, Thermo Fisher, BD Biosciences or BioLegend: Prdm16 (EPR24315-59), CD3ε (145-2C11), CD4 (RM4-5), CD11b (M1/70), CD11c (N418), CD25 (PC61.5), CD40 (3/23), CD44 (IM7), CD45 (30-F11), CD45.1 (A20), CD45.2 (104), CD62L (MEL-14), CD90.1 (HIS51), CD90.2 (53-2.1), IL-7R (SB/199), CXCR6 (SA051D1), CCR6 (140706), Nkp46 (29A1.4), MHCII I-A/I-E (M5/114.15.2), Ly6G (1A8), Siglec-F (E50-2440), B220 (RA3-6B2), TCR Vα2 (B20.1), TCRβ (H57-597), TCR Vβ5.1/5.2 (MR9-4), TCR Vβ6 (RR4-7), TCRγδ (GL3), Foxp3 (FJK-16s), RORγt (B2D or Q31-378), GATA3 (TWAJ), T-bet (O4-46), BCL6 (K112-91), and IL-22 (IL22JOP). Anti-mouse CD16/32 (Clone 2.4G2, Bio X Cell BE0307R025MG) was used to block Fc receptors. Live/dead fixable blue (ThermoFisher) was used to exclude dead cells. I-Ab OVA328-337 tetramers (HAAHAEINEA) were provided by the NIH Tetramer Core Facility. |
| Validation | All commercially available antibodies are routinely tested by the vendor. |

## Animals and other research organisms

Policy information about studies involving animals; ARRIVE guidelines recommended for reporting animal research, and Sex and Gender in Research

| | |
|---|---|
| Laboratory animals | B6.Cg-Gt(ROSA)26Sortm14(CAG-tdTomato)Hze/J, (Jax 007914), C57BL/6 mice (Jax 000664), CD45.1 mice (B6.SJL-Ptprca Pepcb/BoyJ, Jax 002014), CD90.1 mice (B6.PL-Thy1a/CyJ, Jax 000406) and Cd11ccre mice (B6.Cg-Tg(Itgax-cre)1-1Reiz/J, Jax 008068) were purchased from the Jackson Laboratories. Rorcfl/fl, Rorc(t)gfp/gfp, and Hh7-2tg mice were generated in our laboratory and have been described15,17,43. Il23rgfp mice44 were provided by M. Oukka. OT-II;UBC-GFP mice28,45 were provided by S. R. Schwab. Tg (Δ +3kb Rorc(t)-mCherry), Rorc(t) +6kb-/-and Rorc(t) +7kb-/- mice were generated as described in 'Generation of BAC transgenic reporter and CRISPR knockout mice'. |
| Wild animals | No wild animals were involved. |
| Reporting on sex | Both males and females were used in this study.  We did not observe any sex-specific phenotypes. |
| Field-collected samples | There were no field-collected samples. |
| Ethics oversight | All animal procedures were performed in accordance with protocols approved by the Institutional Animal Care and Usage Committee of New York University School of Medicine. |

Note that full information on the approval of the study protocol must also be provided in the manuscript.

# Plants

| | |
|---|---|
| Seed stocks | *Report on the source of all seed stocks or other plant material used. If applicable, state the seed stock centre and catalogue number. If plant specimens were collected from the field, describe the collection location, date and sampling procedures.* |
| Novel plant genotypes | *Describe the methods by which all novel plant genotypes were produced. This includes those generated by transgenic approaches, gene editing, chemical/radiation-based mutagenesis and hybridization. For transgenic lines, describe the transformation method, the number of independent lines analyzed and the generation upon which experiments were performed. For gene-edited lines, describe the editor used, the endogenous sequence targeted for editing, the targeting guide RNA sequence (if applicable) and how the editor was applied.* |
| Authentication | *Describe any authentication procedures for each seed stock used or novel genotype generated. Describe any experiments used to assess the effect of a mutation and, where applicable, how potential secondary effects (e.g. second site T-DNA insertions, mosiacism, off-target gene editing) were examined.* |

# Flow Cytometry

## Plots

Confirm that:

☒ The axis labels state the marker and fluorochrome used (e.g. CD4-FITC).

☒ The axis scales are clearly visible. Include numbers along axes only for bottom left plot of group (a 'group' is an analysis of identical markers).

☒ All plots are contour plots with outliers or pseudocolor plots.

☒ A numerical value for number of cells or percentage (with statistics) is provided.

## Methodology

| | |
|---|---|
| Sample preparation | For isolation of cells from lymph nodes and spleens, tissues were mechanically disrupted with the plunger of a 1-ml syringe and passed through 70-μm cell strainers. Bone marrow cells were harvested by flushing out the marrow from cleaned bones using a syringe containing RPMI-1640 wash medium (RPMI-1640 with 3% FBS, 1% GlutaMAX, 1% penicillin–streptomycin, 10 mM HEPES, and 1 mM sodium pyruvate). Red blood cells were lysed with ACK buffer (Thermo Fisher). Cells in bronchoalveolar lavage fluids (BALF) were isolated by flushing the lung with two washes of 0.75 ml PBS via a catheter inserted into a cut made in the trachea. Lung tissues were cut into small pieces and digested in RPMI-1640 wash medium containing 0.5 mg/ml collagenase D (Sigma) and 0.5 mg/ml DNase I (Sigma) at 37°C for 45 min with shaking. After removal of Peyer's patches and cecal patches, the intestines were opened longitudinally, cut into 0.5 cm pieces, and washed in PBS twice. Intestines were then incubated with shaking in HBSS medium (without Ca2+ and Mg2+) containing 3% FBS, 1 mM DTT, 5 mM EDTA, and 10 mM HEPES at 37°C for 20 min twice. After washing with HBSS medium (without Ca2+ and Mg2+) containing 3% FBS and 10 mM HEPES, the tissues were then digested in RPMI-1640 wash medium containing 1 mg/ml collagenase D (Sigma), 0.25 mg/ml DNase I (Sigma), and 0.1 U/ml Dispase (Worthington) at 37°C for 35 min (small intestines) or 55 min (large intestines) with shaking. To isolate leukocytes from the lungs and intestines, the digested tissues were homogenized and passed through 70-μm cell strainers. Mononuclear cells were then collected from the interphase of an 80% and 40% Percoll gradient after a spin at 2,000 rpm for 20 min. |
| Instrument | Flow cytometric analysis was performed on an LSR II (BD Biosciences) or an Aria II (BD Biosciences). |
| Software | We used FACSDiva software to collect data, and performed analysis using FlowJo software (Tree Star). |
| Cell population abundance | Sort purity was determined to be 95% by running post sort sample. |
| Gating strategy | Naïve Hh7-2tg T cells were sorted as CD4+TCRβ+CD44low/-CD62L+CD25–Vβ6+ (Hh7-2tg). Naïve OT-II T cells were sorted as CD4+TCRβ+CD44low/-CD62L+CD25–Vα2+Vβ5.1/5.2+ (OT-II). |

☒ Tick this box to confirm that a figure exemplifying the gating strategy is provided in the Supplementary Information.

