## [Peer Review File · Nature]

Prdm16-dependent antigen-presenting cells induce tolerance to intestinal antigens

Corresponding Author: Dr Dan Littman

Version 0:

Reviewer comments:

Referee #1

(Remarks to the Author)

In this paper, Fu et al explore the molecular characteristics of the recently described population of unusual Rorgt+ cells found in mesenteric lymph node, that appear to play a crucial role in driving the generation of FoxP3+ Tregs in response to the intestinal microbiota. As has been shown previously, mice lacking the Rorgt+ cells preferentially generate Th1 and Th17 responses when colonized with *Helicobacter hepaticus*, although this dysregulation becomes Th2-polarized when Hh is not present. Importantly, these effects are not intrinsic to CD4+ T cells. A novel finding is the identification of a region +7kb from the Rorgt locus that is important in controlling the development of these cells and show that deleting this region prevents the induction of FoxP3+ Tregs and the tolerance of Th2-driven, allergic lung disease that occur after oral administration of ovalbumin as a dietary antigen. In addition, using selective expression of Prdm16 as a marker of the Rorgt+ population, the authors present evidence that similar cells may be present in human MLN. Many of these findings are novel and represent important advances in our understanding of this novel cell population. However a number of important issues are not addressed fully:

- 1) While the authors refer to the Rorgt+ cells as "APC" and this was shown in some of the previous papers, none of the current experiments show a direct role for them presenting antigen to T cells in vivo or in vitro. Thus it remains feasible that the differences in T cell polarization could reflect effects of the Rorgt+ cells in shaping the environment. The oral tolerance model provides a potential system for doing this.
- 2) Related to this, the work does not attempt to isolate the Rorgt+ cells from MLN, which would allow antigen presenting cell activity to be addressed. In addition, it would be important to see absolute numbers and surface phenotype of the cells in the different models, especially in the new $\Delta+7\text{kb}$ mouse.
- 3) Previous work on MLN and Peyer's patches has suggested that the presence of Rorgt+ APC is highly age dependent, with these cells appearing to be restricted to the neonatal period, the time at which induction of oral tolerance is likely to be most relevant. While I note that the scRNAseq experiments were conducted with 3 week old MLN, the other work, including that on oral tolerance, seems to have used older, adult mice. This issue needs commented on and if possible, additional experiments carried out to identify when the Rorgt+ cells appear in MLN in the authors' hands and how this influences their functions.
- 4) As the exact nature of this novel population has been controversial, it is disappointing that the authors do not use the opportunity of having single cell transcriptional data to make some detailed comparisons with conventional dendritic cells and ILC.
- 5) No information or images are presented on whether there is intestinal inflammation or pathology in any of the deletion models. This is crucial because the previous evidence indicates that the mice lacking antigen presentation or b8 integrin expression on Rorgt+ APC develop microbiota-driven inflammation, while other immunological KO models that lead to Th2 polarization can also develop intestinal pathology. The presence of such inflammation could be a confounding factory in the loss of oral tolerance.
- 6) While interesting and clearly of importance for understanding the use of the $\Delta+7\text{kb}$ mice, the authors could consider omitting some of the additional information related to its expression in other cell types.
- 7) It is clear that the defect in Rorgt+ cells is much more severe in the *Rorc(t) Δ CD11c* mice compared with the $\Delta+7\text{kb}$ mice and this warrants some comment. This is also where absolute numbers of cells would be useful.
- 8) The methods section indicates that the experiments used age and sex-matched controls. However it is now accepted that littermate controls are needed in all such studies, particularly those in which there a microbiota-dependent phenotype.
- 9) In contrast to the authors' conclusions about the deletion of Rorgt+ cells selectively driving Th2 expansion, Figure 2 and Extended Data Figure 5 seem to show clear increases in both Th1 and Th17 populations in the small and large intestinal

lamina propria, suggesting the immune dysregulation is more general. In this respect, if there is a Th2-specific polarization in the lung, should the PBS-treated Δ +7kb mice not show more susceptibility to the allergic inflammation here?

10) The selective expression of Prdm16 by the Ror γ ⁺ cells is interesting and could be useful in future studies of these cells. Do the authors have any information on what this molecule may be doing in this population and what its expression signifies?

11) While interesting, I have concerns about the interpretation of the human data showing an analogous population. For instance, there seems to be wide variation in the levels of expression of the 29 DEGs overlapping in mouse and human and one wonders what an equivalent volcano plot for non-overlapping DEGs would show. It is notable that AIRE is not expressed by the human cells, while there is no mention of other signature genes of the mouse cells such as Itgb8. Given the possible age-dependence of the cells in mice, a 22 year old human subject may also not be optimal.

Referee #2

(Remarks to the Author)

In this article the authors attempted to characterize the role of the APC subset implicated in driving tolerance to food antigens and Helicobacter. Their previous studies along with others suggested that ROR γ ⁺ expressing APCs regulated tolerance to the gut microbiota but the nature of tolerogenic APC subset remained unclear. In the present study the authors assessed the role of ROR γ ⁺ expression in APCs in inducing tolerance to food antigens. They identified that ROR γ ⁺ expression in CD11c cells was critical in the induction of tolerance to H. hepaticus-specific T cells. Further by analyzing the Rorc locus among different ROR γ ⁺ expressing cells they identify a +6 Kb and +7kb region that differentially regulated Th17, gdT17 and ILC3 responses. Transgenic mice lacking the +6kb Rorc region was associated with defective Th17 response but an intact gdT17 and ILC3 functions. In contrast mice lacking the +7kb region were associated with normal Th17 response but a defect in the ROR γ ⁺ expression in ILC3 and gdT17 cells. The authors thus assume the mice lacking +7kb region to be associated with a defective ROR γ ⁺ expression only in the innate but not the adaptive cells. Using these mice, they show that the ILC3s function were normal in a model of C. rodentium infection and therefore any defects with respect to T cells would be due to other ROR γ ⁺ APCs and not due to ILC3. Baseline analysis of the 7Kb Tg mice indicated a defect in the ROR γ ⁺ Treg cells with dysregulated Th2 responses. Notably assessment of tolerance induction to ovalbumin was defective in the 7kb Tg mice and in a mouse model of allergic lung inflammation failure to induce tolerance was associated with heightened eosinophilia and lung inflammation. scRNA seq from WT and Δ 7Kb Tg mice identified a cluster of PRDM16 expressing APCs to be defective in the Tg mice. Lastly the authors demonstrate from a single human sample that the presence of PRDM16 expressing APCs is conserved in humans.

Overall, the authors do not provide any direct evidence for the APC subset that is regulating oral tolerance. Based on their reliance on a Tg mouse that assumes defects in Ror γ ⁺ APC without demonstrating the defect and their failure to directly demonstrate the ROR γ ⁺ APC subset governing tolerance, the manuscript in its current version does not in my opinion achieve the priority to be published in Nature. This can change if the authors can demonstrate and provide direct evidence and underpin which ROR γ ⁺ subset (Thetis cells or ROR γ ⁺ DCs) is imparting tolerance to food antigens.

Major points

1) While the authors have placed much effort in disproving a role for ILC3 in driving peripheral tolerance to helicobacter hepaticus, very little has been done to pin down the APC subset involved in oral tolerance. The heightened emphasis on Helicobacter hepaticus specific tolerance and the role ILC3 detracts heavily from the major objective of the article, namely the identity and function of a specific ROR γ ⁺ APCs subset in peripheral oral tolerance.

2) The authors show that ROR γ ⁺ expressing CD11c APCs are the drivers of peripheral tolerance to the microbiota (this is somewhat of a redundant point from their previous story). Nevertheless, they use different ROR γ ⁺ expressing cells and perform ATACseq to assess the Rorc locus, allowing them to derive different Tg mice lacking the respective Rorc loci. Tg mice lacking the 7Kb Rorc region is associated with a defect in ROR γ ⁺ expression in ILC3 and gdT17 cells but uniquely the Th17 cells are normal. They assume that in these Tg mice the defect in ROR γ ⁺ is therefore restricted to the innate compartment and not in the adaptive system. They authors show that the ILC3 are functional in a C. rodentium mouse model of infection. However, the Tg mice lacking the 7Kb region are associated with a defect in ROR γ ⁺ Treg cell at baseline thus with a leap of faith they claim this is due to ROR γ ⁺ APCs and not ILC3. This is premature assumption, as just because ILC3s are impacted does not mean that other ROR γ ⁺ innate cells are also defective. The experimental evidence demonstrated does not convince me of this. The authors need to show a direct defect in the ROR γ ⁺ APCs. For example, based on the evidence provided, the 7Kb Tg mice despite showing a defect in the ILC3 compartment, the ILC3 were indeed functional. Thus, why wouldn't the ROR γ ⁺ APCs be functional even if their frequencies are less which from their scRNA seq does not appear to be the case.

3) While the authors demonstrate that the in vitro Th17 differentiation is functional in the 7Kb Tg mice, is this true also for in vitro ROR γ ⁺ Treg cell differentiation?

3) Can the authors provide the Rorc accessibility in Thetis cells, ROR γ ⁺ DCs or ROR γ ⁺PRDM16⁺ DCs. This analysis (at least from the Thetis cells and ROR γ ⁺ DCs should have been included in Fig1b). Do the authors have proof that the Rorc locus in Thetis cells/ROR γ ⁺DCs is dependent on the 7kb Rorc locus for their functionality? This analysis would have rendered some support in favor of using these mice.

4) In Fig 5e using scRNA-seq the authors do not find any difference between Thetis cells and many of the APC subsets. This brings us back to the identity of the ROR γ ⁺ APCs population(s) that are defective in the 7kb Tg mice. The only APC subset shown to be defective was the ROR γ ⁺ PRDM16⁺ APCs, and the authors hypothesize accordingly that it is this

subset of APC that is the tolerogenic subset but again have no proof for this.

- 5) Are there defects in the ROR γ t APCs in food allergic individuals? The availability of such human data would add weight to the observations described in this manuscript.
- 6) The use of allergic airway inflammation model to interrogate oral tolerance poses some challenges. Thus, in figure 3e and extended data fig 6 g,h there is loss of both the OVA-specific ROR γ t+ and ROR γ t- Treg cells in the Δ 7Kb mice, and the majority of the response is in the ROR γ t- Treg cell population. So is this system interrogating the ROR γ t+ Treg cell response as a mechanism of oral tolerance? It does not seem so from the data. Perhaps additional staining in the gut can clarify this issue. Better still, a gut specific oral tolerance system would be also add clarity.
- 7) I am also uncertain about Prdm16 as a mechanistic pathway in the defined APC subpopulation rather than just a marker for this subset. Humans with mutations in PRDM16 are not associated with tolerance breakdown but rather with cardiomyopathy. Do the authors have any data on the ROR γ t+ APC subpopulations in mice with either global or conditional Prdm16 deficiency?

Other comments

- 1) The gating for figure 1a is not clear. The text claims that these are ROR γ t and T-bet-expressing Hh-specific T cells but the figure itself seems to show total/CD4+ T cell staining? Showing the gating strategy would help.
- 2) In figure 2, the use of OT-II cells as means of interrogating OVA-specific responses is not sufficient. Endogenous OVA-specific T cell responses need to be measured by tetramer staining. If mice were orally sensitized with OVA do the authors observe tetramer staining in the ROR γ t+ pTreg cell population and is this staining decreased in the Δ 7KB mice?
- 3) If available, please provide the atopic status of the human subject used in the scRNA Seq analysis?

Referee #3

(Remarks to the Author)

This manuscript by Fu et al builds on recent advances in the field of ROR γ t+ antigen-presenting cell biology, most notably a trio of papers published in 2022 – including one from the same group. This study reports several observations that expand this prior work to demonstrate i) that different enhancer elements within the Rorc locus regulate specific ROR γ t-expressing cell populations, ii) that previously reported induction of ROR γ t+ FoxP3+ Treg is dependent upon an enhancer region +7kb from the transcription start site of Rorc, iii) that in this context the failure to appropriately induce Treg results in elevated Th2 cells, and a subsequent inability to mount oral tolerance to a model antigen, and iv) that a subset of ROR t+ APC marked by Prdm16 transcript is altered in the above animal model.

While this is a rapidly emerging area, the aforementioned studies raised a number of open questions and confusion around the previously reported roles of other ROR γ t+ MHCII+ cells (e.g. ILC3 and eTACs), as well as the roles of classical dendritic cells in these processes. Thus, the observation that Prdm16 transcript may mark a subset of ROR γ t+ APCs is a useful finding that could help clarify and consolidate the currently confusing nomenclature surrounding the expanding family of ROR γ t+ APC. As such further efforts to try and reconcile the current data with previous findings would be of benefit.

Overall, there are several interesting observations that largely build on existing concepts. However, while the first half of the manuscript is well developed the second half is largely built on assumptions and correlations and suffers from a lack of direct evidence and supporting data, as well as novel mechanistic insight that would provide a significant conceptual advance over the prior work in this area. As such the manuscript in its current form is not sufficiently mature and multiple interpretations and conclusions are not sufficiently justified by the data provided at this point.

Specific concerns are listed below:

Analysis of the Δ 7kb mice reveals preferential effects on ROR γ t-expression in ILC3 and a loss of ILC3 in this system (Extended Data Fig2e), most notably of the CD127+ CCR6+ “LTi-like ILC3” subset (Extended Data Fig 3a). In contrast no flow cytometry quantification and in depth analysis of other CD127- ROR γ t+ APCs is provided (including both AIRE+ eTACs and/or Prdm16+ cells), nor is ROR t expression validated to also be perturbed by deletion of the +7kb enhancer in other ROR γ t+ APC populations. It is therefore seemingly counterintuitive that the early data demonstrating marked changes to ILC3 is dismissed later in the manuscript and differences attributed to the Prdm16+ population alone with little direct supporting evidence. This raises a number of issues that should be addressed:

- As non-ILC3 ROR γ t+ APCs have been reported to be CD127 negative (and CXCR6 negative) a quantification of the whole ROR γ t+ APC family in these mice with a much more granular flow panel is required to clarify the discrepancies between the reported changes in abundance of ILC3 and the later observation of decreased Prdm16+ cells (but not ILC3) via scRNA seq – as well as to validate the differences found in transcriptomic data.
- Similarly, to support the authors conclusions and interpretations, the expression levels of ROR γ t in mice lacking specific enhancers should be shown for TC1/eTAC populations, and the Prdm16+ cell population identified later to provide important proof of concept and expand on Extended Data Fig 2.

- The reliability of quantification of cell frequency/numbers by scRNAseq of pooled cells (essentially n=1, one independent experiment) is a matter of debate, thus its critical any changes in cell population frequency by this method are validated by other means (e.g. flow cytometry as above), particularly given the weight placed on the former observation for the interpretations and narrative of the manuscript.
- The rationale for ruling out ILC3 as the Treg-inducing APC in this study is the observation that the capacity of ILC3 to produce IL-22 is retained in the d+7kb mice despite reduced ROR γ t expression (Extended Data Fig 3). The authors cite two previous papers (Withers et al Nat Med, Fiancette et al Nat Immunol) that have previously reported similar persistence of IL-22 in the absence of this transcription factor. However, Fiancette et al appear to show that deletion of ROR t in ILC3 does reduce MHCII levels (Extended Data Fig6 of that paper) and a number of other genes, despite IL-22 levels being retained. Additionally other auxiliary molecules such as OX40L expressed by these cells have Treg promoting function, and could feasibly be perturbed by loss of ROR γ t. Thus, the authors cannot discount a reduced antigen presenting capacity of ILC3 in d+7kb mice that could explain their observations, and broader transcriptional dysregulation when ROR γ t expression is perturbed.
- Do d+7kb retain normal cryptopatches and ILFs? Given the importance of these sites for Treg induction and/or maintenance, it's feasible that reductions in provision of lymphotoxin etc by ROR γ t+ cells could disrupt Treg induction indirectly via loss of lymphoid tissue architecture as opposed to direct antigen-presenting capacity. Interpretation of experiments in Figure 1 with RORc(t)gfp/gfp are similarly confounded by the loss of lymphoid tissue formation, and this should be explicitly stated in the text to acknowledge this potential caveat.
- Conversely, the direct evidence to support the interpretation that Prdm16+ are the only ROR γ t+ APC required for appropriate induction of ROR γ t+ Tregs is drawn from almost exclusively from the single cell data, where capture of this cluster was lost in d+7kb mice – a purely correlative observation that does not provide direct evidence for the authors conclusions. Can Prdm16+ cells be isolated from wild type mouse using markers identified via the transcriptional signatures and demonstrated to induce ROR γ t+ Treg in an ex vivo co-culture system?
- How are Prdm16+ cells transcriptionally altered in d+7kb mice or CD11c mediated deletion of Rorc? Is there evidence that the cells that remain have a reduced antigen presenting capacity in vivo or ex vivo? Are there broader changes to the transcriptome caused by loss of ROR t activity or do these cells simply not develop and/or persist? Generally reporting of transcriptional differences between all ROR t+ subsets in the single cell data sets should be reported to help the reader understand the quality and magnitude of changes in these mouse models.
- Much of the evidence for ROR t+ APCs being the tolerogenic APC responsible for ROR γ t+ Tregs to date has come from comparisons between RORcCre driven deletion of H2-Ab1 in comparison with RoraCre (hits ILC3 but not other ROR γ t APC subsets) and CD11cCre (which hits all ROR γ t+ APCs to differing degrees, and is a notoriously "leaky" Cre system). Thus, the identification of Prdm16 as a unique or unifying signature of this subset of ROR γ t+ APCs provides a unique opportunity to advance understanding of this emerging area. The authors should therefore demonstrate that targeting antigen presenting capacity with a Prdm16Cre, or depleting only the Prdm16+ subset of cells is sufficient to ablate ROR γ t+ Treg induction and rule out redundant/contextual contributions from other MHCII+ ROR γ t+ APC subsets including ILC3 and eTAC.
- What are the activations states, frequencies and numbers of ILC2 in the d+7kb mice? Given that loss of ILC3 populations has been reported to lead to reciprocal expansion of ILC2 (Spencer et al Science 2014) and ILC2 are potent inducers of GATA-3+ Treg via OX40L to restrain Th2 responses (Stockis et al Science Immunology 2024).
- What is the evidence that the enhanced Th2 response at steady state (Extended Data Fig4) is a direct consequence of disrupted Treg induction? While this is a logical assumption there is no direct experiments to demonstrate this and thus, it cannot be ruled out that indirect effects due to disruption of type 3 circuits. Notably ROR γ t+ lymphocytes including ILC3 and γ dT cells have critical roles in modulating barrier tissue integrity, epithelial metabolism and nutrient transport and the microbiota which could feasibly alter responses to dietary antigens and result in altered Th2 responses in a Treg-independent manner. Does deletion of ROR γ t in FoxP3 lineage cell recapitulate this phenotype?
- The abstract states the authors demonstrate "ROR γ t+ APCs are required for differentiation of food antigen-specific pTregs", which is arguably an overstatement. While OVA-induced oral tolerance is undoubtedly a useful model it is unclear if this is sufficient to support the authors interpretation that ROR γ t+ APC-induction of Treg is required for tolerance to food-antigens more generally. Do d+7kb mice exhibit evidence of food allergy or loss of tolerance to normal dietary antigens or any spontaneous pathology without the addition of a model antigen?
- Of note the authors mention Prdm16+ cells as well as TCI both express Aire transcripts in their data set. As such that suggests the identity of these cells is in line with that of ROR γ t+ eTACs and not the previously described ROR γ t+ DC-like cell / TCII / TCIV. This should be discussed in the text to help readers reconcile the findings with the previous nomenclature.
- The authors should provide a direct comparison of the gene signatures of Prdm16+ cells from their murine scRNA seq with previously reported signatures for ROR γ t+ eTACs and TCII-IV populations. In addition a focused bubble plot comparing these signatures demonstrating expression of known distinguishing factors including Il7r, Dpp4, Siglecg, Aire, Spi1, Itgb8, Itgax, Cd80/Cd86 etc would be especially useful.

- Moreover, it could be argued these data is in fact more in line with the previous reports that the ROR γ t+ APC pool constitutes ILC3 and two distinct populations of ROR γ t+ eTACs (e.g. Yamano et al J Exp Med 2019, Lyu et al Nature 2022). Comparison of transcriptional data sets with the annotations of data from these prior scRNA seq data sets would also prove a useful comparison to clarify commonalities and nomenclature.

- A recent study from the Powrie lab (Gu et al Nature 2024) elegantly mapped Treg interaction networks in the gut and lymph nodes, with the data publicly available. Notably, they observed multiple MHCII+/mid ILC3 subsets that differentially localised with Tregs in different compartments. This presents another opportunity compare transcriptional signatures of the ILC3, TCl and Prdm16+ cells captured in this data and extrapolate where they may be communicating with Tregs in vivo. Imaging of Prdm16 or another proxy marker alongside Tregs in these tissues would also help to contextualise the findings and uncover the niche in which proposed interactions between these cell types occur.

Version 1:

Reviewer comments:

Referee #1

(Remarks to the Author)

I thank the authors for their detailed responses to the original comments, particularly for the additional experiments using Prdm16 Δ ROR γ t mice, and for the more in-depth characterization of the novel APC. These changes have added significantly to the impact of the paper and increased the understanding of how these cells might contribute to Treg mediated oral tolerance to protein antigen.

Some issues remain:

- 1) In my opinion, the most important of these is that the novel APC should not be referred to as a "tolerogenic DC" as the authors now define them. First, it is now considered inappropriate to use a functional term to label DC as if they were a distinct lineage, as it is clear that DC of all lineages can exist in a functional spectrum. More to the point, the current work does not show direct evidence for the APC belonging to the DC lineage at all. Certainly they share some features of conventional DC and the new, epigenetic studies are interesting. However many of the phenotypic properties are not specific to DC (eg CD11c, MHCII, CSF1R) and their expression by the new APC was heterogeneous. To identify the APC as bona fide DC, it will be necessary to show eg that they are derived from a pre-cDC in a Flt3L dependent manner in vivo and in vitro. In the absence of this evidence, the term "tolerogenic DC" should be omitted.
- 2) While the authors compare their APC with ILC3 in some detail, the reliance on "tolerogenic DC" as a definition has prevented a full discussion of how the current cell might be related to the other, recently described Ror γ t+ APC populations such as Thetis/Janus cells. For instance, despite the added transcriptional data shown in Supplemental Table 1, this material is extremely complex, making it difficult still to see information on the absolute levels of expression of eg avb8 integrin, AIRE
- 3) In this respect, how do the authors see the Nrg1_Pos cells with such a mixed transcriptional profile fitting into the overall picture and their relationship to the Prdm16+ APC?
- 4) I found it very difficult to understand the first paragraph of the section "Characterization of Prdm16+ APCs as tolerogenic DC", especially the second sentence. The authors might be able to clarify what was done here to integrate the different databases.
- 5) As noted above, the new phenotyping data provided in Figures 2 and Extended Data 6 often describe heterogeneous expression of non-specific markers or morphology and conclude that the APC are "similar to DC" or "closely resembling DC" as opposed to proving they are a bona fide DC. The interpretation of these data should be more cautious.
- 6) The new Prdm16 Δ ROR γ t mice are interesting and potentially important. While I accept it seems unlikely, can the authors be sure that this approach might target other, non-immune cells expressing Ror γ t and Prdm16?
- 7) Is the enteropathy seen in older Δ +7kb mice accompanied by infiltration with eosinophils?
- 8) While interesting, the additional information provided for OTII transfer experiments shown in Figure 3 and Extended Data 8 seems incomplete for concluding there is a Th2-specific process in the absence of the Prdm16+ APC. For instance, there are still virtually no Th2 cells in LP of the Prdm16 Δ ROR γ t mice on d12, with there appearing to be a much greater effect on Th17 and Th1 cells. Furthermore, no data are shown for day5 in either MLN or LP of the Prdm16 Δ ROR γ t mice, which would be needed to make a direct comparison with the Δ +7kb and MHCII Δ ROR γ t mice.
- 9) Although I appreciate the difficulties in studying the APC activity of the Prdm16+ cells in vitro and the complications raised by the lack of Prdm16-cre mice, it remains the case that the work does not directly prove a role for these cells as APC in vivo. Using the Prdm16 Δ ROR γ t mice in an allergy tolerance experiment might help to some extent, but otherwise, it would be appropriate to acknowledge this limitation more explicitly.
- 10) The associations between Prdm16 SNPs and human disease are intriguing, but the nature of the human cells that are compared with the mouse APC remains somewhat unclear. As well as the data in the Supplemental Table being extremely dense, many of the individual genes highlighted as being expressed by the human Prdm16+ cells in the intestinal or LNs are not particularly typical of DC - eg Prox1, IL3R, Kit, PIGR, IDO2, DLGAP1, MACC1.

Referee #2

(Remarks to the Author)

The revised manuscript by Lihui Fu et al is much improved compared to the original. The authors further consolidate their identification of Prdm16+ tolerogenic DC cells (tDC) by employing additional tools include a Rorc-Cre x Prdm16flox mice, a

food allergy testing paradigm of oral tolerance and newly added multiome scRNA-seq and scATAC-seq analyses of mouse tDC and human scRNAseq data sets. In particular, I find the depletion of tDC upon the inactivation of Prdm16 and the resulting loss of (OTII) pTreg to be an incisive intervention. Overall the authors have satisfactorily answered most of my queries. I have the following comments:

1) The authors would have been better served to have carried out the experiments in Figure 4 (allergen specific oral tolerance induction) on the Rorc-Cre x Prdm16flox mice. This is after all the subset (tDC) under investigation in the manuscript. I would advocate at least repeating the food allergy study using the Rorc-Cre x Prdm16flox mice.

2) Prdm16+ tDCs appear to be sustained in numbers during development albeit with sharply reduced frequencies over time (Ext. Fig. 6c). It is unclear on which gut tissues these studies were carried out (which intestinal regions). Are there developmentally-regulated regional differences in tDC frequencies/numbers? While tDCs appear closely related to the previously described Thetis cells by the Chrysothemis Brown group, the latter were described as developmentally regulated. Is this simply a dilution effect (same numbers, lower frequencies?).

Referee #3

(Remarks to the Author)

Fu et al have made significant revisions to their manuscript. The additional granularity of analysis of their single cell RNA seq and by flow cytometry adds important information to address the identity of these cells and to allow comparison with other RORgt+ APC populations. Moreover, the generation of new mice targeting Prdm16 in RORgt+ cells (coupled with alternative CD11c Cre targeting of RORgt) together appears to support the conclusion that the pTreg-inducing capacity is contained within this population of cells, which the authors term "tolerogenic DCs". Nonetheless, there remain some points that need further analysis and clarification, while further attempts to reconcile their data with prior studies in the text is critical to maximise understanding of this complex and rapidly developing area for a non-expert readership.

- In new data the authors demonstrate that deletion of Prdm16 in RORgt-expressing cells phenocopies the changes in Treg reported previously, and in both CD11c-intrinsic RORgt-deletion and RORc 7kb mice. This is the most definitive proof in the manuscript that this population of cells may directly promote intestinal Tregs. One important caveat of this experiment and its interpretation is the authors state "Prdm16+ APCs were barely detectable". While at face value this appears true it could be misleading based on the data currently provided as the authors utilize the targeted gene product (Prdm16) as the only phenotypic marker to identify these cells, hence this is a self-fulfilling prophecy and makes an assumption that Prdm16 is essential for the development or persistence of these cells. To distinguish between the successful deletion of Prdm16 in these cells and loss of this population per se the authors must utilize alternative gating strategies (building on gating and marker combinations shown in Figure 2f+g). For example, can they demonstrate a loss of RORgt+ MHCII+ cells within the CXCR6- IL-7R- without using Prdm16 directly to quantify, that would further support a "loss" of these cells.

- The authors should make further attempts to integrate their findings within the existing framework of nomenclature for this emerging population of RORgt+ APCs in the results and discussion. The proliferation of "new cell types" makes it difficult for non-expert readers to follow the latest developments in this field and can "muddy the waters". As a logical nomenclature has recently been proposed (Abramson et al Nat Rev Imm 2024), I would suggest applying this where possible and discussing the findings in the framework of this discussion (i.e. if Prdm16+ and Nrg1+ cells express AIRE, should they not be referred to as eTACs, if not further explanation is needed as to why they don't fit this definition, see also next point below). Additionally, they should include discussion of the recently published study by Canesso et al in Science and potentially also reconcile/compare their findings with parallel studies in pre-print (e.g. Parisotto & Brown).

- On similar lines - one thing that remains unclear is how heterogeneous Prdm16+ RORgt+ APCs are. The authors refer to them as "tolerogenic DCs", yet they do not express CD11b and only a small proportion express CD11c (despite presumed effective CD11c-Cre targeting) as per their flow cytometry. In addition, this cluster exhibits AIRE expression, and to some degree CCR6 suggestive of the previously described phenotype of eTACs (and/or TC1 as per Brown et al), rather than DC-like RORgt+ APC (which were reported to lack both of these). This raises the possibility that this cluster is heterogeneous and that Prdm16 marks both an eTAC and a RORgt+ DC population. This should be investigated with granularity by further sub-clustering of the scRNA seq data set and/or further flow cytometry analysis using previously reported markers defining these different subsets (as requested in first review). It might be expected that combinations of CD11c, AIRE and CCR6 as well as other novel TC/eTAC markers reported would split this population into two discrete cell types, and that Prdm16 may be a common signature for multiple non-ILC RORgt+ APCs.

- Another interesting observation in the manuscript that could help reconcile confusion in the field is the observation that d+7KB, 11c-intrinsic RORgt deficient mice and RORgt-intrinsic Prdm16 deficient mice all failed to exhibit increased Th17 responses or spontaneous intestinal inflammation in the absence of H. hepaticus colonization (Extended Data Fig 7). While this may indicate the endogenous microflora is not sufficient to drive these effector responses, it is interesting that some of the initial studies that described MHCII+ ILC3 regulation of CD4+ T cells proposed direct suppression of microbiota-responsive Th17 by MHCII+ ILC3 in vitro and ex vivo and reported that restricted expression of MHCII via RORc Cre, but not CD11c Cre, suppressed Th17 responses without effects on Treg (Hepworth et al Nature 2013, Science 2015). This suggests one integrated model that might reconcile these two sets of studies whereby MHCII+ ILC3 can suppress activated microbiota-elicited Th17 effectors, and other DC-like or eTAC RORgt+ APC populations preferentially induce pTreg. This may warrant further discussion, and as per my other comments would help to try and reconcile and integrate the findings of these different studies for the readership.

- The issue raised by multiple reviewers that direct evidence of antigen presentation by Prdm16+ APCs, still holds. In the revisions the authors use MHCII-dRORc mice to directly address antigen-presentation, however these experiments still suffer from the caveat that these mice also hit ILC3. Nonetheless, as they mention isolation of Prdm16+ cells was technically challenging to address this question *ex vivo*, and a Cre model to exclusively target these cells is not currently available this is understandable, but one that should be acknowledged as a limitation of the study.

- Related to this point the authors should demonstrate whether other aspects of Prdm16+ APC "tolerogenic function" is lost in the +7kb mouse, such as Itgb8 expression and ability to cleave TGF-beta which could potentially be as important mechanistically as direct antigen presentation.

- Aspects of the initial ILC3 data require further explanation to the reader, as there is potential for confusion, and in some places interpretations and conclusions should be reworded or tempered. For example, the 7kb deletion in the Rorc locus leads to a "loss" of CCR6+ ILC3, yet this is not recapitulated in the scRNA seq data set. As mentioned by the authors this is likely because deletion of RORgt in mature ILC3 results in loss of phenotype, without loss of cells per se (e.g. as reported in the Withers and Fiancette papers cited), but to help the non-expert reader reconcile these observations this would be useful to also mention in the text of the results when discussing Figure 2c+d.

- Conversely, the authors use the observation that ILC3 are not lost in scRNA seq in Figure 2 to suggest Prdm16+ population are more likely responsible for tolerogenic function. However, one key piece of information lacking in the first figures of the manuscript is whether +7Kb mutants lose MHCII expression on dysfunctional ILC3, or on residual Prdm16+ cells. Given ILC3 are seemingly retained but lose phenotypic markers, including MHCII as per previous publications, the authors should directly address this point and temper their interpretation of the data at this early point in the manuscript. Indeed, the authors have now provide this data in Reviewer Figure 3 which shows MHCII expression is reduced on both subsets, thus it is critical that this data be included in the manuscript (along with representative histograms/flow plots) to address this important point.

- Further to this point, in response to my initial comment the authors argue in their rebuttal that "there was little effect on ILC3 or LTi function" and that data from the d+7kb mice are "most consistent with a [...] role for RORgt in the function of mature tDC, but not ILC3". I'm not sure how these points can be supported given the loss of phenotype and MHCII (and thus potential APC function) in ILC3 in this model and care should be taken not to make sweeping statements that cannot be supported by individual models/data sets. In line with this I would generally ask that the authors temper some of the wording and interpretations in earlier parts of the study that somewhat pre-empt conclusions that can only be fully supported when considering the totality of the data and models in the manuscript e.g. page 7 regarding prd16+ APCs "suggest that these cells have tolerogenic function in response to the microbiota" – at this point in the manuscript no direct evidence for this has been provided, only correlative predictions.

- The text addressing links to GWAS studies in the results section would be more appropriate for the discussion, suggest to move.

Version 2:

Reviewer comments:

Referee #1

(Remarks to the Author)

The authors have provided some new data and extensive discussion of the reviewers' comments. Unfortunately however, the revised manuscript itself still fails to address some of the most important issues raised about how the authors describe the nature of the Rorgt+ APC and its place in context of other recently identified APC subsets. To me, the most critical of these issues remains the definitive identification of the current cells as "tolerogenic DC". As the authors say, DC heterogeneity is a contentious area, with much of the confusion having arisen because of imprecise approaches and lack of information on how populations identified by individual groups might relate to each other. By ascribing a new name to yet another subset without full characterization and lack of appropriate contextualization, the current work risks adding to, rather than clarifying this confusion. I strongly believe that the use of this nomenclature based on the current work is inappropriate and very unhelpful.

1) While the authors show convincingly that their APC are unlikely to be ILC3, as I noted in the previous reviews, their identification as DC remains much less definitive than the manuscript makes out; it is based mostly on assumptions from overlapping/inconsistent markers. Too many phrases such as "by process of elimination", "closely resembling" etc are not sufficiently precise for unravelling such a complicated issue. Interestingly the recent PNAS paper by the Schraml group now referred to by the authors points out how this could be done, using eg Flt3L and IL2Rg KO mice, expression of DC-subset specific markers at protein level. Here it should be noted that the Schraml paper finds the apparently analogous population of Prdm16+Rorgt+ APC to express SIRPa, which seems to be at very low levels in the current work, while unlike here, CD11c expression appears stable with age in the other work. In contrast, Rorgt protein expression is found to decrease with age as assessed by the reporter gene tracking used by Schraml. Although it would not necessarily be appropriate for the current authors to carry out additional experiments, consideration of these apparent discrepancies is needed and it is essential that the resulting gaps in the current work are acknowledged, with the results interpreted less conclusively.

2) Of equal importance is the fact that many "tolerogenic" populations of DC have been proposed over the years, without any having stood the test of time. In each case, it was been shown that the relevant cell was plastic and could respond flexibly to environmental conditions, with more, rather than less confusion resulting because of attempts to characterize the hypothetical cell. Again, the Schraml work is instructive here, where the Prdm16+Rorgt+ APC was capable of inducing both Treg and effector T cells depending on the context.

3) While I appreciate there may be space limitations, there is still too little specific discussion of how the current APC population may fit in context of the other Rorgt+ APC which have been described recently. Readers will expect to see this and the novel findings on induction of tolerance deserve better contextualization.

Referee #2

(Remarks to the Author)

The authors have answered all my queries satisfactorily.

Referee #3

(Remarks to the Author)

The authors have provided a response to concerns raised in the previous round of review. In some cases my questions remain, in particular as to the addition of more granularity and direct mechanistic links between Prdm16 deletion and loss of Treg induction. Specifically, it remains unclear if deletion of Prdm1 in RORgt+ cells drives a loss of this cell population and its antigen presenting capacity, or rather dysregulates the regulatory capacity of a non-ILC3 RORgt+ APC cell type. Similarly, the relationship of these cells to previously reported TC/eTAC populations is not fully explored, despite the presence of published and validated flow cytometry markers and gating strategies. This makes it difficult to understand the degree of conceptual advance. While the identification of a gene associated with RORgt+ APCs is interesting it remains unclear the extent to which it is a common feature of some or all previously reported RORgt+ APC cell type(s), moreover the biological function and relevance of Prdm16 indicating cell-intrinsic regulatory functions (versus being a convenient marker of a subset of RORgt+ APCs) is not fully explored.

Furthermore, I agree with Reviewer 1 that the terminology "tolerogenic DC" may not be warranted based on current evidence, and more generally there are a number of speculative statements and interpretations I feel should be avoided, including several new statements speculating about lineage relationships (Page 8, Line 245) , potential loss of tolerogenic function without direct evidence (Page 9, line 265)etc.

Nonetheless, the manuscript provides convincing evidence that the Treg inducing capacity of RORgt+ APCs is contained within a population of cells expressing Prdm16 in mice, with evidence of an analogous population in humans. The additional description of the +7kb regulatory region of Rorc also provides new insights into the regulation of this transcription factor and RORgt-expressing lymphocyte populations, while the authors demonstrate across mouse models that RORgt+ APC (requiring Prdm16+ cells or their functions) induce Treg in a model of oral tolerance.

Referees' comments:

Referee #1 (Remarks to the Author):

In this paper, Fu et al explore the molecular characteristics of the recently described population of unusual Rorgt⁺ cells found in mesenteric lymph node, that appear to play a crucial role in driving the generation of FoxP3⁺ Tregs in response to the intestinal microbiota. As has been shown previously, mice lacking the Rorgt⁺ cells preferentially generate Th1 and Th17 responses when colonized with *Helicobacter hepaticus*, although this dysregulation becomes Th2-polarized when Hh is not present. Importantly, these effects are not intrinsic to CD4⁺ T cells. A novel finding is the identification of a region +7kB from the Rorgt locus that is important in controlling the development of these cells and show that deleting this region prevents the induction of FoxP3⁺ Tregs and the tolerance of Th2-driven, allergic lung disease that occur after oral administration of ovalbumin as a dietary antigen. In addition, using selective expression of Prdm16 as a marker of the Rorgt⁺ population, the authors present evidence that similar cells may be present in human MLN. Many of these findings are novel and represent important advances in our understanding of this novel cell population. However a number of important issues are not addressed fully:

1) While the authors refer to the Rorgt⁺ cells as "APC" and this was shown in some of the previous papers, none of the current experiments show a direct role for them presenting antigen to T cells *in vivo* or *in vitro*. Thus it remains feasible that the differences in T cell polarization could reflect effects of the Rorgt⁺ cells in shaping the environment. The oral tolerance model provides a potential system for doing this.

Thank you for pointing out this important omission. To address the role of RORγt⁺ cells as antigen-presenting cells (APCs) for oral antigen-induced pTreg cells, we have performed additional experiments:

- 1) We transferred naïve OT-II CD4⁺ T cells into OVA-treated *MHCII^{ΔRORγt} (Rorc(t)^{cre}I-AB^{fl/fl})* mice and analyzed their differentiation. By day 12 post-transfer, very few OT-II pTregs were detected in the SILP of *MHCII^{ΔRORγt}* mice (Fig. 3f), indicating that antigen presentation by RORγt⁺ cells is necessary for food antigen-specific pTreg differentiation.
- 2) In an OVA oral tolerance model, OVA-pre-fed *MHCII^{ΔRORγt}* mice exhibited increases in lung eosinophils and Th2 cells, comparable to non-tolerized mutants (Fig. 4h and Extended Data Fig. 9k), confirming that antigen presentation by RORγt⁺ cells is essential for oral tolerance.

2) Related to this, the work does not attempt to isolate the Rorgt⁺ cells from MLN, which would allow antigen presenting cell activity to be addressed. In addition, it would be important to see absolute numbers and surface phenotype of the cells in the different models, especially in the new Δ+7kb mouse.

We have attempted to isolate the tolerogenic APCs for *in vitro* priming experiments, but that has proven to be very challenging due to limited cell numbers and difficulty avoiding contamination with cDCs. However, we developed a flow cytometry panel that allows us to quantify and better phenotype the Prdm16⁺ cells. Gating on CD45⁺Ly6G⁻B220⁻TCRγδ⁻TCRβ⁻MHCII⁺RORγt⁺CXCR6⁻Prdm16^{high} cells

indicates that these cells express CD40 but lack ILC3 markers IL-7R and CD90 (Fig. 2f,g and Extended Data Fig. 6a). Our further flow cytometry analysis showed that the absolute numbers of both these Prdm16⁺ APCs and MHCII⁺ ILC3 were reduced in the mLN of $\Delta+7kb$ mice. However, in *Rorc(t)^{ACD11c}* mice, MHCII⁺ ILC3 remained unchanged, whereas Prdm16⁺ APCs were depleted (Fig. 2h,i and Extended Data Fig. 6d,e). Similarly, in *Prdm16^{ΔRORγt}* mice, which fail to generate pTregs specific to both microbiota and food antigens (Fig. 3a,d), MHCII⁺ ILC3 numbers were unchanged in both mLN and intestines, while Prdm16⁺ APCs were barely detectable (Extended Data Fig. 7a,b). Based on these and additional experiments, we have designated this RORγt-dependent, Prdm16-expressing APC population as tolerogenic dendritic cells (tDC).

Comparative analyses with cDC and MHCII⁺ ILC3 confirmed that tDC and MHCII⁺ ILC3 are distinct populations, with tDC showing greater phenotypic similarity to cDC (Fig. 2f,g and Extended Data Fig. 6a,b). Multiome scRNA-seq and scATAC-seq analyses revealed that the chromatin accessibility profile of tDC closely resembles that of classical DC populations (cDC1, cDC2A, and cDC2B) but is distinct from ILC3. For example, although tDC do not transcribe *Clec9a*, the *Clec9a* locus remains accessible, showing peaks shared with cDC but absent in ILC3 (Fig. 2e). Similar patterns were observed for *Csf1r*, *Flt3*, *Clecl0a*, *Sirpa*, *Itgax*, *Itgam*, and *Itgae* loci (Extended Data Fig. 5). In contrast, *Cxcr6* exhibited both gene expression and chromatin accessibility in ILC3, but was negative in tDC and cDC populations.

3) Previous work on MLN and Peyer's patches has suggested that the presence of Rorγt⁺ APC is highly age dependent, with these cells appearing to be restricted to the neonatal period, the time at which induction of oral tolerance is likely to be most relevant. While I note that the scRNAseq experiments were conducted with 3 week old MLN, the other work, including that on oral tolerance, seems to have used older, adult mice. This issue needs commented on and if possible, additional experiments carried out to identify when the Rorγt⁺ cells appear in MLN in the authors' hands and how this influences their functions.

Thank you for highlighting this important point. Our analysis of tDC abundance across a developmental time frame (using our newly-implemented flow cytometry panel), from 1-week-old to 12-week-old mice, revealed a slight increase in their numbers with age, followed by stabilization in adulthood (Extended Data Fig. 6c). These findings indicate that tDC are not restricted to the neonatal period but are maintained into adulthood, even as their proportion relative to other CD45⁺ cells may progressively decrease.

In our manuscript, the experiments involving microbiota- and food antigen-specific pTreg induction, as well as oral tolerance were performed using adult mice, consistent with previous reports demonstrating the ability of adult mice to induce tolerance to gut microbiota and food antigens (Esterházy et al., 2016; Kedmi et al., 2022; Xu et al., 2018). These results suggest that tDC maintain their functional persistence into adulthood, supporting their role in promoting tolerance beyond the early developmental period.

4) As the exact nature of this novel population has been controversial, it is disappointing that the authors do not use the opportunity of having single cell transcriptional data to make some detailed comparisons with conventional dendritic cells and ILC.

In our revised manuscript, we have included detailed comparisons between tDC, cDC, and ILC using

single-cell transcriptional data. These analyses are now presented in Fig. 2b and Extended Data Fig. 4a-c.

Additionally, we performed further analyses to characterize these cell populations, including chromatin accessibility profiling by multiome scRNA-seq/scATAC-seq (Fig. 2e and Extended Data Fig. 5), as well as surface and transcription factor marker analysis by flow cytometry (Fig. 2f,g and Extended Data Fig. 6a,b). These results are included in the revised manuscript to provide a comprehensive comparison.

5) No information or images are presented on whether there is intestinal inflammation or pathology in any of the deletion models. This is crucial because the previous evidence indicates that the mice lacking antigen presentation or b8 integrin expression on Ror γ ^t APC develop microbiota-driven inflammation, while other immunological KO models that lead to Th2 polarization can also develop intestinal pathology. The presence of such inflammation could be a confounding factor in the loss of oral tolerance.

Thank you for raising this important point. We examined both 8-week-old and 40-week-old Δ +7kb mice for signs of intestinal inflammation and pathology.

In 8-week-old Δ +7kb mice, no overt intestinal inflammation or type 2 gastrointestinal pathology was observed. Notably, this is the age at which most of our microbiota tolerance and oral tolerance experiments were conducted, indicating that the loss of oral tolerance in these mice is not confounded by underlying intestinal inflammation (Extended Data Fig. 7i-l).

In contrast, while no overt intestinal inflammation was detected in 40-week-old Δ +7kb mice by hematoxylin and eosin staining, these older mice exhibited features indicative of spontaneous type 2 gastrointestinal pathology (Josefowicz et al., 2012; Schneider et al., 2018), including muscularis propria hypertrophy, increased small intestine length, and elevated serum total IgE levels (Extended Data Fig. 7i-l).

These findings collectively suggest that while Δ +7kb mice are free of confounding intestinal inflammation at 8 weeks of age, they may develop age-associated type 2 gastrointestinal pathology later in life. The presence of this pathology in older mice is also a key indication of the loss of food tolerance.

6) While interesting and clearly of importance for understanding the use of the Δ +7kb mice, the authors could consider omitting some of the additional information related to its expression in other cell types.

Thank you for the suggestion. However, we believe that retaining this information is important for providing a broader context on the Δ +7kb locus and its regulatory roles, which supports the interpretation of our findings. That said, we will condense the textual description if needed.

7) It is clear that the defect in Ror γ ^t cells is much more severe in the Rorc(t) Δ CD11c mice compared with the Δ +7kb mice and this warrants some comment. This is also where absolute numbers of cells would be useful.

Our additional flow cytometry analysis confirmed that the absolute numbers of both tDC and MHCII⁺ ILC3 were reduced in the mLN of Δ +7kb mice, with a reduction of approximately 50–70%. In contrast,

Rorc(t)^{ΔCD11c} mice exhibited a more severe phenotype: MHCII⁺ ILC3 remained unchanged, while tDC were barely detectable (Fig. 2h,i and Extended Data Fig. 6d,e).

These findings highlight that the Δ+7kb deletion, which downregulates RORγt expression, may partially impair the development of tDC, whereas *Rorc(t)^{ΔCD11c}* mice may exhibit a profound defect in tDC development due to the deletion of RORγt. Notably, although residual tDC are present in Δ+7kb mice, these cells appear to be non-functional. This is supported by the results of the Hh7-2 and OT-II transfer experiments, which showed that Δ+7kb mice generated scarcely any Hh7-2 pTregs (Fig. 1c) or OT-II pTregs (Fig. 3e).

8) The methods section indicates that the experiments used age and sex-matched controls. However it is now accepted that littermate controls are needed in all such studies, particularly those in which there a microbiota-dependent phenotype.

We agree with the critical importance of littermate controls, particularly in studies involving microbiota-dependent phenotypes. For experiments requiring smaller sample sizes (e.g., two groups with n = 4 each), we ensured the use of strict mutant and littermate controls. However, for experiments with larger sample size requirements (e.g., total n = 18), it is challenging to obtain such a large number of perfectly matched littermate controls. In these cases, we combined mutant and littermate controls from different cages to perform the experiments.

9) In contrast to the authors' conclusions about the deletion of Rorgt⁺ cells selectively driving Th2 expansion, Figure 2 and Extended Data Figure 5 seem to show clear increases in both Th1 and Th17 populations in the small and large intestinal lamina propria, suggesting the immune dysregulation is more general. In this respect, if there is a Th2-specific polarization in the lung, should the PBS-treated Δ+7kb mice not show more susceptibility to the allergic inflammation here?

Thank you for the comment. We agree that the deletion of tDC does not selectively drive Th2 expansion but rather correlates with intensified effector Th cell responses more generally. The specific Th cell subset favored, however, depends on the local tissue environment, as we have noted in the manuscript.

Regarding the Th2-specific polarization in the lung, the observed differences in OVA-specific Th2 responses are driven by differences in OVA-specific Treg induction. In non-tolerized conditions, intraperitoneal OVA + alum strongly promotes OVA-specific Th2 differentiation, while inducing only minimal OVA-specific Tregs. This is supported by our OVA:I-A^b tetramer staining results, which show that OVA:I-A^b tetramer⁺ Tregs are present at very low levels in both non-tolerized control and Δ+7kb mice, with no significant differences between the two (Fig. 4e,f). As a result, non-tolerized Δ+7kb mice do not exhibit increased OVA-specific Th2 polarization compared to non-tolerized controls (Fig. 4e,f).

10) The selective expression of Prdm16 by the Rorgt⁺ cells is interesting and could be useful in future studies of these cells. Do the authors have any information on what this molecule may be doing in this population and what its expression signifies?

This is a very important point that is central to our conclusion that the Prdm16⁺ cells are the relevant tolerogenic APCs. In the revised manuscript, we have conditionally inactivated Prdm16 in RORγt-expressing cells (*Prdm16^{ΔRORγt}* mice) and assessed the fate of adoptively transferred

antigen-specific T cells. In these mice, MHCII⁺ ILC3 numbers were unchanged in the mLN and intestines, while tDC were depleted (Extended Data Fig. 7a,b). Two weeks after transfer of *Hh*-specific Hh7-2 CD4 T cells, Hh7-2 pTreg differentiation was abolished in the LILP and mLN of *Hh*-colonized *Prdm16*^{ΔRORγt} mice, accompanied by an increase in inflammatory Th17/Th1 cells (Fig. 3a and Extended Data Fig. 7c,d). At twelve days post-transfer of OVA-specific OT-II CD4 T cells into OVA-fed mice, there were few OT-II pTregs in the SILP of *Prdm16*^{ΔRORγt} mice, and, instead, the OVA-specific T cells displayed Th2, Th17, Th1 and Tfh phenotypes (Fig. 3d and Extended Data Fig. 8c). These results indicate that *Prdm16* is critical for tDC development and function. However, the precise mechanisms by which *Prdm16* regulates tDC remain to be elucidated. We have acknowledged this in the discussion section of the manuscript and plan to address it in future studies.

11) While interesting, I have concerns about the interpretation of the human data showing an analogous population. For instance, there seems to be wide variation in the levels of expression of the 29 DEGs overlapping in mouse and human and one wonders what an equivalent volcano plot for non-overlapping DEGs would show. It is notable that AIRE is not expressed by the human cells, while there is no mention of other signature genes of the mouse cells such as *Itgb8*. Given the possible age-dependence of the cells in mice, a 22 year old human subject may also not be optimal.

We share the Reviewer's concern regarding the generalizability of our initial human data. Therefore, we have proceeded to add public datasets from 6 intestinal lamina propria resections, as well as 9 tonsillectomies, where we additionally observed distinct populations of tDC (Figure 5d,f). This increased our total to 67,638 human cells, of which 988 we identify as human tDC. While our first manuscript submission utilized a default threshold of $\text{Log}_2(\text{Fold Change}) = 0.1$, our substantially expanded revised dataset allows us to now set a much more stringent $\text{Log}_2(\text{Fold Change}) = 3.2$ to identify 141 human DEGs, and observe that 8 of those are conserved with mouse (Extended Data Fig. 10b,c). This approach identified DEGs overlapping across species for cDC1, with expected results including *XCRI*, *CLEC9A*, and *CADMI*, as well as expected results for conserved ILC3 genes including *IL23R*, *IL22*, and *RORC*. Therefore, we also now include a Supplemental Table detailing all tDC DEGs, their Fold Change, and their p values, such that the reader can now find *ITGB8* as well as look for other genes of interest. Finally, our data highlight a much less age-dependent preponderance of tDC in mouse (Extended Data Fig 6c), and our additional human datasets identify the analogous cells across multiple ages as well.

Referee #2 (Remarks to the Author):

In this article the authors attempted to characterize the role of the APC subset implicated in driving tolerance to food antigens and Helicobacter. Their previous studies along with others suggested that ROR γ t expressing APCs regulated tolerance to the gut microbiota but the nature of tolerogenic APC subset remained unclear. In the present study the authors assessed the role of ROR γ t expression in APCs in inducing tolerance to food antigens. They identified that ROR γ t expression in CD11c cells was critical in the induction of tolerance to H. hepaticus-specific T cells. Further by analyzing the RORc locus among different ROR γ t expressing cells they identify a +6 Kb and +7kb region that differentially regulated Th17, gdT17 and ILC3 responses. Transgenic mice lacking the +6kb Rorc region was associated with defective Th17 response but an intact gdT17 and ILC3 functions. In contrast mice lacking the +7kb region were associated with normal Th17 response but a defect in the ROR γ t expression in ILC3 and gdT17 cells. The authors thus assume the mice lacking +7kb region to be associated with a defective ROR γ t expression only in the innate but not the adaptive cells. Using these mice, they show that the ILC3s function were normal in a model of C. rodentium infection and therefore any defects with respect to T cells would be due to other ROR γ t APCs and not due to ILC3. Baseline analysis of the 7Kb Tg mice indicated a defect in the ROR γ t+ Treg cells with dysregulated Th2 responses. Notably assessment of tolerance induction to ovalbumin was defective in the 7kb Tg mice and in a mouse model of allergic lung inflammation failure to induce tolerance was associated with heightened eosinophilia and lung inflammation. scRNA seq from WT and Δ 7Kb Tg mice identified a cluster of PRDM16 expressing APCs to be defective in the Tg mice. Lastly the authors demonstrate from a single human sample that the presence of PRDM16 expressing APCs is conserved in humans. Overall, the authors do not provide any direct evidence for the APC subset that is regulating oral tolerance. Based on their reliance on a Tg mouse that assumes defects in Ror γ t APC without demonstrating the defect and their failure to directly demonstrate the ROR γ t APC subset governing tolerance, the manuscript in its current version does not in my opinion achieve the priority to be published in Nature. This can change if the authors can demonstrate and provide direct evidence and underpin which ROR γ t subset (Thetis cells or ROR γ t+ DCs) is imparting tolerance to food antigens.

We thank the Referee for this fair assessment regarding the identity of the tolerogenic APC. In the revised paper, we have used a combination of genetic approaches and phenotypic analyses to clearly demonstrate that the relevant ROR γ t subset responsible for both microbiota- and oral antigen-specific pTreg induction is a Prdm16⁺ cell that most closely resembles classical DC and is conserved between mouse and human.

Major points

1) While the authors have placed much effort in disproving a role for ILC3 in driving peripheral tolerance to helicobacter hepaticus, very little has been done to pin down the APC subset involved in oral tolerance. The heightened emphasis on Helicobacter hepaticus specific tolerance and the role ILC3 detracts heavily from the major objective of the article, namely the identity and function of a specific ROR γ t APCs subset in peripheral oral tolerance.

Thank you for pointing out this concern. To address this, we have reorganized the manuscript to better highlight the major objectives: identifying the precise identity of the ROR γ t-APCs that induce microbiota-specific pTregs and revealing their role in peripheral oral tolerance. Specifically, we have moved the original Figure 4 (with additional new experiments) to become the new Figure 2. This

reorganization directly supports the central focus of the study and ensures that the key findings are emphasized earlier in the manuscript.

2) The authors show that ROR γ t expressing CD11c APCs are the drivers of peripheral tolerance to the microbiota (this is somewhat of a redundant point from their previous story). Nevertheless, they use different ROR γ t expressing cells and perform ATACseq to assess the *Rorc* locus, allowing them to derive different Tg mice lacking the respective *Rorc* loci. Tg mice lacking the 7Kb *Rorc* region is associated with a defect in ROR γ t expression in ILC3 and gdT17 cells but uniquely the Th17 cells are normal. They assume that in these Tg mice the defect in ROR γ t is therefore restricted to the innate compartment and not in the adaptive system. They authors show that the ILC3 are functional in a *C. rodentium* mouse model of infection. However, the Tg mice lacking the 7Kb region are associated with a defect in ROR γ t Treg cell at baseline thus with a leap of faith they claim this is due to ROR γ t+ APCs and not ILC3. This is premature assumption, as just because ILC3s are impacted does not mean that other ROR γ t+ innate cells are also defective. The experimental evidence demonstrated does not convince me of this. The authors need to show a direct defect in the ROR γ t+ APCs. For example, based on the evidence provided, the 7Kb Tg mice despite showing a defect in the ILC3 compartment, the ILC3 were indeed functional. Thus, why wouldn't the ROR γ t APCs be functional even if their frequencies are less which from their scRNA seq does not appear to be the case.

Thank you for the comment. We now provide extensive new data that helps solidify our conclusion that a tolerogenic DC-related cell rather than ILC3 induces Tregs. While in our 2022 study we demonstrated that the relevant APCs express ROR γ t during ontogeny, it remained unclear whether ROR γ t is directly required for these cells to develop or function, or if it merely serves as a marker. In this manuscript, we specifically demonstrate that ROR γ t is essential for the development and function of these APCs, providing novel mechanistic insight.

We have performed additional experiments to establish the precise identity of pTreg-inducing APCs, which allowed us to designate the ROR γ t-dependent, Prdm16-expressing/dependent APC population as tolerogenic dendritic cells (tDC).

- 1) Multiome scRNA-seq and scATAC-seq analyses revealed that the chromatin accessibility profile of tDC closely resembles that of classical DC populations (cDC1, cDC2A, and cDC2B) but is distinct from ILC3. For example, although tDC do not transcribe *Clec9a*, the *Clec9a* locus remains accessible, showing peaks shared with cDC but absent in ILC3 (Fig. 2e). Similar patterns were observed for *Csf1r*, *Flt3*, *Clecl0a*, *Sirpa*, *Itgax*, *Itgam*, and *Itgae* loci (Extended Data Fig. 5). In contrast, *Cxcr6* exhibited both gene expression and chromatin accessibility in ILC3, but was negative in tDC and cDC populations.
- 2) We identified tDC as CD45⁺Ly6G⁻B220⁻TCR γ δ ⁻TCR β ⁻MHCII⁺ROR γ t⁺CXCR6⁻Prdm16^{high} by flow cytometry. These cells express CD40, while lacking IL-7R and CD90. Comparative analyses with cDC and MHCII⁺ ILC3 confirmed that tDC and MHCII⁺ ILC3 are distinct populations, with tDC showing greater phenotypic similarity to cDC (Fig. 2f,g and Extended Data Fig. 6a,b).
- 3) The absolute numbers of both tDC and MHCII⁺ ILC3 were reduced in the mLN of Δ +7kb mice. However, in *Rorc(t) ^{Δ CD11c}* mice, MHCII⁺ ILC3 remained unchanged, whereas tDC were depleted

(Fig. 2h,i and Extended Data Fig. 6d,e). In OT-II transfer experiments, five days post-transfer, very few OT-II pTregs were detected in the mLNs of OVA-fed *Rorc(t)^{ΔCD11c}* mice (Fig. 3c and Extended Data Fig. 8b). In the oral tolerance model, OVA-pre-fed *Rorc(t)^{ΔCD11c}* mice exhibited comparable increases in lung eosinophils and Th2 cells to those observed in non-tolerized mutant mice, indicative of loss of tolerance (Fig. 4g and Extended Data Fig. 9j).

- 4) We conditionally inactivated Prdm16 in ROR γ t-expressing cells (*Prdm16^{ΔROR γ t}* mice) and assessed the fate of adoptively transferred antigen-specific T cells. In these mice, MHCII⁺ ILC3 numbers were unchanged in the mLN and intestines, while tDC were barely detectable (Extended Data Fig. 7a,b). Two weeks after transfer of Hh-specific Hh7-2 CD4 T cells, Hh7-2 pTreg differentiation was abolished in the LILP and mLN of Hh-colonized *Prdm16^{ΔROR γ t}* mice (Fig. 3a and Extended Data Fig. 7c,d). At twelve days post-transfer of OVA-specific OT-II CD4 T cells, there were few OT-II pTregs in the SILP of OVA-fed *Prdm16^{ΔROR γ t}* mice (Fig. 3d and Extended Data Fig. 8c). These results indicate that Prdm16 is critical for tDC development and function.

Collectively, these findings demonstrate that Prdm16- and ROR γ t-dependent tDC, but not ILC3, are the APC population inducing T cell-mediated tolerance to microbiota and food antigens.

- 3) While the authors demonstrate that the in vitro Th17 differentiation is functional in the 7Kb Tg mice, is this true also for in vitro ROR γ t Treg cell differentiation?

Thank you for the comment. We have addressed the relevant point using BM chimeric experiments (Extended Data Fig. 3d,e,h), which demonstrate that there is no intrinsic defect in ROR γ t⁺ Treg differentiation in Δ +7kb mice.

- 3) Can the authors provide the *Rorc* accessibility in Thetis cells, ROR γ t⁺ DCs or ROR γ t⁺PRDM16⁺ DCs. This analysis (at least from the Thetis cells and ROR γ t DCs should have been included in Fig1b). Do the authors have proof that the *Rorc* locus in Thetis cells/ROR γ t⁺DCs is dependent on the 7kb *Rorc* locus for their functionality? This analysis would have rendered some support in favor of using these mice.

For this revision, we have now further defined the tDC population with sc-ATAC-seq of mouse mLN (Fig 2e, Extended Data Fig. 5). Indeed, the green shaded range demarcating *Rorc(t)* +7kb CRE in Fig. 2e demonstrates prominent peaks in both tDC and ILC3, as expected.

- 4) In Fig 5e using scRNA-seq the authors do not find any difference between Thetis cells and many of the APC subsets. This brings us back to the identity of the ROR γ t APCs population(s) that are defective in the 7kb Tg mice. The only APC subset shown to be defective was the ROR γ t⁺ PRDM16⁺ APCs, and the authors hypothesize accordingly that it is this subset of APC that is the tolerogenic subset but again have no proof for this.

We acknowledge that the original scRNA-seq data alone are insufficient to definitively identify the tolerogenic APC subset. To address this, we have conducted additional experiments, which provide strong supporting evidence for our hypothesis. As described above, in mice with Prdm16 specifically deleted in ROR γ t-expressing cells, which led to a selective loss of tDC, we observed a complete failure

to generate pTregs in response to both microbiota and food antigens. These findings strongly indicate that tDC are the APCs inducing T cell-mediated tolerance to intestinal antigens.

5) Are there defects in the RORgt APCs in food allergic individuals? The availability of such human data would add weight to the observations described in this manuscript.

We have utilized the “NIH National Human Genome Research Institute Catalog of Human Genome-Wide Association Studies” to examine precisely this concept raised by the Reviewer. We include these findings as the final paragraph of our results:

“Genome-wide association studies have identified numerous single nucleotide polymorphisms (SNPs) of PRDM16 having strong statistical association with autoimmune and inflammatory diseases, including asthma in African Americans (mostly pediatric), allergic rhinitis (IgE-mediated inflammation of the upper airway), rheumatoid arthritis, and inflammatory bowel disease (IBD) in the Basque population.”

6) The use of allergic airway inflammation model to interrogate oral tolerance poses some challenges. Thus, in figure 3e and extended data fig6 g,h there is loss of both the OVA-specific RORgt⁺ and RORgt⁻ Treg cells in the Δ 7Kb mice, and the majority of the response is in the RORgt⁻ Treg cell population. So is this system interrogating the RORgt⁺ Treg cell response as a mechanism of oral tolerance? It does not seem so from the data. Perhaps additional staining in the gut can clarify this issue. Better still, a gut specific oral tolerance system would be also add clarity.

Thank you for the comment and suggestion. We would like to clarify that our study is not solely interrogating ROR γ t⁺ Treg cells as the mechanism of oral tolerance, but rather pTregs as a whole. pTregs express different additional transcription factors depending on the timing of antigen exposure and the tissue environment. In the OT-II transfer experiment, pTregs in wild-type mice included both ROR γ t⁻ and ROR γ t⁺ phenotypes, while GATA3⁺ pTregs were fewer (Fig. 3c-f). In contrast, in the oral tolerance protection from airway inflammation model, OVA-specific pTregs in the lungs of tolerized wild-type mice predominantly displayed a GATA3⁺ phenotype, with fewer ROR γ t⁺ pTregs (Fig. 4e,f and Extended Data Fig. 9h).

To further investigate whether tDC-induced pTregs are broadly required for tolerance to food antigens, we employed a food allergy model. Mice were either pre-fed or not fed OVA before sensitization intragastrically with OVA plus cholera toxin (CT), followed by an intraperitoneal OVA challenge (Fig. 4i). Systemic OVA challenge resulted in comparable anaphylactic responses, assessed by a rapid reduction in core body temperature and elevated serum OVA-specific IgE and IgG1 levels, in both tolerized and non-tolerized Δ +7kb mice, whereas tolerized wild-type mice were protected (Fig. 4j,k). Together with the data from the asthma model, these findings indicate that tDCs are crucial for the development of oral tolerance, underscoring their essential role in regulating immune responses to dietary antigens and preventing allergic responses.

7) I am also uncertain about Prdm16 as a mechanistic pathway in the defined APC subpopulation rather than just a marker for this subset. Humans with mutations in PRDM16 are not associated with tolerance breakdown but rather with cardiomyopathy. Do the authors have any data on the RORgt⁺ APC subpopulations in mice with either global or conditional Prdm16 deficiency?

While complete loss of PRDM16 (such as seen with 1p36 deletion syndrome), is associated with noncompaction dilated cardiomyopathy and other severe pathologies, such presentations are quickly lethal at an early age. Rather, we investigated for SNPs that arise in older pediatric and adult patients, and now include the following text:

“Genome-wide association studies have identified numerous single nucleotide polymorphisms (SNPs) of *PRDM16* having strong statistical association with autoimmune and inflammatory diseases, including asthma in African Americans (mostly pediatric), allergic rhinitis (IgE-mediated inflammation of the upper airway), rheumatoid arthritis, and inflammatory bowel disease (IBD) in the Basque population.”

As described above, in *Prdm16*^{ΔRORγt} mice, where tDC were selectively depleted but MHCII⁺ ILC3 remained intact, both Hh7-2 and OT-II pTreg differentiation were severely impaired. These results highlight Prdm16-dependent tDC as the APCs inducing T cell-mediated tolerance to intestinal antigens.

Other comments

1) The gating for figure 1a is not clear. The text claims that these are RORγt and T-bet-expressing Hh-specific T cells but the figure itself seems to show total/CD4⁺ T cell staining? Showing the gating strategy would help.

Thank you for the suggestion. We have clarified the gating strategy in the figure legend to ensure better understanding of the data.

2) In figure 2, the use of OT-II cells as means of interrogating OVA-specific responses is not sufficient. Endogenous OVA-specific T cell responses need to be measured by tetramer staining. If mice were orally sensitized with OVA do the authors observe tetramer staining in the RORγt⁺ pTreg cell population and is this staining decreased in the Δ7KB mice?

Thank you for the suggestion. We have performed tetramer staining to assess endogenous OVA-specific T cell responses in the SILP of OVA-treated mice. The results confirm the presence of OVA:I-A^b tetramer⁺ RORγt⁺ pTregs, and as suggested, this population is indeed reduced in the Δ+7kb mice compared to controls (Reviewer Fig. 1).

Reviewer Figure 1. Phenotype of endogenous OVA-specific pTregs in the SILP of OVA-treated control and Δ+7kb mice. Mice were provided with OVA-containing drinking water for 8 weeks starting immediately after weaning. The flow cytometry plots are gated on OVA:I-A^b tetramer⁺ CD4⁺ T cells.

3) If available, please provide the atopic status of the human subject used in the scRNA Seq analysis?

We clarified this with the transplant surgeons, and we now specify in the text that the donor had “no known history of atopy or chronic disease.”

Referee #3 (Remarks to the Author):

This manuscript by Fu et al builds on recent advances in the field of ROR γ t⁺ antigen-presenting cell biology, most notably a trio of papers published in 2022 – including one from the same group. This study reports several observations that expand this prior work to demonstrate i) that different enhancer elements within the *Rorc* locus regulate specific ROR γ t⁺-expressing cell populations, ii) that previously reported induction of ROR γ t⁺ FoxP3⁺ Treg is dependent upon an enhancer region +7kb from the transcription start site of *Rorc*, iii) that in this context the failure to appropriately induce Treg results in elevated Th2 cells, and a subsequent inability to mount oral tolerance to a model antigen, and iv) that a subset of ROR γ t⁺ APC marked by *Prdm16* transcript is altered in the above animal model.

While this is a rapidly emerging area, the aforementioned studies raised a number of open questions and confusion around the previously reported roles of other ROR γ t⁺ MHCII⁺ cells (e.g. ILC3 and eTACs), as well as the roles of classical dendritic cells in these processes. Thus, the observation that *Prdm16* transcript may mark a subset of ROR γ t⁺ APCs is a useful finding that could help clarify and consolidate the currently confusing nomenclature surrounding the expanding family of ROR γ t⁺ APC. As such further efforts to try and reconcile the current data with previous findings would be of benefit.

Overall, there are several interesting observations that largely build on existing concepts. However, while the first half of the manuscript is well developed the second half is largely built on assumptions and correlations and suffers from a lack of direct evidence and supporting data, as well as novel mechanistic insight that would provide a significant conceptual advance over the prior work in this area. As such the manuscript in its current form is not sufficiently mature and multiple interpretations and conclusions are not sufficiently justified by the data provided at this point.

We agree with the Referee that the expression of *Prdm16* in the relevant tolerogenic APC offered the opportunity to clarify the identity of the cell and point future research in the right direction. Using a combination of mouse genetic approaches, flow cytometry allowing us to track these cells, and single cell sequencing approaches, we have now shown that the tolerogenic APCs closely resemble cDCs and represent a distinct lineage dependent on both ROR γ t and *Prdm16* and present in both mouse and human. We have reorganized the manuscript to better highlight the major objective: identifying the precise identity of the ROR γ t-APC subset that induces microbiota-specific pTregs and revealing its role in peripheral oral tolerance. Specifically, we have moved the original Figure 4 (with additional new experiments) to become the new Figure 2. This reorganization directly supports the central focus of the study and ensures that the key findings are emphasized earlier in the manuscript.

Specific concerns are listed below:

Analysis of the d+7kb mice reveals preferential effects on ROR γ t-expression in ILC3 and a loss of ILC3 in this system (Extended Data Fig2e), most notably of the CD127⁺ CCR6⁺ “LTi-like ILC3” subset (Extended Data Fig 3a). In contrast no flow cytometry quantification and in depth analysis of other CD127⁻ ROR γ t⁺ APCs is provided (including both AIRE⁺ eTACs and/or *Prdm16*⁺ cells), nor is ROR γ t expression validated to also be perturbed by deletion of the +7kb enhancer in other ROR γ t⁺ APC populations. It is therefore seemingly counterintuitive that the early data demonstrating marked changes to ILC3 is dismissed later in the manuscript and differences attributed to the *Prdm16*⁺ population alone with little direct supporting evidence. This raises a number of issues that should be addressed:

- As non-ILC3 ROR γ t⁺ APCs have been reported to be CD127 negative (and CXCR6 negative) a quantification of the whole ROR γ t⁺ APC family in these mice with a much more granular flow panel is required to clarify the discrepancies between the reported changes in abundance of ILC3 and the later observation of decreased Prdm16⁺ cells (but not ILC3) via scRNA seq – as well as to validate the differences found in transcriptomic data.

This is a very important point that we have now resolved with a new flow cytometry panel, as well as additional single cell genomics analyses, allowing us to designate the relevant cell as a tolerogenic DC (tDC).

We can now readily identify the tolerogenic APCs as distinct from ILC3 and with a phenotype of CD45⁺Ly6G⁻B220⁻TCR γ δ ⁻TCR β ⁻MHCII⁺ROR γ t⁺CXCR6⁻Prdm16^{high} by flow cytometry. These cells express CD40, while lacking IL-7R and CD90. Comparative analyses with cDC and MHCII⁺ ILC3 confirmed that these cells and MHCII⁺ ILC3 are distinct populations (Fig. 2f,g and Extended Data Fig. 6a,b). Importantly, the tDC (along with pTregs) are lost in mice if ROR γ t is inactivated using CD11c-Cre, but there is no effect on ILC3 (Fig. 2h,i and Extended Data Fig. 6d,e).

- 1) Multiome scRNA-seq and scATAC-seq analyses revealed that the chromatin accessibility profile of tDC closely resembles that of classical DC populations (cDC1, cDC2A, and cDC2B) but is distinct from ILC3. For example, although tDC do not transcribe *Clec9a*, the *Clec9a* locus remains accessible, showing peaks shared with cDC but absent in ILC3 (Fig. 2e). Similar patterns were observed for *Csf1r*, *Flt3*, *Clecl0a*, *Sirpa*, *Itgax*, *Itgam*, and *Itgae* loci (Extended Data Fig. 5). In contrast, *Cxcr6* exhibited both gene expression and chromatin accessibility in ILC3, but was negative in tDC and cDC populations.
- 2) The results with the Δ +7kb mutant mice indicate that this enhancer is required for optimal expression of ROR γ t in both ILC3 and the tDC. Even though both cell populations are present in the mutant mice (with minimal reduction in ILC3 number and about 60% reduction in tDC), there was little effect on ILC3 or LTi function, but complete loss of pTreg inducing function. Our results are most consistent with a requirement for ROR γ t in early development of both cell types, but a role for ROR γ t in the function of mature tDC, but not ILC3. It is possible that the enhancer mutation affects ROR γ t expression in mature ILC3 and not in progenitors, where it is essential.

- Similarly, to support the authors conclusions and interpretations, the expression levels of ROR γ t in mice lacking specific enhancers should be shown for TC1/eTAC populations, and the Prdm16⁺ cell population identified later to provide important proof of concept and expand on Extended Data Fig 2.

In our revised manuscript, we presented data showing that the ROR γ t MFI of both tDC and MHCII⁺ ILC3 was decreased in the mLN of Δ +7kb mice (Extended Data Fig. 6d).

- The reliability of quantification of cell frequency/numbers by scRNAseq of pooled cells (essentially n=1, one independent experiment) is a matter of debate, thus its critical any changes in cell population frequency by this method are validated by other means (e.g. flow cytometry as above), particularly given the weight placed on the former observation for the interpretations and narrative of the manuscript.

Thank you for the suggestion. We have validated the changes in cell population frequency observed in scRNA-seq through flow cytometry. The absolute numbers of both tDC and MHCII⁺ ILC3 were decreased in the mLN of $\Delta+7\text{kb}$ mice. However, in *Rorc(t)^{ΔCD11c}* mice, MHCII⁺ ILC3 remained unchanged, whereas tDC were barely detectable (Fig. 2h,i and Extended Data Fig. 6d,e).

- The rationale for ruling out ILC3 as the Treg-inducing APC in this study is the observation that the capacity of ILC3 to produce IL-22 is retained in the $\Delta+7\text{kb}$ mice despite reduced ROR γ t expression (Extended Data Fig 3). The authors cite two previous papers (Withers et al Nat Med, Fiancette et al Nat Immunol) that have previously reported similar persistence of IL-22 in the absence of this transcription factor. However, Fiancette et al appear to show that deletion of ROR γ t in ILC3 does reduce MHCII levels (Extended Data Fig6 of that paper) and a number of other genes, despite IL-22 levels being retained. Additionally other auxillary molecules such as OX40L expressed by these cells have Treg promoting function, and could feasibly be perturbed by loss of ROR γ t. Thus, the authors cannot discount a reduced antigen presenting capacity of ILC3 in $\Delta+7\text{kb}$ mice that could explain their observations, and broader transcriptional dysregulation when ROR γ t expression is perturbed.

We agree that in the original submission we could not completely rule out a role for ILC3 in induction of Tregs. However, with new experiments showing that inactivation of Prdm16 in ROR γ t⁺ cells has no effect on ILC3 but eliminates pTreg activity (Fig. 3a,d and Extended Data Fig. 7a-d and 8c), we are able to confidently conclude that ILC3 do not have a role in the process.

- Do $\Delta+7\text{kb}$ retain normal cryptopatches and ILFs? Given the importance of these sites for Treg induction and/or maintenance, it's feasible that reductions in provision of lymphotoxin etc by ROR γ t⁺ cells could disrupt Treg induction indirectly via loss of lymphoid tissue architecture as opposed to direct antigen-presenting capacity. Interpretation of experiments in Figure 1 with *Rorc(t)gfp/gfp* are similarly confounded by the loss of lymphoid tissue formation, and this should be explicitly stated in the text to acknowledge this potential caveat.

Thank you for raising this point. To address the potential impact of lymphoid tissue architecture on Treg induction, we performed H&E staining of gut sections and confirmed that $\Delta+7\text{kb}$ mice retain normal isolated lymphoid follicles (ILFs) (Reviewer Fig. 2). This suggests that the architecture of lymphoid tissues, including ILFs, is intact and unlikely to contribute to the observed Treg induction defects.

As for the experiments involving *Rorc(t)^{gfp/gfp}* mice in Figure 1, we acknowledge that the loss of lymphoid tissue formation in *Rorc(t)^{gfp/gfp}* mice could introduce confounding factors. To ensure clarity and avoid overinterpretation, we have removed these data from the revised manuscript.

Reviewer Figure 2. Hematoxylin and eosin staining of the distal small intestine sections from 8-week-old and control and $\Delta+7\text{kb}$ mice. The arrows indicate an isolated lymphoid follicle (ILF).

- Conversely, the direct evidence to support the interpretation that Prdm16⁺ are the only RORγt⁺ APC required for appropriate induction of RORγt⁺ Tregs is drawn from almost exclusively from the single cell data, where capture of this cluster was lost in Δ+7kb mice – a purely correlative observation that does not provide direct evidence for the authors conclusions. Can Prdm16⁺ cells be isolated from wild type mouse using markers identified via the transcriptional signatures and demonstrated to induce RORγt⁺ Treg in an ex vivo co-culture system?

At present, we lack suitable surface markers to specifically and efficiently isolate Prdm16⁺ APCs from wild-type mice. However, we have performed additional experiments to establish the precise identity of pTreg-inducing APCs that we designate as tDC. We conditionally inactivated Prdm16 in RORγt-expressing cells (*Prdm16*^{ΔRORγt} mice) and assessed the fate of adoptively transferred antigen-specific T cells. In these mice, MHCII⁺ ILC3 numbers were unchanged in the mLN and intestines, while tDC were barely detectable (Extended Data Fig. 7a,b). Two weeks after transfer of *Hh*-specific Hh7-2 CD4 T cells, Hh7-2 pTreg differentiation was abolished in the LILP and mLN of *Hh*-colonized *Prdm16*^{ΔRORγt} mice (Fig. 3a and Extended Data Fig. 7c,d). At twelve days post-transfer of OVA-specific OT-II CD4 T cells, there were few OT-II pTregs in the SILP of OVA-fed *Prdm16*^{ΔRORγt} mice (Fig. 3d and Extended Data Fig. 8c). These findings demonstrate that Prdm16-dependent tDC are the APC population inducing T cell-mediated tolerance to intestinal antigens.

- How are Prdm16⁺ cells transcriptionally altered in Δ+7kb mice or CD11c mediated deletion of Rorc? Is there evidence that the cells that remain have a reduced antigen presenting capacity in vivo or ex vivo? Are there broader changes to the transcriptome caused by loss of RORγt activity or do these cells simply not develop and/or persist? Generally reporting of transcriptional differences between all RORγt⁺ subsets in the single cell data sets should be reported to help the reader understand the quality and magnitude of changes in these mouse models.

In *Rorc(t)*^{ΔCD11c} mice, tDC were barely detectable, making it impossible to directly assess transcriptional changes in these cells (Fig. 2i and Extended Data Fig. 6e). However, in Δ+7kb mice, we were able to analyze the remaining tDC. Flow cytometry revealed that both the absolute numbers and MHCII MFI of tDC were significantly reduced in the mLN (Fig. 2h and Extended Data Fig. 6d) (Reviewer Fig. 3), suggesting an impaired antigen-presenting capacity *in vivo*.

Reviewer Figure 3. MHCII MFI of tolerogenic DC and MHCII⁺ ILC3 in the mLN of control and Δ+7kb mice.

- Much of the evidence for RORγt⁺ APCs being the tolerogenic APC responsible for RORγt⁺ Tregs to date has come from comparisons between RORcCre driven deletion of H2-Ab1 in comparison with

RoraCre (hits ILC3 but not other ROR γ t APC subsets) and CD11cCre (which hits all ROR γ t+ APCs to differing degrees, and is a notoriously “leaky” Cre system). Thus, the identification of Prdm16 as a unique or unifying signature of this subset of ROR γ t+ APCs provides a unique opportunity to advance understanding of this emerging area. The authors should therefore demonstrate that targeting antigen presenting capacity with a Prdm16Cre, or depleting only the Prdm16+ subset of cells is sufficient to ablate ROR γ t+ Treg induction and rule out redundant/contextual contributions from other MHCII+ ROR γ t+ APC subsets including ILC3 and eTAC.

We thank the referee for the suggestion. First, we wish to clarify that the Cd11c-Cre system is not inherently “leaky”; rather, it is expressed in multiple cell types at varying levels.

Currently, Prdm16-Cre mice are not available. However, we conducted additional experiments to confirm the precise identity of the pTreg-inducing APCs, which we have designated as tDC. As described above, in *Prdm16* ^{Δ ROR γ t} mice, where tDC were selectively depleted but MHCII+ ILC3 remained intact, both Hh7-2 and OT-II pTreg differentiation were severely impaired. These results highlight Prdm16-dependent tDC as the APCs driving T cell-mediated tolerance to microbiota and dietary antigens.

- What are the activation states, frequencies and numbers of ILC2 in the d+7kb mice? Given that loss of ILC3 populations has been reported to lead to reciprocal expansion of ILC2 (Spencer et al Science 2014) and ILC2 are potent inducers of GATA-3+ Treg via OX40L to restrain Th2 responses (Stockis et al Science Immunology 2024).

Our analysis revealed that the number of ILC2 in the mLN of Δ +7kb mice was increased (Reviewer Figure 4), indicating the observed loss of pTregs in Δ +7kb mice is not due to a disruption in ILC2. Regardless, as we observe expansion of Th2 cells in mice in which there is no disruption of ILC3 (*Rorc(t)* ^{Δ CD11c} and *Prdm16* ^{Δ ROR γ t} mice) (Extended Data Fig. 7g,h), it is unlikely that ILC2 have a major role in the observed phenotype.

Reviewer Figure 4. ILC2 number in the mLN of control and Δ +7kb mice.

- What is the evidence that the enhanced Th2 response at steady state (Extended Data Fig4) is a direct consequence of disrupted Treg induction? While this is a logical assumption there is no direct experiments to demonstrate this and thus, it cannot be ruled out that indirect effects due to disruption of type 3 circuits. Notably ROR γ t+ lymphocytes including ILC3 and γ dT cells have critical roles in modulating barrier tissue integrity, epithelial metabolism and nutrient transport and the microbiota which could feasibly alter responses to dietary antigens and result in altered Th2 responses in a Treg-independent manner. Does deletion of ROR γ t in FoxP3 lineage cell recapitulate this phenotype?

As mentioned above, although $\Delta+7\text{kb}$ mice indeed show alterations in both ILC3 and $\gamma\delta\text{T}$ cells, we also utilized *Rorc(t)^{ΔCD11c}* and *Prdm16^{ΔRORγt}* mice, in which ILC3 numbers remained unchanged, but tDC were barely detectable. In these models, we consistently observed disrupted Treg induction and enhanced Th2 responses at steady state (Extended Data Fig. 7e-h), suggesting that the Th2 skewing is likely due to the loss of tDC rather than indirect effects from disruption of type 3 circuits.

Additionally, previous work by (Ohnmacht et al., 2015) demonstrated that mice with ROR γt deletion specifically in the FoxP3 lineage also exhibited an enhanced Th2 response in the gut, supporting the notion that disrupted Treg induction directly contributes to the observed Th2 phenotype.

- The abstract states the authors demonstrate “ROR γt + APCs are required for differentiation of food antigen-specific pTregs”, which is arguably an overstatement. While OVA-induced oral tolerance is undoubtedly a useful model it is unclear if this is sufficient to support the authors interpretation that ROR γt + APC-induction of Treg is required for tolerance to food-antigens more generally. Do $\Delta+7\text{kb}$ mice exhibit evidence of food allergy or loss of tolerance to normal dietary antigens or any spontaneous pathology without the addition of a model antigen?

In 40-week-old $\Delta+7\text{kb}$ mice, we observed features indicative of spontaneous type 2 gastrointestinal pathology, including muscularis propria hypertrophy, increased small intestine length, and elevated serum total IgE levels (Extended Data Fig. 7i-l). These findings suggest a loss of tolerance to normal dietary antigens and support the notion that tDC play a critical role in maintaining immune homeostasis in the gastrointestinal tract, even in the absence of a model antigen.

- Of note the authors mention Prdm16+ cells as well as TCI both express Aire transcripts in their data set. As such that suggests the identity of these cells is in line with that of ROR γt + eTACs and not the previously described ROR γt + DC-like cell / TCII / TCIV. This should be discussed in the text to help readers reconcile the findings with the previous nomenclature.

We now include an explicit integration of all the original data (with standard quality controls, but no further filtering) from Akagbosu et al. (2022) alongside our aggregate murine data (Extended Data Fig. 4f,g). With that, we find robust classification of their previous dataset into ILC3, Prdm16+ cells, and Nrg1+ cells, just as we observe in our own experiments. However, the authors from 2022 did not provide the computational code they used to cluster their TC 1-4 populations, so we unfortunately cannot further reconcile how those specific subsets relate to what we report here.

- The authors should provide a direct comparison of the gene signatures of Prdm16+ cells from their murine scRNA seq with previously reported signatures for ROR γt + eTACs and TCII-IV populations. In addition a focused bubble plot comparing these signatures demonstrating expression of known distinguishing factors including *Il7r*, *Dpp4*, *Siglecg*, *Aire*, *Spi1*, *Itgb8*, *Itgax*, *Cd80/Cd86* etc would be especially useful.

While we cannot clearly reconcile specific TC populations, as mentioned above, we did notice multiple similarities between our described Nrg1_Pos population, and certain gene expression patterns described for the TC 1 subset, and our main text specifically mentions these observations. Their similarities are demonstrated with the bubble plot in Fig. 2b, and more extensively with the bubble plot in Extended

Data Fig. 4c.

- Moreover, it could be argued these data is in fact more in line with the previous reports that the ROR γ t⁺ APC pool constitutes ILC3 and two distinct populations of ROR γ t⁺ eTACs (e.g. Yamano et al J Exp Med 2019, Lyu et al Nature 2022). Comparison of transcriptional data sets with the annotations of data from these prior scRNA seq data sets would also prove a useful comparison to clarify commonalities and nomenclature.

We hope our revised manuscript now makes a much stronger case with gene expression, chromatin accessibility, and surface protein data, that this tDC population is not compatible with the ILC3 cell type. We instead chose to more closely compare with the data from Akagbosu et al. 2022 (Extended Fig. 4f,g) given certain similarities that we have commented on, but we otherwise deferred on making detailed comparisons with all prior hypotheses.

- A recent study from the Powrie lab (Gu et al Nature 2024) elegantly mapped Treg interaction networks in the gut and lymph nodes, with the data publicly available. Notably, they observed multiple MHCII⁺/mid ILC3 subsets that differentially localised with Tregs in different compartments. This presents another opportunity compare transcriptional signatures of the ILC3, TCI and Prdm16⁺ cells captured in this data and extrapolate where they may be communicating with Tregs in vivo. Imaging of Prdm16 or another proxy marker alongside Tregs in these tissues would also help to contextualise the findings and uncover the niche in which proposed interactions between these cell types occur.

We wholeheartedly agree with the Reviewer that spatially identifying the interaction of these tDC with naive T cells (along with whichever other cell types may be required within the niche) is an important next line of investigation. We kindly view this as outside the scope of our current manuscript, but it is a high priority for our future work.

References

- Bunyavanich, S., Schadt, E. E., Himes, B. E., Lasky-Su, J., Qiu, W., Lazarus, R., Ziniti, J. P., Cohain, A., Linderman, M., Torgerson, D. G., Eng, C. S., Pino-Yanes, M., Padhukasahasram, B., Yang, J. J., Mathias, R. A., Beaty, T. H., Li, X., Graves, P., Romieu, I., ... Weiss, S. T. (2014). Integrated genome-wide association, coexpression network, and expression single nucleotide polymorphism analysis identifies novel pathway in allergic rhinitis. *BMC Medical Genomics*, 7, 48.
- Chang, X., March, M., Mentch, F., Qu, H., Liu, Y., Glessner, J., Sleiman, P., & Hakonarson, H. (2023). Genetic architecture of asthma in African American patients. *The Journal of Allergy and Clinical Immunology*, 151(4), 1132–1136.
- Esterházy, D., Loschko, J., London, M., Jove, V., Oliveira, T. Y., & Mucida, D. (2016). Classical dendritic cells are required for dietary antigen-mediated induction of peripheral T(reg) cells and tolerance. *Nature Immunology*, 17(5), 545–555.
- Garcia-Etxebarria, K., Merino, O., Gaite-Reguero, A., Rodrigues, P. M., Herrarte, A., Etxart, A., Ellinghaus, D., Alonso-Galan, H., Franke, A., Marigorta, U. M., Bujanda, L., & D'Amato, M. (2022). Local genetic variation of inflammatory bowel disease in Basque population and its effect in risk prediction. *Scientific Reports*, 12(1), 3386.
- Josefowicz, S. Z., Niec, R. E., Kim, H. Y., Treuting, P., Chinen, T., Zheng, Y., Umetsu, D. T., & Rudensky, A. Y. (2012). Extrathymically generated regulatory T cells control mucosal TH2 inflammation. *Nature*, 482(7385), 395–399.
- Kedmi, R., Najjar, T. A., Mesa, K. R., Grayson, A., Kroehling, L., Hao, Y., Hao, S., Pokrovskii, M., Xu, M., Talbot, J., Wang, J., Germino, J., Lareau, C. A., Satpathy, A. T., Anderson, M. S., Laufer, T. M., Aifantis, I., Bartleson, J. M., Allen, P. M., ... Littman, D. R. (2022). A ROR γ ⁺ cell instructs gut microbiota-specific Treg cell differentiation. *Nature*, 610(7933), 737–743.
- Ohnmacht, C., Park, J.-H., Cording, S., Wing, J. B., Atarashi, K., Obata, Y., Gaboriau-Routhiau, V., Marques, R., Dulauroy, S., Fedoseeva, M., Busslinger, M., Cerf-Bensussan, N., Boneca, I. G., Voehringer, D., Hase, K., Honda, K., Sakaguchi, S., & Eberl, G. (2015). The microbiota regulates type 2 immunity through ROR γ ⁺ T cells. *Science*, 349(6251), 989–993.
- Saad, M. N., Mabrouk, M. S., Eldeib, A. M., & Shaker, O. G. (2019). Studying the effects of haplotype partitioning methods on the RA-associated genomic results from the North American Rheumatoid Arthritis Consortium (NARAC) dataset. *Journal of Advanced Research*, 18, 113–126.
- Schneider, C., O'Leary, C. E., von Moltke, J., Liang, H.-E., Ang, Q. Y., Turnbaugh, P. J., Radhakrishnan, S., Pellizzon, M., Ma, A., & Locksley, R. M. (2018). A Metabolite-Triggered Tuft Cell-ILC2 Circuit Drives Small Intestinal Remodeling. *Cell*, 174(2), 271–284.e14.
- Xu, M., Pokrovskii, M., Ding, Y., Yi, R., Au, C., Harrison, O. J., Galan, C., Belkaid, Y., Bonneau, R., & Littman, D. R. (2018). c-MAF-dependent regulatory T cells mediate immunological tolerance to a gut pathobiont. *Nature*, 554(7692), 373–377.

Referees' comments:

Referee #1 (Remarks to the Author):

I thank the authors for their detailed responses to the original comments, particularly for the additional experiments using Prdm16 Δ ROR γ t mice, and for the more in-depth characterization of the novel APC. These changes have added significantly to the impact of the paper and increased the understanding of how these cells might contribute to Treg mediated oral tolerance to protein antigen.

Some issues remain:

1) In my opinion, the most important of these is that the novel APC should not be referred to as a "tolerogenic DC" as the authors now define them. First, it is now considered inappropriate to use a functional term to label DC as if they were a distinct lineage, as it is clear that DC of all lineages can exist in a functional spectrum. More to the point, the current work does not show direct evidence for the APC belonging to the DC lineage at all. Certainly they share some features of conventional DC and the new, epigenetic studies are interesting. However many of the phenotypic properties are not specific to DC (eg CD11c, MHCII, CSF1R) and their expression by the new APC was heterogeneous. To identify the APC as bona fide DC, it will be necessary to show eg that they are derived from a pre-cDC in a Flt3L dependent manner in vivo and in vitro. In the absence of this evidence, the term "tolerogenic DC" should be omitted.

We agree that identifying the exact dendritic cell precursor, as well as whether the developmental branch point occurs before or after cDC1 or cDC2 specification, will require careful and extensive lineage tracing. (Indeed, we have already started such long-term experiments but kindly view these as beyond the scope of this manuscript.) We find our current results across gene expression, protein markers, and chromatin accessibility (including at the Flt3 locus shown in Extended Data Fig 5g, and emphasized by the Reviewer) to be entirely consistent with DC and incompatible with ILC3.

This field has undergone rapid terminology expansion since 2022, including eTAC, JC (1-4), TC (1-4), and R-DC-like, reflecting ongoing debates and varying interpretations. However, none gained wide acceptance nor resolved biological complexity, most likely since others could not reproduce those categorizations. In this context, our study presents strong evidence that Prdm16- and ROR γ t-dependent tDC are the Treg-inducing APCs, using Prdm16 Δ ROR γ t mouse experiments that others can build upon. We have also established a flow cytometry panel that clearly distinguishes tDC from other ROR γ t⁺ APCs. As our sc-RNA-seq genotyping and cytometric phenotyping are readily reproducible, this offers a valuable resource for future investigations and streamlines follow-up nomenclature. We believe this can minimize confusion and allow for more focused progress.

Finally, the use of functional terms in immunology is well-precedented. While "Thetis cell" and "Janus cell" are perhaps colorful nomenclature, such naming is not informative, especially to future trainees. In fact, functional naming of cell populations (such as regulatory T cells, cytotoxic T cells, helper T cells, natural killer cells, etc.) is often the best didactic; the name immediately and objectively informs the student. We hope "tolerogenic DC" will similarly foster clarity. In addition, while these cells may have the potential for plasticity, our previous study showed clearly that they are not only necessary, but also sufficient, to induce pTreg cells and no other T cell phenotype. Combined with the other results in our

2022 paper, we concluded that there are APCs dedicated to distinct T cell programs, although that does not rule out plasticity of such cells under some circumstances.

2) While the authors compare their APC with ILC3 in some detail, the reliance on "tolerogenic DC" as a definition has prevented a full discussion of how the current cell might be related to the other, recently described Rorgt⁺ APC populations such as Thetis/Janus cells. For instance, despite the added transcriptional data shown in Supplemental Table 1, this material is extremely complex, making it difficult still to see information on the absolute levels of expression of eg avb8 integrin, AIRE

We agree with the Reviewer that there is already some consensus that the Treg-inducing APC will harbor a distinct gene expression signature. We also agree that the raw data offered in Supplemental Table 1 would be cumbersome for a reader to query for such a signature, but we instead refer the Reviewer to Figure 2b where we have explicitly measured relevant gene expression features, as recommended. This plot includes *Itgb8*, *Aire*, *Rorc*, *Cd40*, *Itgax*, *Ccr7*, *Zbtb46*, and *H2-Ab1*. We also kindly refer the Reviewer to our Extended Data 4c, which includes additional genes to facilitate direct comparison to the Janus/eTAC cell subsets reported in 2021 (Wang et al., 2021) (such as in their Fig 4c, 5c, and 6a) as well as the Thetis cell subsets reported in 2022 (Akagbosu et al., 2022) (such as in their Fig 2d, 3g-h, and 4a). Our plot explicitly includes *Nrg1*, *Nrxn1*, *Nrn1*, *Ncam1*, *Trp63*, *Tnfrsf11b*, *Twist1*, *Cd80*, and *Cd86*.

3) In this respect, how do the authors see the Nrg1_Pos cells with such a mixed transcriptional profile fitting into the overall picture and their relationship to the Prdm16⁺ APC?

The Nrg1_Pos population is certainly intriguing, since it is the other non-ILC cluster with *Rorc* expression within the mLN, and since it harbors expression of multiple neurotrophic factors, as well as *Aire* (as currently described within the text). It is entirely possible that this population developmentally precedes or follows the tDC population in ontogeny, interfaces with tDC on some level, or that it stands alone in development and function. However, we did not observe a homologous Nrg1_Pos population in human mLN nor in other tissues, nor did we perform any experimentation to specifically target this population in mice. Therefore, we chose to refrain from making any such claims.

4) I found it very difficult to understand the first paragraph of the section "Characterization of Prdm16⁺ APCs as tolerogenic DC", especially the second sentence. The authors might be able to clarify what was done here to integrate the different databases.

We utilized a data integration algorithm that was developed by the Satija lab and has since become widely implemented by computational biologists (Stuart et al. Comprehensive Integration of Single-Cell Data, *Cell* 2019. PMID 31178118). As recommended, we have now added this citation for clarity.

5) As noted above, the new phenotyping data provided in Figures 2 and Extended Data 6 often describe heterogeneous expression of non-specific markers or morphology and conclude that the APC are "similar to DC" or "closely resembling DC" as opposed to proving they are a bona fide DC. The interpretation of these data should be more cautious.

We appreciate the Reviewer's concern. We have indeed been careful in our interpretation and wording, using terms like "sharing more similarities with cDC" and "resembled cDC" to describe phenotypic

features without prematurely asserting lineage identity. We hope that the referee agrees that the results shown clearly distinguish the Prdm16⁺ cells from ILC3. We conclude only at the very end of this section that all our collective findings support their designation as tolerogenic dendritic cells. As noted above, we hope that establishing tDC as a clear nomenclature, as well as our phenotyping strategy, will facilitate further research and enable others to build on our findings.

6) The new Prdm16 Δ ROR γ t mice are interesting and potentially important. While I accept it seems unlikely, can the authors be sure that this approach might target other, non-immune cells expressing Rorgt and Prdm16?

Thank you for raising this point. While we acknowledge the theoretical possibility of Prdm16 Δ ROR γ t targeting non-immune cells, available data suggest that ROR γ t expression is highly restricted to immune cell populations. Although another isoform, ROR γ , is expressed in certain non-immune tissues, such as kidney, liver, muscle, and brown adipose, our ROR γ t-Cre mouse model specifically targets ROR γ t-expressing cells, without affecting ROR γ -expressing cells. Therefore, the effects observed in Prdm16 Δ ROR γ t mice are primarily due to immune-specific deletion, rather than off-target effects in non-immune cells.

7) Is the enteropathy seen in older Δ +7kb mice accompanied by infiltration with eosinophils?

We did not observe a notable infiltration of eosinophils within small intestines of older Δ +7kb mice.

8) While interesting, the additional information provided for OTII transfer experiments shown in Figure 3 and Extended Data 8 seems incomplete for concluding there is a Th2-specific process in the absence of the Prdm16⁺ APC. For instance, there are still virtually no Th2 cells in LP of the Prdm16 Δ ROR γ t mice on d12, with there appearing to be a much greater effect on Th17 and Th1 cells. Furthermore, no data are shown for day5 in either MLN or LP of the Prdm16 Δ ROR γ t mice, which would be needed to make a direct comparison with the Δ +7kb and MHCII Δ ROR γ t mice.

We would like to clarify that we do not claim that there is a unique Th2-specific response in the absence of Prdm16⁺ APCs. In the LP of Prdm16 Δ ROR γ t mice on day 12, Th2 cells were still present (with an average proportion of approximately 10%), although Th17 and Th1 cells were more abundant (each averaging around 30%). Indeed, data from other mutant models (Δ +7kb, ROR γ t Δ CD11c, and MHCII Δ ROR γ t mice) demonstrate that in the absence of functional Prdm16⁺ APCs, different effector Th cell populations are affected to varying degrees depending on the mouse model and tissue examined, as we noted in the manuscript: “Dysfunction of these APCs correlates with intensified effector Th cell responses, although the specific Th cell subset favored depends on the local tissue environment.” (Last sentence of the first paragraph of the section “tDC required for oral tolerance”). Furthermore, in our oral tolerance experiments, we observed a dominant Th2 response primarily because we used Th2-skewing adjuvants, alum and cholera toxin.

Additionally, in the revised manuscript, we have now included day 5 data from the mLN of Prdm16 Δ ROR γ t mice. The results also show that there were few OT-II pTregs in the mutant mice, and, instead, OT-II T cells exhibited Th2 and Tfh phenotypes (Fig. 3c and Extended Data Fig. 8c).

9) Although I appreciate the difficulties in studying the APC activity of the Prdm16⁺ cells in vitro and the complications raised by the lack of Prdm16-cre mice, it remains the case that the work does not directly prove a role for these cells as APC in vivo. Using the Prdm16 Δ ROR γ t mice in an allergy tolerance experiment might help to some extent, but otherwise, it would be appropriate to acknowledge this limitation more explicitly.

In our previous submission, we provided data from the MHCII Δ ROR γ t mice in both the OT-II transfer experiment and the asthma-oral tolerance experiments. Although MHCII Δ ROR γ t mice affect both tDC and ILC3, the use of Prdm16 Δ ROR γ t in the OT-II transfer experiment clearly demonstrates that tDC, rather than ILC3, are responsible for inducing pTregs. By process of elimination, we believe this provides substantial evidence supporting the role of tDC as APCs *in vivo*. While cells expressing both ROR γ t and Prdm16 may include more than one cell subpopulation, there are no current tools available to further subdivide these cells for functional analyses.

Additionally, in the revised manuscript, we have included new data using Prdm16 Δ ROR γ t mice in the asthma-oral tolerance experiments. The results show that OVA-pre-fed Prdm16 Δ ROR γ t mice exhibited increases in lung eosinophils and Th2 cells comparable to those observed in non-tolerized mutant mice (Fig. 4g and Extended Data Fig. 9i). These findings further support our conclusion regarding the role of tDC as APCs *in vivo*.

10) The associations between Prdm16 SNPs and human disease are intriguing, but the nature of the human cells that are compared with the mouse APC remains somewhat unclear. As well as the data in the Supplemental Table being extremely dense, many of the individual genes highlighted as being expressed by the human Prdm16⁺ cells in the intestinal or LNs are not particularly typical of DC - eg Prox1, IL3R, Kit, PIGR, IDO2, DLGAP1, MACC1.

We agree that these genes have not previously been regarded as typical for conventional DCs. However, this is somewhat expected, since this list highlights differentially expressed genes for tDCs in mouse and human, as compared to all other APC subsets (including genes that differentiate tDC against other DC subsets).

In fact, *PIGR* and *KIT* were already observed in work from the Colonna lab (Ulezko Antonova et al., 2023) looking at ROR γ t⁺ human APCs (their Fig 1b and 3c), and the same genes are mentioned in the paper reporting Thetis cell subsets (Akagbosu et al., 2022) (their Fig 3g). Another paper just recently published from Barbara Schraml's lab (Narasimhan et al., 2025) also describes ROR γ t-expressing dendritic cells conserved across species, and their Supplementary Table 2 highlights differentially expressed genes for the population they examine. That list notably includes *IDO2*, *IL3R*, *DLGAP1*, and *PROX1*. Their Supplementary Figure 10 also highlights *MACC1*. We believe that these independent observations of prominent gene expression within ROR γ t⁺ APCs corroborate what we observe within our own data.

Referee #2 (Remarks to the Author):

The revised manuscript by Liuhui Fu et al is much improved compared to the original. The authors further consolidate their identification of Prdm16⁺ tolerogenic DC cells (tDC) by employing additional tools include a Rorc-Cre x Prdm16^{flox} mice, a food allergy testing paradigm of oral tolerance and newly added multiome scRNA-seq and scATAC-seq analyses of mouse tDC and human scRNAseq data sets. In particular, I find the depletion of tDC upon the inactivation of Prdm16 and the resulting loss of (OTII) pTreg to be an incisive intervention. Overall the authors have satisfactorily answered most of my queries.

I have the following comments:

1) The authors would have been better served to have carried out the experiments in Figure 4 (allergen specific oral tolerance induction) on the Rorc-Cre x Prdm16^{flox} mice. This is after all the subset (tDC) under investigation in the manuscript. I would advocate at least repeating the food allergy study using the Rorc-Cre x Prdm16^{flox} mice.

Thank you for raising this important point. To address this, we have now included new data in the revised manuscript, showing results from asthma-oral tolerance experiments with Prdm16 Δ ROR γ t mice in order to provide more direct evidence for the role of tDC. Our findings indicate that OVA-pre-fed Prdm16 Δ ROR γ t mice displayed similar increases in lung eosinophils and Th2 cells as seen in non-tolerized mutant mice (Fig. 4g and Extended Data Fig. 9i). We believe this strengthens our conclusion about the function of tDC in tolerance induction.

2) Prdm16⁺ tDCs appear to be sustained in numbers during development albeit with sharply reduced frequencies over time (Ext. Fig. 6c). It is unclear on which gut tissues these studies were carried out (which intestinal regions). Are there developmentally-regulated regional differences in tDC frequencies/numbers? While tDCs appear closely related to the previously described Thetis cells by the Chrysothemis Brown group, the latter were described as developmentally regulated. Is this simply a dilution effect (same numbers, lower frequencies?).

The data presented in Extended Fig. 6c were obtained from mLN (mesenteric lymph nodes), as noted in the figure legend. Regarding the possibility of developmentally regulated regional differences, we agree that exploring tDC in other tissues, such as different intestinal regions, lungs, skin, and their corresponding draining lymph nodes, would offer valuable insights. However, this falls beyond the scope of the current study, and we plan to investigate this in future research.

We appreciate the referee's curiosity as to how our results may compare to those of Chrysothemis Brown's group, which described Thetis cells as being developmentally regulated in the mLN and appearing primarily in the neonatal period. It is important to note, however, that their quantification was based solely on gating strategies lacking any specific transcription factors. In contrast, we quantify tDC using the transcription factor Prdm16. Our results indicate that tDC are present beyond the neonatal stage and persist into adulthood, although their relative frequency among CD45⁺ cells gradually declines over time (the dilution effect mentioned above). We believe that our findings align with existing literature showing that adult mice are equally capable of inducing pTreg-mediated tolerance to gut microbiota and food antigens (Esterházy et al., 2016; Kedmi et al., 2022; Xu et al., 2018). In our manuscript, experiments on microbiota- and food antigen-specific pTreg induction and oral tolerance

were also conducted using adult mice, suggesting that tDC maintain their functional persistence into adulthood.

Referee #3 (Remarks to the Author):

Fu et al have made significant revisions to their manuscript. The additional granularity of analysis of their single cell RNA seq and by flow cytometry adds important information to address the identity of these cells and to allow comparison with other RORgt+ APC populations. Moreover, the generation of new mice targeting Prdm16 in RORgt+ cells (coupled with alternative CD11c Cre targeting of RORgt) together appears to support the conclusion that the pTreg-inducing capacity is contained within this population of cells, which the authors term “tolerogenic DCs”. Nonetheless, there remain some points that need further analysis and clarification, while further attempts to reconcile their data with prior studies in the text is critical to maximise understanding of this complex and rapidly developing area for a non-expert readership.

- In new data the authors demonstrate that deletion of Prdm16 in RORgt-expressing cells phenocopies the changes in Treg reported previously, and in both CD11c-intrinsic RORgt-deletion and RORc 7kb mice. This is the most definitive proof in the manuscript that this population of cells may directly promote intestinal Tregs. One important caveat of this experiment and its interpretation is the authors state “Prdm16+ APCs were barely detectable”. While at face value this appears true it could be misleading based on the data currently provided as the authors utilize the targeted gene product (Prdm16) as the only phenotypic marker to identify these cells, hence this is a self-fulfilling prophecy and makes an assumption that Prdm16 is essential for the development or persistence of these cells. To distinguish between the successful deletion of Prdm16 in these cells and loss of this population per se the authors must utilize alternative gating strategies (building on gating and marker combinations shown in Figure 2f+g). For example, can they demonstrate a loss of RORg+ MHCII+ cells within the CXCR6-IL-7R- without using Prdm16 directly to quantify, that would further support a “loss” of these cells.

The referee is correct in pointing out that loss of tDC in Prdm16 Δ ROR γ t mice does not prove that the population was lost. However, we similarly observe the unique absence of these cells in CD11c-Cre/ROR γ t conditional mutant mice using both flow cytometry and scRNAseq. In addition, we did not explicitly state that Prdm16⁺ APCs were lost. Rather, we described them as “barely detectable” using Prdm16 as a marker.

We recognize, however, that without Prdm16 as a marker we are currently unable to reliably identify tDC, as other surface markers do not uniquely define this subset. Nevertheless, the use of Prdm16 Δ ROR γ t mice has provided compelling evidence of at least a loss of tolerogenic function in Prdm16⁺ APCs, as demonstrated by the abolition of *Hh*-specific pTreg differentiation and the increase in inflammatory Th17/Th1 cells.

To ensure more precise language and avoid any potential misinterpretation, we have revised the manuscript as follows:

“...while Prdm16⁺ APCs were barely detectable when identified using Prdm16 as a marker.” (page 9)
“...accompanied by an increase in inflammatory Th17/Th1 cells, suggesting at least a loss of tolerogenic function in Prdm16⁺ APCs.” (page 9)

We believe that these changes clarify our interpretation and more accurately convey our findings.

- The authors should make further attempts to integrate their findings within the existing framework of nomenclature for this emerging population of ROR γ ⁺ APCs in the results and discussion. The proliferation of “new cell types” makes it difficult for non-expert readers to follow the latest developments in this field and can “muddy the waters”. As a logical nomenclature has recently been proposed (Abramson et al Nat Rev Imm 2024), I would suggest applying this where possible and discussing the findings in the framework of this discussion (i.e. if Prdm16⁺ and Nrg1⁺ cells express AIRE, should they not be referred to as eTACs, if not further explanation is needed as to why they don't fit this definition, see also next point below). Additionally, they should include discussion of the recently published study by Canesso et al in Science and potentially also reconcile/compare their findings with parallel studies in pre-print (e.g. Parisotto & Brown).

Thank you for these suggestions. We note that a similar concern regarding nomenclature was also raised in the first comment of Referee 1, so please refer to our response there, which we hope addresses all points mentioned. We appreciate the reference to (Abramson et al., 2024) in Nat Rev Imm, which indeed attempts to organize and consolidate the rapidly expanding nomenclature surrounding ROR γ ⁺ APCs. We understand and respect the intention to bring greater clarity to this complex field.

However, since 2022, the field has witnessed a rapid proliferation of terminologies, including ILC3, eTAC, JC (1-4), TC (1-4), and R-DC-like, reflecting ongoing debates and divergent interpretations. While Abramson et al. made a valuable effort to integrate these findings, the confusion and controversies persist, as evidenced by their Fig. 2, which illustrates that at least three different populations (ILC3, JC2/JC3/TC2/TC3, and TC4/cDC2B) can induce pTregs, highlighting the complexity and unresolved nature of this issue.

In light of this, our study provides compelling evidence that Prdm16⁻ and ROR γ ⁺-dependent tDC are the specific APC population responsible for inducing T cell-mediated tolerance to microbiota and food antigens. We have also developed a flow cytometry panel that phenotypes tDC among the diverse ROR γ ⁺ APCs in lymph nodes and intestinal tissues. This phenotyping strategy offers a reliable and reproducible method that will facilitate future research in this area, enabling others to follow up and build on our findings with greater ease and precision. We believe that the functional clarity and phenotypic precision provided by our study offer a more straightforward solution to the current confusion than a debate over nomenclature.

Regarding (Canesso et al., 2024) in Science, we appreciate the importance of acknowledging recent literature. However, their conclusions appear to directly contradict both our findings and those from pre-prints by Rudnitsky & Kedmi and Parisotto & Brown, by proposing that cDC1 are the primary inducers of food antigen-specific pTregs, which conflicts with their own earlier findings from 2016 (Fig. 7a-c) (Esterházy et al., 2016) demonstrating that oral tolerance was intact in mice with IRF8-deficient cDC. We believe that including a detailed discussion of their findings would further complicate the current debate. However, we are open to revisiting this comparison in future studies as the field progresses.

Similarly, we opted not to include a discussion of the pre-prints by Rudnitsky & Kedmi and Parisotto & Brown as they have not yet been peer reviewed. Since their findings and interpretations may change prior to final publication, premature comparisons could mislead readers or unintentionally contribute to further confusion.

For these reasons, we believe that by focusing on the functional and phenotypic characterization in our study, we are providing a clear and practical framework for future research, without becoming entangled in ongoing nomenclature debates.

- On similar lines - one thing that remains unclear is how heterogeneous Prdm16⁺ RORgt⁺ APCs are. The authors refer to them as “tolerogenic DCs”, yet they do not express CD11b and only a small proportion express CD11c (despite presumed effective CD11c-Cre targeting) as per their flow cytometry. In addition, this cluster exhibits AIRE expression, and to some degree CCR6 suggestive of the previously described phenotype of eTACs (and/or TC1 as per Brown et al), rather than DC-like RORgt⁺ APC (which were reported to lack both of these). This raises the possibility that this cluster is heterogeneous and that Prdm16 marks both an eTAC and a RORgt⁺ DC population. This should be investigated with granularity by further sub-clustering of the scRNA seq data set and/or further flow cytometry analysis using previously reported markers defining these different subsets (as requested in first review). It might be expected that combinations of CD11c, AIRE and CCR6 as well as other novel TC/eTAC markers reported would split this population into two discrete cell types, and that Prdm16 may be a common signature for multiple non-ILC RORgt⁺ APCs.

We would like to clarify that tDC are not completely negative for CD11b. In fact, approximately 50% of tDC express CD11c and about 20% express CD11b in our flow analysis. The reason that CD11c-Cre targets tDC is not due to effective targeting, but because CD11c is expressed earlier in the lineage and is likely partially downregulated in mature tDC. We have added this clarification in the revised manuscript: “Notably, CD11c and CD11b expression was limited to a portion of Prdm16⁺ APCs, suggesting that there is partial downregulation of CD11c during lineage progression.” (Page 8)

We appreciate the importance of understanding the heterogeneity of Prdm16⁺ APCs, and we fully intend to investigate this further in future studies. However, as stated above, we believe that including a detailed discussion of the different nomenclature would further complicate the current debate, especially given the ongoing confusion and controversy in this field.

- Another interesting observation in the manuscript that could help reconcile confusion in the field is the observation that d+7KB, 11c-intrinsic RORgt deficient mice and RORgt-intrinsic Prdm16 deficient mice all failed to exhibit increased Th17 responses or spontaneous intestinal inflammation in the absence of *H. hepaticus* colonization (Extended Data Fig 7). While this may indicate the endogenous microflora is not sufficient to drive these effector responses, it is interesting that some of the initial studies that described MHCII⁺ ILC3 regulation of CD4⁺ T cells proposed direct suppression of microbiota-responsive Th17 by MHCII⁺ ILC3 in vitro and ex vivo and reported that restricted expression of MHCII via RORc Cre, but not CD11c Cre, suppressed Th17 responses without effects on Treg (Hepworth et al Nature 2013, Science 2015). This suggests one integrated model that might reconcile these two sets of studies whereby MHCII⁺ ILC3 can suppress activated microbiota-elicited Th17 effectors, and other DC-like or eTAC RORgt⁺ APC populations preferentially induce pTreg. This may warrant further discussion, and as per my other comments would help to try and reconcile and integrate the findings of these different studies for the readership.

The lack of spontaneous Th17 responses in our mutant mice in the absence of *H. hepaticus*, combined with the presence of increased Th17 responses in these mice after *H. hepaticus* colonization, clearly

demonstrates that these effector responses are microbiota-dependent. One possible explanation for the discrepancies in reported results is the variation in *Helicobacter* species abundance across different animal facilities, where some are free of *Helicobacter* species, while others have high levels of *Helicobacter* or similar pathobionts.

We appreciate the suggestion to integrate our data with the model proposed by Hepworth et al. (Nature 2013, Science 2015) from the Sonnenberg group, which posited that MHCII⁺ ILC3 can suppress Th17 responses without affecting Tregs. However, several points warrant consideration:

1. Their conclusion of "without affecting Tregs" was based on using FOXP3⁺ as the sole marker for Tregs. This approach included both tTregs and pTregs within their gating strategy, potentially masking key differences, as the presence of tTregs would obscure any selective impact on pTregs.
2. Interestingly, in 2022, the Sonnenberg group, along with our group and the Brown & Rudensky groups, reported that MHCII Δ ROR γ t mice exhibited a loss of microbiota-dependent ROR γ t⁺ pTregs, which provides an alternative explanation for the increased Th17 responses. Notably, the Sonnenberg group interpreted this as evidence that ILC3 induces pTregs, which directly contrasts with our current conclusion that Prdm16⁺ tDC but not ILC3 are responsible.

Given the ongoing controversy and the contradictory interpretations in the field, we believe that integrating our findings with this model would add more complexity without providing definitive clarity.

- The issue raised by multiple reviewers that direct evidence of antigen presentation by Prdm16⁺ APCs, still holds. In the revisions the authors use MHCII-dRORc mice to directly address antigen-presentation, however these experiments still suffer from the caveat that these mice also hit ILC3. Nonetheless, as they mention isolation of Prdm16⁺ cells was technically challenging to address this question *ex vivo*, and a Cre model to exclusively target these cells is not currently available this is understandable, but one that should be acknowledged as a limitation of the study.

We appreciate the reviewer's concern regarding the use of MHCII Δ ROR γ t mice due to the involvement of ILC3. However, our experiments have already demonstrated that tDC, but not ILC3, are required for inducing pTregs. Therefore, although MHCII Δ ROR γ t mice target both tDC and ILC3, by process of elimination, we believe the data provide sufficient evidence for antigen presentation by tDC. We acknowledge that the ideal approach would be to use a more specific Cre model. However, in the absence of such a tool, we believe the current evidence is sufficiently robust to conclude that the tDC are required for induction of pTregs. We cannot yet conclude that these cells are sufficient for this process, and there is a possibility, albeit remote, that more than one cell type is needed for the generation of the pTregs. This question requires the use of a Prdm16-cre driver strain for expressing MHCII uniquely in non-ILC3, but we feel this is beyond the scope of the current study.

- Related to this point the authors should demonstrate whether other aspects of Prdm16⁺ APC "tolerogenic function" is lost in the +7kb mouse, such as Itgb8 expression and ability to cleave TGF-beta which could potentially be as important mechanistically as direct antigen presentation.

Beyond the clearly impaired Treg induction in +7kb mutants, as well as the significant loss of the tDC population that we describe, it is difficult to pinpoint how the few remaining tDC in the mutants may be functionally altered. We examined “pseudo-bulk” normalized gene expressions of +7kb mutant tDC, correlating those to measurements in control mice tDC (Reviewer Figure 1). This illustrates that the majority of genes correlate relatively well along the diagonal, with perhaps *Prdm16* slightly trending as an outlier. The decrements in *Itgb8*, *Itgax*, and *Rorc* expression are even less impressive. Given the lack of any clear result, and especially since this analysis relies on a very small population of remaining cells, we do not think this would be informative for the general reader.

- Aspects of the initial ILC3 data require further explanation to the reader, as there is potential for confusion, and in some places interpretations and conclusions should be reworded or tempered. For example, the 7kb deletion in the *Rorc* locus leads to a “loss” of CCR6+ ILC3, yet this is not recapitulated in the scRNA seq data set. As mentioned by the authors this is likely because deletion of ROR γ t in mature ILC3 results in loss of phenotype, without loss of cells per se (e.g. as reported in the Withers and Fiancette papers cited), but to help the non-expert reader reconcile these observations this would be useful to also mention in the text of the results when discussing Figure 2c+d.

We would like to clarify that the +7kb deletion in the *Rorc* locus indeed leads to a decrease of CCR6⁺ ILC3 in the gut. In the previous submission, the scRNA-seq data did not explicitly show CCR6⁺ ILC3 but rather the overall ILC3 population. To address this, we have included new data in our revised manuscript (Extended Data Fig. 4e), specifically measuring *Ccr6* expression across the ILC3 cluster.

- Conversely, the authors use the observation that ILC3 are not lost in scRNA seq in Figure 2 to suggest Prdm16⁺ population are more likely responsible for tolerogenic function. However, one key piece of information lacking in the first figures of the manuscript is whether +7Kb mutants lose MHCII expression on dysfunctional ILC3, or on residual Prdm16⁺ cells. Given ILC3 are seemingly retained but lose phenotypic markers, including MHCII as per previous publications, the authors should directly address this point and temper their interpretation of the data at this early point in the manuscript. Indeed, the authors have now provide this data in Reviewer Figure 3 which shows MHCII expression is reduced on both subsets, thus it is critical that this data be included in the manuscript (along with representative histograms/flow plots) to address this important point.

We would like to emphasize that while the Δ 7kb single-cell RNA-seq data was indeed helpful in narrowing down potential candidates for tolerogenic APCs, the definitive evidence supporting Prdm16⁺ APCs as the candidate tolerogenic APCs comes from the ROR γ t Δ CD11c single-cell RNA-seq data, which demonstrated a complete loss of the Prdm16⁺ cluster without affecting other clusters.

We agree that the Δ 7kb mutation affects both ILC3 and tDC populations to varying degrees. However, we believe that overemphasizing these intermediate effects at this point in the manuscript would complicate the narrative and confuse the reader, potentially detracting from the clear logical progression leading to the identification of Prdm16⁺ APCs as the key tolerogenic population.

- Further to this point, in response to my initial comment the authors argue in their rebuttal that “there was little effect on ILC3 or LTi function” and that data from the d+7kb mice are “most consistent with a [...] role for ROR γ t in the function of mature tDC, but not ILC3”. I’m not sure how these points can be supported given the loss of phenotype and MHCII (and thus potential APC function) in ILC3 in this model and care should be taken not to make sweeping statements that cannot be supported by individual models/data sets. In line with this I would generally ask that the authors temper some of the wording and interpretations in earlier parts of the study that somewhat pre-empt conclusions that can only be fully supported when considering the totality of the data and models in the manuscript e.g. page 7 regarding prdm16⁺ APCs “suggest that these cells have tolerogenic function in response to the microbiota” – at this point in the manuscript no direct evidence for this has been provided, only correlative predictions.

We agree that it is essential to avoid overstatements and ensure that conclusions are consistent with the totality of the data. To address this, we have revised the wording to provide a more balanced interpretation:

“.....indicating that ROR γ t-dependent lymphoid tissue inducer (LTi) cells in mutant mice maintain sufficient functionality to support lymphoid organ development.” (page 4)

“These results suggested that mature ILC3 in the LILP of Δ +7kb mice retain functional capacity in response to *C. rodentium*.....” (page 4/5)

“..... and suggest that these cells are likely candidates for tolerogenic APCs in response to the microbiota.” (page 7)

- The text addressing links to GWAS studies in the results section would be more appropriate for the discussion, suggest to move.

We utilized the “NIH National Human Genome Research Institute Catalog of Human Genome-Wide Association Studies” to systematically identify the SNPs of PRDM16 relevant to human inflammatory and autoimmune diseases. Each referenced study identifies precise SNP loci which were the output of that objective database search. Therefore, we chose to mention this alongside other results, rather than within our discussion which provides speculation toward future work.

References

- Abramson, J., Dobeš, J., Lyu, M., & Sonnenberg, G. F. (2024). The emerging family of ROR γ t⁺ antigen-presenting cells. *Nature Reviews. Immunology*, *24*(1), 64–77.
- Akagbosu, B., Tayyebi, Z., Shibu, G., Paucar Iza, Y. A., Deep, D., Parisotto, Y. F., Fisher, L., Pasolli, H. A., Thevin, V., Elmentaite, R., Knott, M., Hemmers, S., Jahn, L., Friedrich, C., Verter, J., Wang, Z.-M., van den Brink, M., Gasteiger, G., Grünewald, T. G. P., ... Brown, C. C. (2022). Novel antigen-presenting cell imparts Treg-dependent tolerance to gut microbiota. *Nature*, *610*(7933), 752–760.
- Canesso, M. C. C., Castro, T. B. R., Nakandakari-Higa, S., Lockhart, A., Luehr, J., Bortolatto, J., Parsa, R., Esterházy, D., Lyu, M., Liu, T.-T., Murphy, K. M., Sonnenberg, G. F., Reis, B. S., Victora, G. D., & Mucida, D. (2024). Identification of antigen-presenting cell-T cell interactions driving immune responses to food. *Science (New York, N.Y.)*, eado5088.
- Esterházy, D., Loschko, J., London, M., Jove, V., Oliveira, T. Y., & Mucida, D. (2016). Classical dendritic cells are required for dietary antigen-mediated induction of peripheral T(reg) cells and tolerance. *Nature Immunology*, *17*(5), 545–555.
- Kedmi, R., Najjar, T. A., Mesa, K. R., Grayson, A., Kroehling, L., Hao, Y., Hao, S., Pokrovskii, M., Xu, M., Talbot, J., Wang, J., Germino, J., Lareau, C. A., Satpathy, A. T., Anderson, M. S., Laufer, T. M., Aifantis, I., Bartleson, J. M., Allen, P. M., ... Littman, D. R. (2022). A ROR γ t⁺ cell instructs gut microbiota-specific Treg cell differentiation. *Nature*, *610*(7933), 737–743.
- Narasimhan, H., Richter, M. L., Shakiba, R., Papaioannou, N. E., Stehle, C., Ravi Rengarajan, K., Ulmert, I., Kendirli, A., de la Rosa, C., Kuo, P.-Y., Altman, A., Münch, P., Mahboubi, S., Küntzel, V., Sayed, A., Stange, E.-L., Pes, J., Ulezko Antonova, A., Pereira, C.-F., ... Schraml, B. U. (2025). ROR γ t-expressing dendritic cells are functionally versatile and evolutionarily conserved antigen-presenting cells. *Proceedings of the National Academy of Sciences of the United States of America*, *122*(9), e2417308122.
- Ulezko Antonova, A., Lonardi, S., Monti, M., Missale, F., Fan, C., Coates, M. L., Bugatti, M., Jaeger, N., Fernandes Rodrigues, P., Brioschi, S., Trsan, T., Fachi, J. L., Nguyen, K. M., Nunley, R. M., Moratto, D., Zini, S., Kong, L., Deguine, J., Peeples, M. E., ... Colonna, M. (2023). A distinct human cell type expressing MHCII and ROR γ t with dual characteristics of dendritic cells and type 3 innate lymphoid cells. *Proceedings of the National Academy of Sciences of the United States of America*, *120*(52), e2318710120.
- Wang, J., Lareau, C. A., Bautista, J. L., Gupta, A. R., Sandor, K., Germino, J., Yin, Y., Arvedson, M. P., Reeder, G. C., Cramer, N. T., Xie, F., Ntranos, V., Satpathy, A. T., Anderson, M. S., & Gardner, J. M. (2021). Single-cell multiomics defines tolerogenic extrathymic Aire-expressing populations with unique homology to thymic epithelium. *Science Immunology*, *6*(65), eabl5053.
- Xu, M., Pokrovskii, M., Ding, Y., Yi, R., Au, C., Harrison, O. J., Galan, C., Belkaid, Y., Bonneau, R., & Littman, D. R. (2018). c-MAF-dependent regulatory T cells mediate immunological tolerance to a gut pathobiont. *Nature*, *554*(7692), 373–377.

Referees' comments:

Referee #1 (Remarks to the Author):

The authors have provided some new data and extensive discussion of the reviewers' comments. Unfortunately however, the revised manuscript itself still fails to address some of the most important issues raised about how the authors describe the nature of the Rorgt+ APC and its place in context of other recently identified APC subsets. To me, the most critical of these issues remains the definitive identification of the current cells as "tolerogenic DC". As the authors say, DC heterogeneity is a contentious area, with much of the confusion having arisen because of imprecise approaches and lack of information on how populations identified by individual groups might relate to each other. By ascribing a new name to yet another subset without full characterization and lack of appropriate contextualization, the current work risks adding to, rather than clarifying this confusion. I strongly believe that the use of this nomenclature based on the current work is inappropriate and very unhelpful.

1) While the authors show convincingly that their APC are unlikely to be ILC3, as I noted in the previous reviews, their identification as DC remains much less definitive than the manuscript makes out; it is based mostly on assumptions from overlapping/inconsistent markers. Too many phrases such as "by process of elimination", "closely resembling" etc are not sufficiently precise for unravelling such a complicated issue. Interestingly the recent PNAS paper by the Schraml group now referred to by the authors points out how this could be done, using eg Flt3L and IL2Rg KO mice, expression of DC-subset specific markers at protein level. Here it should be noted that the Schraml paper finds the apparently analogous population of Prdm16+Rorgt+ APC to express SIRPa, which seems to be at very low levels in the current work, while unlike here, CD11c expression appears stable with age in the other work. In contrast, Rorgt protein expression is found to decrease with age as assessed by the reporter gene tracking used by Schraml. Although it would not necessarily be appropriate for the current authors to carry out additional experiments, consideration of these apparent discrepancies is needed and it is essential that the resulting gaps in the current work are acknowledged, with the results interpreted less conclusively.

While our results are consistent with those described in the recent Schraml paper in PNAS describing ROR γ t⁺ dendritic cells (including their Fig 1D mentioned by the Reviewer), we disagree with their interpretation that one cannot ascribe a distinct function to any subset and that function must always be context-dependent (concluded based on their *in vitro* experiments). We illustrate with Reviewer Figure 1 above how heterogeneous ROR γ t⁺ dendritic cells are no better resolved by previously established markers such as *Sirpa* (to use just one example mentioned by the Reviewer). This UMAP can be recognized as the exact same from our analysis of all combined murine sequencing data (Extended Data Figure 4f-g), here subsetting only dendritic cell populations. The heat map for *Rorc* is adjusted to be very sensitive (any positive gene expression appears red), with a similarly sensitive green heat map overlaid for *Sirpa*, such that any co-expression is immediately recognizable in yellow. If one used flow cytometry to sort double positives for ROR γ t and CD172a, it would capture the yellow cells seen within our tolDC population, but this would also clearly include many “contaminants” from other cell types such as cDC2B and Nrg1_Pos.

Reviewer Figure 2. ROR γ t expression in tolDC at different ages

In addition, we appreciate that the Schraml group has provided additional evidence—using *Flt3L* and *Il2rg* knockout mice—to support these cells as dendritic cells. In our study, we support this classification from a complementary angle, using chromatin accessibility profiling to show that tolDC exhibit epigenetic features consistent with bona fide DC. However, we fully agree that the most definitive approach to resolving the ontogeny of these cells will require careful and extensive lineage-tracing experiments. We explicitly acknowledged this point in the Discussion. In fact, we have already initiated such long-term lineage-tracing experiments, but we respectfully consider them to be beyond the scope of the current manuscript.

Regarding CD11c expression, we note that the Schraml paper uses CD11c as a gating marker for identifying ROR γ t⁺ DC. It is therefore expected that CD11c expression remains stably detected in their analysis. As for ROR γ t protein levels, our data do not conflict with the Schraml findings: as shown in Reviewer Figure 2, ROR γ t MFI in tolDC is indeed higher in 1-week-old mice compared to 12-week-old mice.

2) Of equal importance is the fact that many "tolerogenic" populations of DC have been proposed over the years, without any having stood the test of time. In each case, it was been shown that the relevant cell was plastic and could respond flexibly to environmental conditions, with more, rather than less confusion resulting because of attempts to characterize the hypothetical cell. Again, the Schraml work is instructive here, where the Prdm16⁺Rorgt⁺ APC was capable of inducing both Treg and effector T cells depending on the context.

We have added a sentence to the Discussion acknowledging the potential plasticity. However, as noted above, we believe that the population analyzed in the Schraml study likely included contaminating ROR γ ⁺ cDC2, which may have contributed to the observed effector T cell-inducing capacity. Further studies will be needed to clarify this point. It can also be misleading to interpret in vitro data, such as those from Schraml, as indicative of in vivo function. Our conclusions as to function are based solely on in vivo studies with both loss- and gain-of-function approaches using multiple genetic strategies. As shown in Kedmi et al. (2022), expressing MHCII exclusively in ROR γ ⁺ cells resulted in the differentiation of microbiota-specific pTreg cells and no other types of T cells. However, when MHCII was expressed in CD11c⁺ cells, which include not only the tolDC, but also cDC1 and cDC2, there was differentiation of both pTregs and Tfh cells. We cannot at this stage rule out that ROR γ ⁺Prdm16⁺ DCs can assume other functions in different environments and under diverse perturbation conditions, but in the context of homeostatic induction of T cells specific for food and microbiota antigen, we can conclude that they induce only pTregs and are, therefore, tolerizing DCs.

3) While I appreciate there may be space limitations, there is still too little specific discussion of how the current APC population may fit in context of the other ROR γ ⁺ APC which have been described recently. Readers will expect to see this and the novel findings on induction of tolerance deserve better contextualization.

We appreciate the reviewer's comment. We have now added a discussion of how the tolDC relate to previously described non-ILC ROR γ ⁺ APC subsets, including JC, TC, and ROR γ ⁺ DC.

Referee #2 (Remarks to the Author):

The authors have answered all my queries satisfactorily.

We sincerely appreciate the help in improving our work.

Referee #3 (Remarks to the Author):

The authors have provided a response to concerns raised in the previous round of review. In some cases my questions remain, in particular as to the addition of more granularity and direct mechanistic links between Prdm16 deletion and loss of Treg induction. Specifically, it remains unclear if deletion of Prdm16 in ROR γ ⁺ cells drives a loss of this cell population and its antigen presenting capacity, or rather dysregulates the regulatory capacity of a non-ILC3 ROR γ ⁺ APC cell type. Similarly, the relationship of these cells to previously reported TC/eTAC populations is not fully explored, despite the presence of published and validated flow cytometry markers and gating strategies. This makes it difficult to understand the degree of conceptual advance. While the identification of a gene associated with ROR γ ⁺ APCs is interesting it remains unclear the extent to which it is a common feature of some or all previously reported ROR γ ⁺ APC cell type(s), moreover the biological function and relevance of Prdm16 indicating cell-intrinsic regulatory functions (versus being a convenient marker of a subset of ROR γ ⁺ APCs) is not fully explored.

We respectfully disagree with the reviewer's suggestion that Prdm16 may merely serve as a convenient marker. We conditionally inactivated Prdm16 in ROR γ ⁺-expressing cells and observed a complete loss of adoptively transferred *Hh*-specific pTreg differentiation, accompanied by a marked increase in inflammatory Th17 and Th1 responses when *Prdm16* was inactivated in ROR γ ⁺-Cre mice. While we are currently unable to definitively determine whether tolDC are lost in these mice—due to the fact that Prdm16 is also the key marker used to identify this population—we can confidently conclude that tolDC in this context have at least lost their pTreg-programming function. These results strongly support a functional, cell-intrinsic role for Prdm16, rather than it being a passive or correlative marker.

Furthermore, we believe the conceptual advance of our study lies in establishing a functionally validated and clearly defined subset—tolDC—rather than building upon previously proposed, yet poorly resolved populations such as JC or TC. While we recognize potential overlap with these reported cell types, our goal was not to retrofit our findings into existing, ambiguous nomenclature, but rather to provide a more robust framework grounded in both phenotypic identity and functional relevance. Additionally, we have included a discussion of the potential relationships between tolDC and previously reported populations, in order to help readers better understand their place within the broader ROR γ ⁺ APC landscape.

Furthermore, I agree with Reviewer 1 that the terminology "tolerogenic DC" may not be warranted based on current evidence, and more generally there are a number of speculative statements and interpretations I feel should be avoided, including several new statements speculating about lineage relationships (Page 8, Line 245) , potential loss of tolerogenic function without direct evidence (Page 9, line 265)etc.

We appreciate the reviewer's concern and have carefully revisited the statements in question. We believe that both are supported by our data and are appropriately worded. First, our previous work showed that Cd11c-Cre-mediated deletion of MHCII or CCR7 completely abolished pTreg generation, and in the present study, Cd11c-Cre-mediated deletion of ROR γ ⁺ also led to a complete loss of pTregs. These results demonstrate that the relevant APCs have expressed CD11c at some point in their developmental history. Then our flow cytometry analysis shows that approximately 50% of tolDC in the mLN express CD11c, indeed suggesting a partial downregulation of CD11c during lineage progression. Second, impaired pTreg differentiation in Prdm16 Δ ROR γ ⁺ mice provides compelling evidence for at least a loss of tolerance-inducing function in Prdm16⁺ APCs.

Nonetheless, the manuscript provides convincing evidence that the Treg inducing capacity of RORgt⁺ APCs is contained within a population of cells expressing Prdm16 in mice, with evidence of an analogous population in humans. The additional description of the +7kb regulatory region of Rorc also provides new insights into the regulation of this transcription factor and RORgt-expressing lymphocyte populations, while the authors demonstrate across mouse models that RORgt⁺ APC (requiring Prdm16⁺ cells or their functions) induce Treg in a model of oral tolerance.

We appreciate that the referee finds novelty in our results that combine both genetic and phenotypic characterization of the pTreg-inducing APCs.